# CGTFra: General Graph Transformer Framework for Consistent Inter-series Dependency Modeling in Multivariate Time Series

## Abstract

Transformers have emerged as dominant predictors in multivariate time series forecasting (MTSF), prompting an in-depth investigation into their limitations within this application. Firstly, the conventional temporal information for timestamps in MTSF suffers from the unavailability of future timestamps and the diversity of timestamp formats across real-world datasets, which poses a significant practical challenge and necessitates cumbersome adjustments for a unified forecasting model. Secondly, existing Variate Transformers, such as iTransformer, typically model inter-variate dependencies (IVD) predominantly within shallow self-attention layers, neglecting the critical requirement for deep-layer IVD modeling, thereby causing dependency information loss and difficulties in model optimization. We refer to this phenomenon as **inconsistent IVD modeling**. To address these limitations, CGTFra, is designed as a general Graph Transformer framework to promote consistent IVD modeling. Specifically, we introduce a frequency-domain masking and resampling method for feature enhancement that preserves periodic characteristics in the frequency domain. Additionally, by comprehensive analysis of the distinctions and connections between self-attention mechanisms of Variate Transformers and Graph Neural Networks (GNNs) in capturing IVD, a dynamic graph learning framework is integrated into the Transformer to explicitly model IVD in deep network layer. Crucially, we then propose a consistency-constrained alignment to strengthen the network to learn more robust IVD and temporal feature representations. The core design philosophy of CGTFra can be integrated into any existing Variate Transformer-based framework and CGTFra demonstrates superior predictive performance across 13 long- and short-term datasets with high computational efficiency. Code is available at https://anonymous.4open.science/r/CGTFra.

## 1 Introduction

Multivariate time series, such as traffic flow, are critical for forecasting the future dynamics of real-world systems. Multivariate time series forecasting (MTSF) is challenged mainly by two factors: the intricate temporal patterns of individual variables (intra-series dependency) and the dynamic dependencies among these variables (i.e., inter-series or inter-variate dependency), where one variable's fluctuation can affect the others. To illustrate, Figure 1 presents the raw traces of seven variables from the ETTh1 dataset, supplemented by their Pearson Correlation Coefficient Matrix (PCM) and Dynamic Time Warping (DTW) distance matrix, which reveal strong correlations and similarities between two pairs of variables: variable 0 with variable 2, and variable 1 with variable 3.

To achieve more accurate MTSF, numerous advanced methods have been developed, including CNNs, RNNs, MLPs, and GNN-based forecasters. More recently, Transformer-based networks have gained prominence due to their inherent strength in capturing long-range dependencies (Vaswani et al., 2017). However, after a comprehensive analysis of existing Transformer-based approaches, we argue that they still face two following significant limitations.

(1) **Over-reliance on Timestamps for Input Representation**. Existing methods typically employ learnable encodings derived from timestamps to capture temporal positional infor-

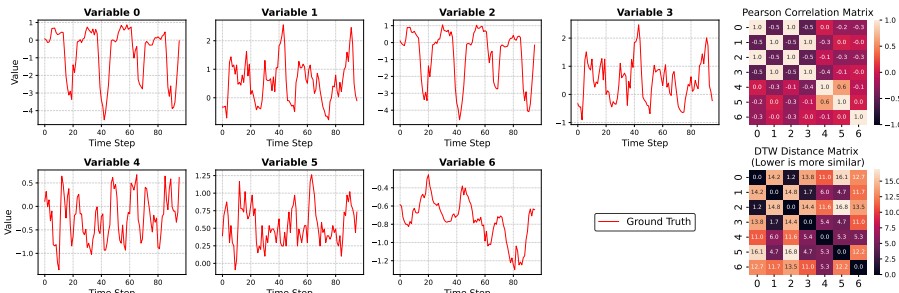

Figure 1: Intra- and inter-series dynamics on ETTh1 dataset. PCM and DTW are used to reveal inter-variable similarities and dependencies (See Appendix A.6 for more details). We observe two highly similar pairs of variables: variables 0 with 2, and variables 1 with 3, and these pairs exhibit high PCM coefficients and low DTW distances, as indicated at coordinates (2,0) and (3,1), where (x-axis, y-axis) correspond to variable indices. Furthermore, their dependency patterns with other variables are also analogous (see row 0 vs. row 2, row 1 vs. row 3) in both the PCM and DTW matrices. Additionally, the strong correlation between variables 4 and 5 (see coordinate (5,4) in PCM and DTW), is noteworthy and will be further discussed in the context of Figure 9.

mation, as seen in models including Informer (Zhou et al., 2021), Autoformer (Wu et al., 2021), iTransformer (Liu et al., 2024), VCformer (Yang et al., 2024) and others. However, *future timestamps are often unavailable in real-world scenarios, timestamp formats can vary across datasets, and issues such as missing or erroneous timestamps all cannot be effectively handled. Its actual effectiveness, moreover, is yet to be fully established.*

To investigate the actual efficacy of such temporal information, we conducted an ablation study on iTransformer where we removed the timestamp embedding and instead up-sampled the input signal using a single linear layer. As shown in Figure 2, this substitution leads to performance improvements on eight datasets (Full results and more analysis are provided in Appendix A.4).

To address the limitations of Transformer-based forecasters relying on timestamp information, we propose a novel and universal Frequency-domain Masking and Resampling (FMR) method, which performs learnable feature enhancement and periodicity capture directly on the frequency components of the signal. Specifically, a per-variable resampling is performed in the spectral space by applying a learnable mask and a subsequent linear interpolation. Through this process, the signal's periodicity is robustly preserved and enhanced (see Appendix A.5), thereby significantly diminishing the importance of timestamp information that traditionally serve to retain periodic or seasonal information.

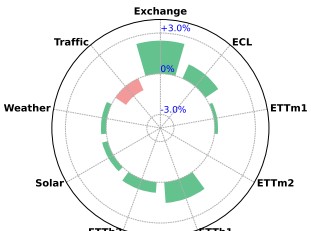

Figure 2: Impact of timestamps on iTransformer. The radar chart presents an improved (green) or decreased (red) percentage.

(2) **Inconsistency in Modeling Inter-variate Dependencies**. Transformer consists of two key stages: the multi-head self-attention (MHSA) layer and the subsequent feed-forward network (FFN). iTransformer introduced the "Variate Transformer" paradigm, which explicitly models IVD by encoding each variable as an individual token. This foundational work has inspired further improvements, such as Soatten (Wu, 2025). Nevertheless, we argue that a potential limitation exists here: an inconsistency in how temporal and inter-variate dependencies are modeled, that is, IVD are modeled exclusively within the shallow self-attention layers. The deeper FFNs, in contrast, completely disregard these dependencies, focusing solely on capturing the temporal dynamics within each individual variable (see Figure 3(a)). We acknowledge that numerous Transformer variants have been proposed to better model IVD, including approaches based on metric learning, such as DUET (Qiu et al., 2025), and methods employing graph transformers, like STGAGRTN (Wu et al., 2023a) and GL-STGTN (Li et al., 2024). However, a typical trait in these methods is that they integrate the learned variable dependencies into the self-attention mechanism, typically as an attention mask or a bias term. We categorize this parallel fusion strategy as the method depicted in Figure 3(b). We argue that **these approaches do not address the inconsistency in modeling temporal and inter-variate**

**dependencies, and such inconsistency poses challenges for model optimization** (see Figure 8), stemming from the degradation or even loss of deep-layer inter-variate dependencies.

The challenge then lies in how to implement IVD modeling within the Transformer deep layers. In this study, we resort to Graph Neural Network (GNN). Notably, the self-attention mechanism in a Transformer can be interpreted as a GNN operating on a fully-connected graph (Joshi, 2025). The primary distinction lies in the scope of the aggregation: GNNs aggregate information from the local neighborhood nodes, while in Transformer's self-attention, the aggregation is performed over the entire set of tokens in the sequence. For a more detailed theoretical analysis supporting these arguments and **elucidating our rationale for employing GNNs in deep layers to model IVD**, see Appendix A.7 and A.8. Therefore, we propose a Dynamic Graph Learning (DGL) framework that dynamically optimizes the graph structure based on global input and explicitly models IVD via a message-passing mechanism. Concurrently, it employs two linear layers to aggregate and extract deep temporal features. This dual-component design for feature extraction allows us to replace the FFNs in the Transformer with our DGL, as depicted in Figure 3(c), we consistently model both temporal and inter-variate dependencies, underscoring the importance of modeling IVD throughout the network, not only in shallow MHSA.

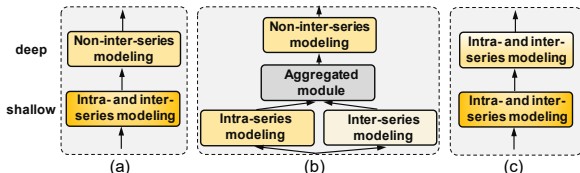

Figure 3: A Comparison of Paradigms for Modeling Temporal and Inter-variable Dependencies. (a) and (b) illustrate existing Transformer-based approaches for modeling IVD. (c) depicts the consistent dependency modeling paradigm proposed in this work.

However, since both the self-attention and the graph learning mechanisms model dependencies from the global inputs, they should, in theory, depict the same "latent true dependency correlations". This raises a critical question: **do the dependency structures modeled at these two different stages exhibit similarity or correlation?** To the best of our knowledge, this question remains unexplored in existing research. By analyzing the dependency matrices actually captured by self-attention and DGL, they indeed exhibited a high degree of similarity (their Kullback-Leibler (KL) divergence is 0.0260, as detailed in the Appendix A.2).

Despite these similarities, discrepancies exist because GNNs and self-attention layers aggregate information from distinct perspectives. **Acknowledging that both perspectives (i.e., local and global) offer unique advantages, we aim to find a balance between these two types of dependency modeling.** Therefore, we introduce Kullback-Leibler (KL) divergence to quantify the distance between these two dependency distributions. This divergence is then incorporated as a regularization term into the overall loss function (the theoretical guarantees based on Information Bottleneck (IB) principle are provided in Appendix A.9). After the introduction of the alignment constraint, the correlation matrices captured by self-attention and DGL are converged to be more similar (the KL divergence decreased from 0.0260 to 0.0249). And importantly, the graph structure retains specific correlations that are difficult for the standard self-attention mechanism to capture, such as the strong dependency at coordinate (5,4) (See Figure 9(b) in the Appendix A.2).

Synthesizing the foregoing analysis, we propose CGTFra, a compact framework that considers consistency in modeling IVD. Our primary contributions are as follows:

- We propose a novel, position-agnostic approach based on learnable frequency-domain masking and linear interpolation, which serves not only as an effective supplement but also as a potential replacement for existing timestamps encoding or up-sampling methods.

- Motivated by the need for consistent modeling of both intra- and inter-series dependencies across shallow and deep network layers, we propose a novel graph transformer framework named CGTFra. Furthermore, the proposed Dynamic Graph Learning in CGTFra can be integrated into existing variate transformers as a universal method for modeling IVD.

- We are the first to investigate the relationship between IVD modeled at shallow and deep network layers. To enforce consistency, we introduce an explicit constraint that aligns these two dependency structures, which is integrated as a regularizer into the main loss function.

- Our proposed CGTFra sets a new state-of-the-art in both long- and short-term time series forecasting on 13 datasets with superior computational efficiency (see Appendix A.18).

## 2 RELATED WORK

**Application of Timestamp Encoding in Time Series Forecasting.** Inspired by the effectiveness of positional encoding in NLP, numerous Transformer-based studies in MTSF have adopted this technique. The fusion of timestamp positional and data encodings is primarily achieved through two strategies: **direct summation**, as seen in models like Informer, TimesNet (Wu et al., 2023b), Autoformer (Wu et al., 2021), and Fedformer (Zhou et al., 2022), or **concatenation**, employed by iTransformer and VCformer. Notably, a distinct approach is presented in GLAFF (Wang et al., 2024). This work proposes the independent learning of timestamp information—encompassing both historical and future timestamps—and the data features. These two streams of information are then fused using an adaptive weighting mechanism, leading to superior forecasting performance. However, such approaches face significant practical challenges. *In many real-world application scenarios, future timestamps are unavailable. Furthermore, timestamp formats can be inconsistent across different datasets.* Methods like GLAFF are ill-equipped to handle these situations effectively.

**Modeling Inter-Variate Dependencies with Transformers.** Conventional temporal Transformers for MTSF typically encode information from different variables at the same timestamp into a single token. This approach, however, leads to a loss of IVD information, as seen in temporal Transformer-based studies (Chen et al., 2024; Luo & Wang, 2024; Nie et al., 2023). Crossformer (Zhang & Yan, 2023) employs a tailored two-stage attention layer to explicitly model both intra- and inter-series dependencies. iTransformer encodes each individual time series as a single token, offering greater universality in modeling IVD compared to Crossformer. TokenGT (Kim et al., 2022) treats nodes and edges as independent, learnable tokens, which are then fed into the Transformer alongside the input tokens. DUET captures IVD in the frequency domain using metric learning. The resulting dependency is then integrated into the self-attention mechanism as a mask for the attention scores.

**Modeling Inter-Variate Dependencies with Graph Transformers.** SageFormer (Zhang et al., 2024) first employs a GNN to capture IVD from the input MTS. The resulting global, graph-enhanced embeddings are then fused with the original series to serve as the input for a vanilla Transformer, which subsequently models temporal dependencies. STGAGRTN (Wu et al., 2023a) utilizes a gating mechanism to fuse the IVD learned separately by a GAT and a spatial Transformer. GL-STGTN (Li et al., 2024) learns the graph structure from both global and local perspectives, and then the learned IVD are then encoded into a spatial attention mechanism. For a more detailed discussion of the implementation specifics of these methods, please see Appendix A.10.

In summary, existing researches can be broadly categorized into two main strategies: (1) methods like DUET, STGAGRTN, and GL-STGTN, which integrate learned inter-variate dependencies into the self-attention mechanism as a mask or bias for attention scores; and (2) approaches such as Sage-Former and TokenGT, which embed graph-structural information directly into the input embeddings. *However, a common limitation of all these methods is their failure to consider the consistency and correlation of IVD modeling between the shallow and deep layers of the network.*

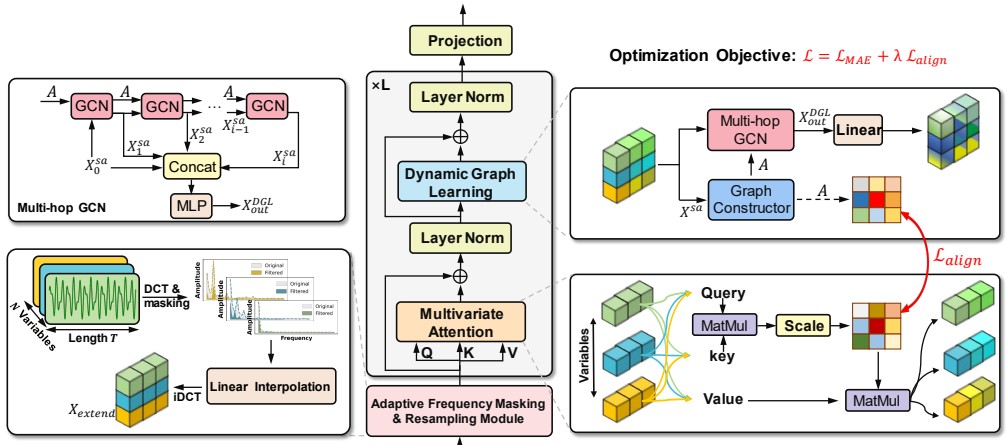

Figure 4: Illustration of proposed CGTFra. The output of the $i$-hop GCN is denoted by $X_i^{sa}$.

## 3 METHODOLOGY

For MTSF tasks, given historical input $X = [X_1^{1:T}, X_2^{1:T}, \ldots, X_N^{1:T}] \in \mathbb{R}^{T \times N}$, where $T$ is the input length and $N$ is the number of variates, and each $X_N^{1:T} \in \mathbb{R}^T$ is the $N$-th variate. We use CGTFra to forecast $Y = [X_1^{T+1:T+F}, X_2^{T+1:T+F}, ..., X_N^{T+1:T+F}] \in \mathbb{R}^{F \times N}$ during future $F$ time steps.

As illustrated in Figure 4, we propose CGTFra, a graph transformer framework designed for consistent IVD modeling. CGTFra inherits the Transformer's proficiency in capturing long-range dependencies while simultaneously demonstrating exceptional capabilities in modeling IVD. Technically, CGTFra is built upon three core design principles: (1) A universal, adaptive Frequency-domain Masking and Resampling (FMR) (Upsampling or downsampling). (2) A Dynamic Graph Learning (DGL) framework that can be integrated into existing transformers. (3) An alignment constraint that promotes consistency between the IVD modeled at the shallow and deep network layers.

### 3.1 ADAPTIVE FREQUENCY MASKING AND RESAMPLING

Compared to resampling directly in time domain with a linear layer, resampling in the frequency domain introduces a powerful inductive bias of a global receptive field. This paper utilizes DCT for frequency domain analysis (the motivation is provided in Appendix A.3). Furthermore, given that each variable possesses its own intrinsic dynamics, we learn an independent frequency mask for each variable. This allows the model to adaptively highlight critical frequencies and attenuating irrelevant or detrimental ones. Given an MTS $X = \{X_1, X_2, ..., X_N\} \in \mathbb{R}^{T \times N}$ where $X_n = [X_n(0), X_n(1)), ..., X_n(T-1)]^\top$ denotes the sequence values for the $n$-th variable (For simplicity, we explicitly denote the variable dimension only when computing the DCT and iDCT), this process is formulated as:

$$F_n(\mu) = c(\mu)\sqrt{\frac{2}{T}} \sum_{t=0}^{T-1} X_n(t) \cos[\frac{\pi\mu(2t+1)}{2T}], c(\mu) = \begin{cases} \sqrt{\frac{1}{2}}, & \mu = 0 \\ 1, & \mu = 1, 2, ..., T-1 \end{cases} \quad (1)$$

$$F^{mask} = F(\mu) \odot softplus(\mathcal{M}) \quad (2)$$

where $F(\mu), F^{mask} \in \mathbb{R}^{T \times N}$ represents the DCT coefficients and the masked frequency coefficients. $\mathcal{M} \in \mathbb{R}^{T \times N}$ denotes the variable-specific learnable mask. In Equation 1, $\mu \in \{0, 1, \ldots, T-1\}$ is the DCT index. Subsequently, a learnable linear layer is employed to perform linear interpolation on the masked frequency components, yielding the expanded frequency representation $F^{extend} \in \mathbb{R}^{D \times N}$, $D$ is the hyperparameter of extended size. Subsequently, the inverse Discrete Cosine Transform (iDCT) is applied to convert the frequency components $F^{extend}$ back into a temporal signal $X^{extend} \in \mathbb{R}^{D \times N}$. This process is formulated as:

$$F^{extend} = \text{Resampling}(F^{mask}) \quad (3)$$

$$X_n^{extend} = \sqrt{\frac{2}{D}} \sum_{t=0}^{D-1} c(\mu) F_n^{extend} \cos[\frac{\pi\mu(2t+1)}{2D}] \quad (4)$$

where Resampling($\cdot$) is implemented by the learnable linear interpolation. By performing masking and resampling within the frequency domain, the signal's periodicity is robustly preserved and even enhanced (see Appendix A.5). Therefore, the importance of timestamps is greatly diminished.

### 3.2 DYNAMIC GRAPH LEARNING

Unlike SageFormer and MSGNet (Cai et al., 2024), which rely solely on self-learned node embeddings to construct graph structure—a process prone to learning spurious correlations (Fan et al., 2023), we inject the input features (i.e., the output $X^{sa} \in \mathbb{R}^{N \times D}$ of the self-attention layer) into the node embedding generation process. This allows us to define the graph topology from a global perspective based on the input tokens, aligning the global modeling by self-attention. Specifically, we first use a linear transformation to derive an adaptive gating weight for each node from the static node embedding and global dynamic input. This weight is then multiplied with a linearly transformed representation of the node's own features, obtaining a dynamic node embedding that is continuously updated throughout the network.

$$\Theta^l = \text{ReLU}(\text{Tanh}(\text{Linear}(\text{Concat}(X^{sa,l}, \Theta^l)))) \odot \text{Linear}(X^{sa,l}) + \Theta^l \quad (5)$$

where $\Theta^l$ includes $\Theta_1^l, \Theta_2^l \in \mathbb{R}^{N \times nd}$, which are trainable parameters (with random initialization) of $l$-th layer, $nd$ is a hyperparameter, denoting the dimension of node. $\Theta_1^l$ and $\Theta_2^l$ employ the same

update strategy as in Equation 5, but without parameter sharing. $\odot$ is the Hadamard Product. Then, the adjacent matrix $A^l \in \mathbb{R}^{N \times N}$ of $l$-th layer can be represented as: $A^l = \text{Softmax}(\text{ReLU}(\Theta_1^l \cdot (\Theta_2^l)^T))$. Therefore, the graph structure at the $l$-th layer can be denoted as $\mathcal{G}^l = (A^l, X^{sa,l})$. To reconcile the discrepancy between the local neighborhood aggregation of GNNs and the global modeling of Transformers' self-attention, we employ a multi-hop GCN (Hamilton et al., 2017) to capture IVD at the deep feature level. The information from different hop neighborhoods is then combined using a linear layer by $X_{out}^{DGL} = \text{MLP}(\text{GCN}(X^{sa}, A))$. By aggregating information from its $i$-hop neighborhood, an $i$-hop GCN effectively enlarges each node's receptive field, enabling the capture of higher-order graph structures. To preserve the deep network's capacity for temporal feature extraction, our DGL strategically mirrors the two-layer MLP design of a conventional FFN. Specifically, the first MLP layer is adapted to aggregate multi-hop neighborhood information, while the second MLP layer extracts temporal features from the deep representations that have already been enriched with IVD.

## 3.3 CONSISTENCY ALIGNMENT LOSS FUNCTION

The self-attention mechanism in a Transformer is essentially a GNN operating on a fully-connected graph, which implies that they can describe the same underlying correlation structure. Based on this insight, our work is the first to propose an explicit constraint alignment between the dependencies captured by the deep-layer GNN and the shallow-layer self-attention. This alignment prevents over-reliance on a single mode of dependency modeling (Figure 9 analyzes the respective disadvantages). Following iTransformer, each variable $X^{extend}[n,:] \in \mathbb{R}^{1 \times D}$, $n = 1, 2, \ldots, N$, is regarded as an independent token and the self-attention layer then is applied to model multivariate correlations:

$$head_i = \text{Softmax}(\frac{(X^{extend}W_i^Q) \cdot (X^{extend}W_i^K)^T}{\sqrt{d_K}}), \text{MCM} = \text{Concat}(head_1, ..., head_h) \quad (6)$$

where $W_i^Q, W_i^K \in \mathbb{R}^{D \times \frac{D}{h}}$ are the projection metrices of $i$-th head, and $h$ is the number of attention heads with a default value 8. We use $\text{MCM} \in \mathbb{R}^{h \times N \times N}$ to represent the multivariate correlation map (a.k.a., attention score). Therefore, the total alignment loss of $l$ layer CGTFra for consistent IVD modeling can be formalized as follows by Kullback-Leibler (KL) Divergence:

$$\mathcal{L}_{align} = \sum_{l=1}^{L} \text{KL}(P_l \parallel Q_l) = \sum_{l=1}^{L} \sum_{k=1}^{N^2} e^{p_{l,k}}(p_{l,k} - q_{l,k}) \quad (7)$$

where $p_l = \log P_l = \text{log\_softmax}(\text{Vec}(\text{Avg}(\text{MCM}^l)))$, and $q_l = \log Q_l = \text{log\_softmax}(\text{Vec}(A^l))$. In our implementation, we directly compute the log-probabilities to avoid $\log(0)$ errors. $\text{Avg}(\cdot)$ denotes averaging the attention score along $h$ attention head, and $\text{Vec}(\cdot)$ denotes vectorizing the correlation matrix into a one-dimensional vector. Therefore, the total loss function for optimizing CGTFra is formulated as:

$$\mathcal{L} = \mathcal{L}_{MAE} + \lambda \mathcal{L}_{align} , \quad (8)$$

where $\mathcal{L}_{MAE} = \frac{1}{F} \sum_{i=1}^{F} |y_i - \hat{y}_i|$ represents the Mean Absolute Error (MAE) for evaluating prediction accuracy with the forecasting length $F$. $y_i$ and $\hat{y}_i$ are the ground truth and predicted value at time $i$, and $\lambda$ is a hyperparameter, controlling the contribution of alignment loss. Here, for simplicity, we omit the batch dimension and illustrate the loss calculation for a single variable.

## 4 EXPERIMENTS

### 4.1 DATASETS

We select 13 real-world datasets to comprehensively verify our CGTFra following iTransformer, including ETT (4 subsets), Weather, Exchange, Electricity (ECL), Solar-Energy, Traffic, PEMS03, PEMS04, PEMS07 and PEMS08. All datasets are preprocessed following iTransformer. And more details of these datasets are provided in Appendix A.11.

### 4.2 BASELINES AND EXPERIMENTAL SETTINGS

We choose 13 sota forecasting methods as our benchmarks, including (1) Transformer-based models: DUET (Qiu et al., 2025), Soatten (Wu, 2025), Vcformer (Yang et al., 2024), iTransformer (Liu

et al., 2024), Crossformer (Zhang & Yan, 2023), and PatchTST (Nie et al., 2023); (2) GNN-based approach, MSGNet (Cai et al., 2024); (3) MLP/Linear-based models: FilterNet (Yi et al., 2024), RLinear (Li et al., 2023), TiDE (Das et al., 2023), and DLinear (Zeng et al., 2023); (4) CNN-based one: TimesNet (Wu et al., 2023b); (5) Mamba-based method, TimePro (Ma et al., 2025). Following established practice, we evaluate our CGTFra using Mean Absolute Error (MAE) and Mean Squared Error (MSE). The input length for all datasets is set as 96 in main comparison scenario. All experiments are implemented in PyTorch 2.0.1 with Python 3.8 on two NVIDIA GeForce RTX 3090 GPUs. Additional implementation details can be found in the Appendix A.12.

### 4.3 MAIN RESULTS

The long-term and short-term forecasting comparison results are presented in Table 1 and Table 8. Overall, CGTFra demonstrates superior performance in both forecasting tasks. This superiority is particularly pronounced on datasets with a large number of variables, such as ECL, and Traffic, where modeling IVD poses a significant challenge for existing methods, such as DUET and VCformer. Specifically, compared to DUET, CGTFra reduces MSE (MAE) by 5.1% (4.5%) on the Traffic dataset. Additionally, in most scenarios, CGTFra exhibits enhanced performance when applied to datasets with inherent low predictability (see Table 6), including ETT and Solar, demonstrating the effectiveness of CGTFra to modeling long-term intra- and inter-variate dependencies.

Table 1: Long-term forecasting results with **fixed input Length** $T$=96 and forecasting horizons $F \in \{96, 192, 336, 720\}$. The results are averaged from four forecasting horizons. Full results, short-term forecasting results, and the additional comparison scenario when $T$=336 are all provided in Appendix A.13. **Bold**: best results, underline: second best one.

| Models | CGTFra (ours) | | DUET (KDD'25) | | TimePro (ICML'25) | | Soatten (AAAI'25) | | VCformer (IJCAI'24) | | FilterNet (NeurIPS'24) | | iTransformer (ICLR'24) | | MSGNet (AAAI'24) | | PatchTST (ICLR'23) | |
|---|---|---|---|---|---|---|---|---|---|---|---|---|---|---|---|---|---|---|
| Metrics | MSE | MAE | MSE | MAE | MSE | MAE | MSE | MAE | MSE | MAE | MSE | MAE | MSE | MAE | MSE | MAE | MSE | MAE |
| ETTm1 | 0.388 | **0.386** | 0.390 | 0.393 | 0.391 | 0.400 | 0.394 | 0.402 | 0.387 | 0.397 | **0.384** | 0.398 | 0.407 | 0.410 | 0.398 | 0.411 | 0.387 | 0.400 |
| ETTm2 | 0.277 | **0.316** | 0.280 | 0.324 | 0.281 | 0.326 | 0.287 | 0.331 | 0.285 | 0.330 | **0.276** | 0.322 | 0.288 | 0.332 | 0.288 | 0.330 | 0.281 | 0.326 |
| ETTh1 | **0.436** | **0.428** | 0.443 | 0.436 | 0.438 | 0.438 | 0.447 | 0.440 | 0.439 | 0.437 | 0.440 | 0.432 | 0.454 | 0.447 | 0.452 | 0.452 | 0.469 | 0.454 |
| ETTh2 | **0.369** | **0.394** | 0.372 | 0.397 | 0.377 | 0.403 | 0.379 | 0.405 | 0.377 | 0.403 | 0.378 | 0.404 | 0.383 | 0.407 | 0.396 | 0.417 | 0.387 | 0.407 |
| Exchange | **0.312** | **0.382** | 0.318 | 0.384 | 0.352 | 0.399 | 0.359 | 0.404 | 0.355 | 0.402 | 0.356 | 0.395 | 0.360 | 0.403 | 0.399 | 0.430 | 0.367 | 0.404 |
| Weather | **0.238** | **0.260** | 0.251 | 0.273 | 0.251 | 0.276 | 0.245 | 0.273 | 0.258 | 0.282 | 0.245 | 0.272 | 0.258 | 0.278 | 0.249 | 0.278 | 0.259 | 0.281 |
| ECL | **0.165** | **0.253** | 0.172 | 0.258 | 0.169 | 0.262 | 0.166 | 0.259 | 0.180 | 0.267 | 0.173 | 0.268 | 0.178 | 0.270 | 0.194 | 0.300 | 0.205 | 0.290 |
| Solar | **0.224** | **0.228** | 0.237 | 0.233 | 0.232 | 0.266 | 0.229 | 0.261 | - | - | - | - | 0.233 | 0.262 | - | - | 0.270 | 0.307 |
| Traffic | **0.427** | **0.257** | 0.451 | 0.269 | - | - | 0.437 | 0.286 | 0.483 | 0.325 | 0.463 | 0.310 | 0.428 | 0.282 | - | - | 0.555 | 0.362 |

### 4.4 FRAMEWORK GENERALITY

To evaluate the effectiveness and scalability of the three core designs in CGTFra: Frequency Masking and Resampling (FMR), Dynamic Graph Learning (DGL) framework, and Consistency Alignment Loss (CAL), we conducted a series of integration and replacement experiments within existing SOTA models, including DUET, iTransformer, VCformer, FilterNet and CASA (Lee et al., 2025). For fair comparison, we use their originally published hyperparameter settings. "+ FMR", "+ DGL": substituting their input up-sampling methods with our FMR and their FFNs with our DGL. "+ CAL": on top of the DGL substitution, we introduce the CAL. The averaged comparison results are presented in Table 2. FMR and DGL demonstrated consistent performance improvements in almost all datasets, and **the substantial performance gains brought by DGL underscore the importance of deep IVD modeling, which has been entirely overlooked in their studies**. In addition, **we observe that by introducing CAL, compared to their original performance, iTransformer (VCformer) reduces the MSE by 4.9% (7.4%) and 7.4% (10.2%) on Weather and ECL datasets, respectively**, approaching or even surpassing the latest sota methods DUET and TimePro. Furthermore, to specifically verify the efficacy of the DGL within the broader family of variate transformers, we integrated it into four architectures mentioned in the iTransformer: iFlashformer (Dao et al., 2022), iFlowformer (Wu et al., 2022), iInformer (Zhou et al., 2021), and iReformer (Kitaev et al., 2020). In Table 12, some variate Transformers integrated into DGL show a slight performance

Table 2: Verification of Framework Generality. Results are averaged from four forecasting horizons. Full results, **additional valuation metrics** and further analysis are in Appendix A.14. For a fair comparison, the results in Table 1 are taken from their officially released reports, whereas the results below are reproduced under our experimental environment, and consequently, some discrepancies exist. "–" denotes that the original method was not evaluated on certain datasets, or that we encountered out-of-memory issues. "iTrans" and "Filter" denote iTransformer and FilterNet, respectively.

| Models | | ETTm1 | | ETTm2 | | ETTh1 | | ETTh2 | | Exchange | | Weather | | ECL | | Solar | | Traffic | |
|---|---|---|---|---|---|---|---|---|---|---|---|---|---|---|---|---|---|---|---|
| Metrics | | MSE | MAE | MSE | MAE | MSE | MAE | MSE | MAE | MSE | MAE | MSE | MAE | MSE | MAE | MSE | MAE | MSE | MAE |
| DUET | original | 0.391 | 0.394 | 0.279 | 0.322 | 0.449 | 0.440 | 0.372 | 0.398 | 0.309 | 0.380 | 0.247 | 0.270 | 0.172 | 0.258 | **0.241** | **0.246** | 0.451 | 0.269 |
| | + DGL | **0.389** | **0.391** | 0.277 | **0.320** | 0.444 | 0.436 | **0.368** | **0.395** | **0.296** | **0.373** | 0.247 | 0.272 | 0.166 | 0.255 | 0.243 | 0.258 | **0.448** | **0.268** |
| | + CAL | 0.391 | 0.393 | 0.282 | 0.324 | **0.438** | **0.432** | 0.373 | 0.397 | 0.305 | 0.376 | **0.237** | **0.263** | **0.164** | **0.253** | 0.242 | 0.253 | 0.452 | 0.269 |
| iTrans | original | 0.408 | 0.412 | 0.293 | 0.337 | 0.457 | 0.449 | 0.384 | 0.407 | 0.369 | 0.409 | 0.262 | 0.283 | 0.176 | 0.268 | 0.235 | 0.261 | **0.422** | 0.282 |
| | + FMR | 0.403 | 0.406 | **0.291** | **0.333** | 0.448 | 0.440 | **0.381** | **0.406** | **0.358** | **0.404** | 0.259 | 0.282 | 0.175 | 0.266 | **0.229** | **0.260** | 0.423 | **0.281** |
| | + DGL | **0.400** | 0.406 | 0.293 | 0.335 | 0.449 | 0.442 | 0.390 | 0.412 | 0.368 | 0.409 | 0.252 | 0.278 | 0.169 | 0.263 | 0.234 | 0.263 | 0.434 | 0.288 |
| | + CAL | 0.402 | **0.405** | 0.292 | 0.335 | **0.444** | **0.440** | 0.386 | 0.408 | 0.365 | 0.408 | **0.249** | **0.276** | **0.163** | **0.257** | 0.233 | 0.262 | 0.440 | 0.286 |
| VCformer | original | 0.404 | 0.406 | 0.292 | 0.334 | 0.488 | 0.460 | **0.384** | **0.405** | **0.358** | **0.403** | 0.269 | 0.286 | 0.186 | 0.278 | - | - | - | - |
| | + FMR | **0.398** | 0.402 | 0.291 | 0.333 | 0.457 | **0.441** | 0.385 | 0.406 | 0.367 | 0.409 | 0.265 | 0.285 | 0.182 | 0.275 | - | - | - | - |
| | + DGL | **0.398** | **0.401** | 0.289 | 0.333 | 0.456 | 0.447 | 0.389 | 0.410 | 0.363 | 0.404 | **0.249** | **0.275** | 0.174 | 0.266 | - | - | - | - |
| | + CAL | 0.401 | 0.405 | **0.287** | **0.331** | **0.451** | 0.444 | 0.388 | 0.410 | 0.361 | 0.406 | **0.249** | **0.275** | **0.167** | **0.261** | - | - | - | - |
| CASA | original | **0.391** | **0.400** | 0.279 | 0.323 | **0.442** | **0.440** | 0.383 | 0.406 | - | - | 0.249 | 0.276 | 0.172 | 0.265 | 0.226 | 0.261 | **0.427** | **0.278** |
| | + FMR | 0.392 | 0.401 | **0.277** | **0.322** | **0.442** | **0.440** | 0.378 | 0.404 | - | - | **0.245** | **0.273** | **0.169** | **0.263** | 0.223 | 0.259 | 0.444 | 0.279 |
| Filter | original | 0.384 | **0.398** | 0.277 | **0.322** | 0.451 | **0.437** | **0.379** | **0.405** | - | - | 0.253 | 0.280 | 0.179 | 0.272 | - | - | 0.460 | 0.304 |
| | + FMR | **0.383** | **0.398** | **0.276** | **0.322** | **0.450** | **0.437** | **0.379** | **0.405** | - | - | **0.248** | **0.276** | **0.177** | **0.271** | - | - | **0.455** | **0.300** |

decline on the Solar, despite a minimal difference in their MAE. Therefore, we provide additional evaluation metrics in Section A.14 to validate the effectiveness of DGL and CAL.

## 4.5 ABLATION STUDY

The comparison between aba1, and CGTFra vs aba3, demonstrates that the FMR, by effectively purifying and enhancing input features, substantially enhances the robustness of deep-layer IVD modeling, particularly on the ECL and Traffic. Furthermore, by introducing DGL and CAL (see aba1 vs aba2 and CGTFra vs aba2), consistent performance improvement indicates that constraining the consistency between shallow- and deep-layer IVD modeling enables the model to achieve a more robust balance of dependencies. To further validate the necessity of modeling inter-variable dependencies at deeper layers, we present experiments on variants of CGTFra in Appendix A.15.

Table 3: Ablation studies on five diverse datasets. The results are averaged from four forecasting horizons. Full results are provided in Table 15 of Appendix.

| Part | FMR | DGL | CAL | ETTm1 | | ETTh1 | | Weather | | ECL | | Traffic | |
|---|---|---|---|---|---|---|---|---|---|---|---|---|---|
| | | | | MSE | MAE | MSE | MAE | MSE | MAE | MSE | MAE | MSE | MAE |
| CGTFra | ✓ | ✓ | ✓ | **0.388** | **0.386** | **0.436** | **0.428** | **0.238** | **0.260** | **0.165** | **0.253** | **0.427** | **0.257** |
| aba1 | ✓ | ✗ | ✗ | 0.397 | 0.393 | 0.442 | 0.431 | 0.245 | 0.266 | 0.170 | 0.256 | 0.431 | 0.261 |
| aba2 | ✓ | ✓ | ✗ | 0.389 | 0.390 | 0.437 | 0.428 | 0.242 | 0.266 | 0.168 | 0.256 | 0.430 | 0.259 |
| aba3 | ✗ | ✓ | ✓ | 0.392 | 0.390 | 0.437 | 0.429 | 0.243 | 0.266 | 0.173 | 0.260 | 0.444 | 0.262 |

## 4.6 ANALYSIS OF INTER-SERIES DEPENDENCY MODELING

To further analyze CGTFra's effectiveness in modeling inter-variate dependencies and extracting complex temporal dynamics, we select a sample from the Weather dataset's test set (all variable dynamics are provided in Figure 10 (b)). Within this sample (with 21 variables), four highly correlated variables (variables 3, 7, 8, and 13) are chosen for visualization and analysis. As depicted in Figure 5, we visualize the prediction curves of these four variables predicted by CGTFra, alongside the PCC and DTW among the ground truth, and CGTFra predicted variables. We observed that although the predicted sequences do not greatly match with the true sequences, the overall trends are correctly captured. Furthermore, the close proximity of the predicted PCC and DTW values to their true counterparts indicates the model's commendable ability to capture inter-variate dependencies.

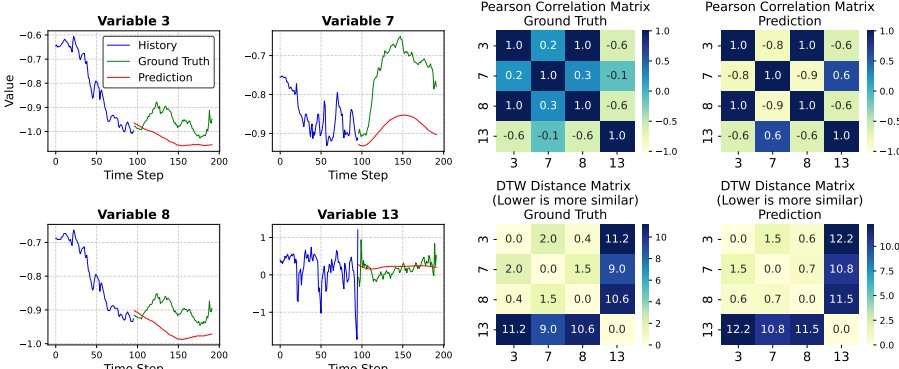

Figure 5: Prediction curves for CGTFra (input 96-predict 96) and the DTW and PCC comparison between ground truth and predicted sequences among variables [3, 7, 8, 13]. According to DTW and PCC, variable 3 exhibits a strong association with variable 8, while variable 7 also shows substantial correlations with both variables 3 and 8, as indicated by small DTW distances and high PCC.

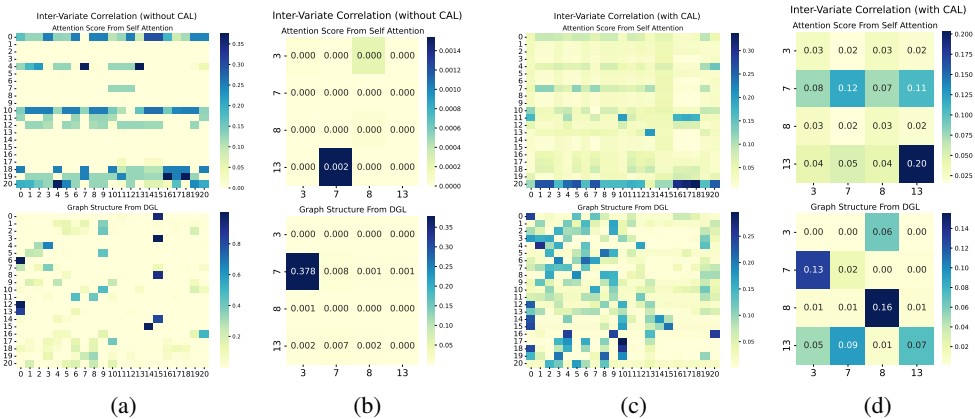

Figure 6: Inter-variate correlation learned by CGTFra on the test sample. (a) and (b): Dependencies from **CGTFra without CAL**; (c) and (d): Dependencies from the **complete CGTFra with CAL**. (a) and (c): Correlation matrices from the shallow self-attention layer and deep DGL; (b) and (d): Zoomed-in visualization of dependencies for variables 3, 7, 8, and 13.

Moving to Figure 6, we present two inter-variate correlation matrices learned by CGTFra from the selected test sample: one from the self-attention layer and the other from the DGL. Observing Figure 6(b), we are surprised to find that, **without the CAL constraint, neither the self-attention layer nor the DGL success to capture critical dependencies. This phenomenon is not attributed to a performance degradation caused by introducing DGL, but rather likely represents an inherent modeling challenge for the network** (CGTFra's performance without CAL in the Weather test set is MSE: 0.159 and MAE: 0.195, both outperforming existing methods, as shown in Table 15 and 7). Nevertheless, DGL still successfully captured the correlation between variables 3 and 7 (see coordinates (3, 7)), **which is consistent with our analysis in Figure 9 on ETTh1, where DGL is shown to capture indirect dependencies (between variable 4 and 5). This finding indicates that, compared with the global self-attention mechanism, GNNs possess an advantage in capturing indirect (or potential) dependencies by aggregating information from adjacent nodes**—for example, in Weather dataset, the relationship between variable 3 and variable 8 is apparent (direct), that between variable 3 (or 8) and variable 7 constitutes an indirect dependency (they also have smaller DTW distances and higher PCC).

Upon the introduction of CAL, both the self-attention layer and DGL effectively model prominent dependency correlations, as illustrated in Figures 6(c) and 6(d). Let us first examine two strongly correlated variables: variable 3 and 8 (see (8, 3)). The self-attention layer capture a weight of 0.03,

whereas DGL captures a weight of 0.06. Subsequently, we observe variable 3 and 7 (see (3, 7)), where the self-attention layer learns a weight of 0.08, while DGL captures a weight of 0.13. Furthermore, for variable 8 and 7 (see (8, 7)), they show 0.07 and 0.0, respectively. **These observations suggest that the self-attention mechanism (which captures global inter-variate correlations) and DGL (which leverages multi-hop GCNs for local dependency capture), possess distinct advantages. Crucially, the introduction of CAL promotes both mechanisms to achieve a more balanced and robust representation of dependency correlations.**

Figure 7 visualizes the t-SNE (Maaten & Hinton, 2008) embeddings learned from 1,500 test samples of the Weather dataset. Consistent with prior analysis, the embeddings for variables 3 and 8 learned by all three models (CGTFra, iTransformer, and DUET) are observed to be nearly overlapping (owing to their strong dependency). Building upon this, CGTFra demonstrates a shorter intra-variable distance, indicating that its representations for the same variable across different samples are more compact. Furthermore, in the embedding space of CGTFra, variable 13 is positioned more distantly from the others, and its sample representations are more tightly clustered. These observations suggest that CGTFra possesses a superior representation capability for learning individual variable features while more accurately capturing their inter-dependencies.

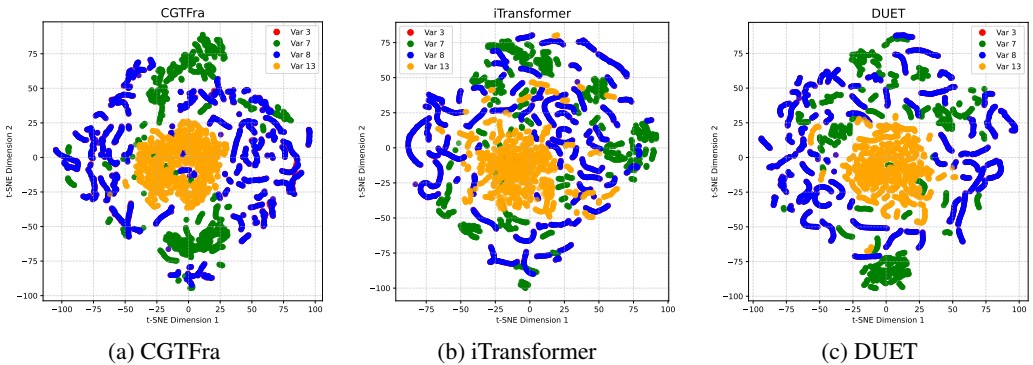

(a) CGTFra            (b) iTransformer            (c) DUET

Figure 7: T-SNE visualization for variable 3, 7, 8, and 13 on the Weather test set.

We plot the training and validation loss curves of iTransformer after incorporating DGL and CAL. **The trajectories indicate that deep modeling of IVD accelerates paremeter adjustment towords lower loss**. Moreover, we observe that introducing CAL yields a similarly stable loss trajectory as integrating DGL (this diminishing gain is expected, as most of the performance boost has already been achieved by DGL), **suggesting that achieve consistency alignment between deep- and shallow-level IVD provides additional effectiveness and robustness.**

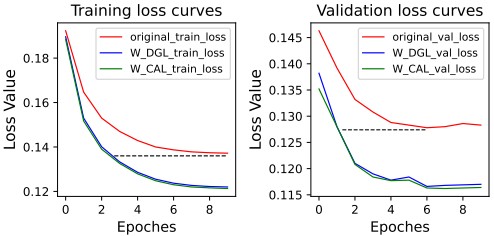

Figure 8: Visualization of training and validation loss curves for iTransformer (ECL: Input 96-Predict 96).

## 5 CONCLUSION

By conducting a theoretical investigation into the distinctions and connections between how variate Transformers and GNNs model IVD, this paper proposes CGTFra. This framework addresses the limitation of existing variate Transformers that neglect deep-layer IVD modeling. Furthermore, we introduce, for the first time, a consistency constraint applied to IVD learned by both self-attention and deep graph learning frameworks. This constraint serves as a regularization term in the total loss function, enabling the model to capture more consistent and robust IVD. This novel learning paradigm has been validated across multiple existing variate Transformers. We believe that exploring further mutual guidance principles between graph structures and Transformer-based inter-variate dependency modeling represents a promising future research direction. Additional limitations about CGTFra are provided in Appendix A.19.

## 6 REPRODUCIBILITY STATEMENT

Although an anonymous GitHub link is provided in the main text, we additionally upload the source code in the supplementary material. The code includes the proposed CGTFra as well as various baselines used to validate FMR, DGL, and CAL, including DUET, iTransformer, VCformer, CASA, and other variant Transformers, ensuring that all comparative results in this paper are reproducible.

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

# A APPENDIX

## A.1 LARGE LANGUAGE MODELS (LLMS) USAGE DISCLOSURE

The research methods, datasets, and open-source code in our study were developed without Large Language Models' assistance (e.g., ChatGPT). During manuscript preparation, GPT-based tools were solely employed to polish of selected words or sentences.

## A.2 DETAILED ANALYSIS OF IVD SIMILARITY BY SELF-ATTENTION AND DGL

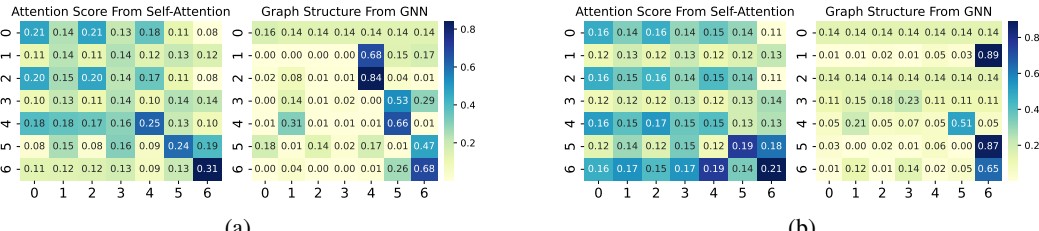

(a)                                                    (b)

Figure 9: Comparative analysis of dependency matrices from self-attention and GNN (derived from the 7 variables in Figure 1). (a) before alignment; (b) after applying the alignment constraint. Consistent with the PCM and DTW matrices in Figure 1, the self-attention mechanism successfully captures dependencies between highly similar variables, such as the pairs (2, 0) and (3, 1), and show similar dependency correlations with other variables (see rows (0 vs. 2) and (1 vs. 3) in Attention Score Map). **However, self-attention fails to capture less direct correlations, such as the one between variables 4 and 5 (See Figure 1, their PCC and DTW are 0.6, 5.3, respectively), which is successfully identified by the GNN (see coordinate (5,4)).** This result effectively demonstrates the efficacy of using DGL to model IVD in the deeper layers of our network. However, we also observed that the dependencies modeled by DGL can be exaggerated in some cases (e.g., at coordinate (4, 2)). To address this, we further introduced CAL based on information bottleneck principle (see Appendix A.9) to impose constraints on the IVD modeling. As shown by the graph structure in Figure (b), this inconsistency is significantly mitigated: compared to Figure (a), the KL divergence between the attention score and graph structure reduces form 0.0260 to 0.0249.

## A.3 MOTIVATION OF FREQUENCY MASKING AND RESAMPLING

Benefiting from the global receptive field of the frequency domain space, analyzing time series in the frequency space has become a prevailing trend, as seen in methods such as Fedformer (Zhou et al., 2022), TSLANet (Eldele et al., 2024), FilterNet (Yi et al., 2024), and DUET (Qiu et al., 2025). However, these approaches rely on the Discrete Fourier Transform (DFT) for frequency-domain analysis. Since DFT involves both real and imaginary components, it is computationally more complex than the Discrete Cosine Transform (DCT). Moreover, methods such as TSLANet and FilterNet primarily perform filtering on frequency components—similar to the masking mechanism proposed in this work—before transforming the filtered components back into the time domain for subsequent abstract feature learning. **This procedure introduces a potential risk: if critical frequency information is inadvertently filtered out, the subsequent feature extractor may struggle to capture informative representations. Consequently, such methods require both carefully designed frequency-domain filters and well-structured downstream feature extractors to achieve competitive performance.** Therefore, this paper proposes leveraging DCT to directly conduct frequency-domain analysis in the real-valued space and applying linear interpolation to the masked frequency components, thereby mitigating the risk of discarding essential information. Furthermore, we provide a theoretical discussion on the relationship between DFT and DCT as well as their computational complexity.

Like Section 3, let $\{f(l)\}$, $l = 0, 1, \ldots, L - 1$ be a input sequence. And let an extended sequence $\{e_l\}$ be symmetric about the $(2L - 1)/2$ point, that is, $e_l$ can be constructed by:

$$e_l = \begin{cases} f(l), & l = 0, 1, ..., L - 1 \\ f(2L - l - 1), & l = L, L + 1, ..., 2L - 1 \end{cases} \tag{9}$$

Here, suppose $L$=4, then the $\{f(l)\}$ and $\{e_l\}$ are:

$$\{f(l)\} = \{f(0), f(1), f(2), f(3)\}$$

$$\{e_l\} = \{f(0), f(1), f(2), f(3), f(3), f(2), f(1), f(0)\}$$

Let $W_{2L}$ denote $\exp(-j2\pi/2L)$, therefore the Discrete Fourier Transform (DFT) of $e_l$ can be given by:

$$E_\mu = \sum_{l=0}^{2L-1} e_l W_{2L}^{l\mu} \tag{10}$$

it can be easily reduced to

$$
\begin{aligned}
E_\mu &= \sum_{l=0}^{L-1} f(l)W_{2L}^{l\mu} + \sum_{l=L}^{2L-1} f(2L-l-1)W_{2L}^{l\mu} \\
&= \sum_{l=0}^{L-1} f(l)W_{2L}^{l\mu} + \sum_{l=0}^{L-1} f(l)W_{2L}^{(2L-l-1)\mu} \\
&= \sum_{l=0}^{L-1} f(l)[W_{2L}^{l\mu} + W_{2L}^{-(l+1)\mu}], \ \mu = 0, 1, ..., 2L-1.
\end{aligned}
\tag{11}
$$

If we use a factor of $\frac{1}{2}W_{2L}^{\mu/2}$ to multiply both sides of Equation 11, resulting in

$$\frac{1}{2}W_{2L}^{\mu/2}E_\mu = \sum_{l=0}^{L-1} f(l)\cos[\frac{\pi\mu(2l+1)}{2L}] \tag{12}$$

We can see that Equation 12 can be approximately Equation 1 of the $L$-point sequence $f(t)$, differing only by the scaling factors. In Equation 10, $E_\mu$ is the $2L$-point DFT of $\{e_l\}$ and Equation 12 indicates that for $\mu = 0, 1, ..., L-1$, after properly scaled, the transformed sequence $\{E_\mu\}$ can become the Type II DCT of $\{f(l)\}$.

When $\{f(l)\}$ is real and $e_l$ is symmetric, $\{E_\mu\}$ can be computed via two $N$-point FFTs instead of via a single $2N$-point FFT. Given that the computational complexity of an $N$-point FFT algorithm scales as $O(Nlog_2N)$ complex operations, this optimization reduces the $Nlog_2N$ FFT operation count by $2N$ complex operations.

### A.4 ACTUAL EFFICACY OF TIMESTAMPS INFORMATION

In the introduction, to investigate the actual contribution of timestamp information to iTransformer, we replace its original timestamp-embedded input upsampling module with a single linear layer without timestamp embedding. The performance comparison in Table 4 shows that timestamp information improves prediction performance only on the Traffic dataset, while leading to degradation on all other datasets, suggesting that its effectiveness deserves reconsideration. To explore this, we visualize partial time segments of the top five variables from the 862 variables in the Traffic dataset (see Figure 10 (a)). The results reveal fixed fluctuation patterns in traffic flow at nearly the same periods each day, and importantly, other variables exhibit highly similar variations. This observation may explain why timestamp information benefits iTransformer on Traffic. However, such characteristics are rare in real-world systems like weather or stock volatility, where variables tend to have more complex dependencies (see Figure 10 (b)).

The frequency-domain representations of the signals inherently provides a global perspective, and the periodic and seasonal characteristics of the signals are effectively represented in its frequency domain components (Zhou et al., 2022). Based on this insight, we propose a frequency-domain masking and resampling method (FMR) that preserves and enhances signal periodicity, thereby mitigating the over-reliance of existing methods on timestamp information for providing additional periodic insights. As shown in Table 4 (also in Table 2 or Table 11), FMR consistently improves performance across almost all datasets, further diminishing the importance of timestamp information.

Table 4: Verification of timestamps with four prediction length $F \in \{96, 192, 336, 720\}$ and fixed input $T$=96. All results were reproduced using their released code and identical hyperparameters. "iTrans" is iTransformer, and "R Linear" represents that we replace the input upsampling method within iTransformer with a sigle linear layer without timestamp embedding. For the "+FMR" scenario, bold results indicate the best performance within all results.

| Models | | ETTm1 | | ETTm2 | | ETTh1 | | ETTh2 | | Exchange | | Weather | | ECL | | Solar | | Traffic | |
|---|---|---|---|---|---|---|---|---|---|---|---|---|---|---|---|---|---|---|---|
| Metrics | | MSE | MAE | MSE | MAE | MSE | MAE | MSE | MAE | MSE | MAE | MSE | MAE | MSE | MAE | MSE | MAE | MSE | MAE |
| iTrans original | 96 | **0.342** | **0.377** | 0.186 | 0.272 | 0.387 | 0.405 | **0.301** | **0.350** | 0.086 | 0.206 | **0.181** | **0.221** | 0.148 | 0.239 | 0.201 | 0.234 | **0.392** | **0.268** |
| | 192 | **0.383** | 0.396 | 0.254 | 0.314 | 0.441 | 0.436 | 0.381 | 0.399 | 0.181 | 0.303 | 0.226 | 0.259 | 0.167 | 0.258 | 0.239 | 0.263 | **0.413** | 0.277 |
| | 336 | 0.418 | 0.418 | **0.317** | **0.353** | 0.491 | 0.462 | 0.423 | 0.432 | 0.338 | 0.422 | 0.283 | 0.300 | 0.181 | 0.275 | 0.248 | **0.272** | **0.425** | 0.283 |
| | 720 | 0.487 | 0.456 | 0.416 | 0.408 | 0.509 | 0.494 | 0.430 | 0.446 | 0.869 | 0.704 | 0.359 | 0.351 | **0.209** | **0.299** | 0.250 | 0.275 | 0.459 | **0.300** |
| iTrans R Linear | 96 | 0.347 | 0.377 | **0.184** | **0.267** | 0.383 | **0.401** | 0.303 | 0.352 | **0.085** | **0.205** | 0.183 | 0.223 | **0.147** | **0.239** | 0.201 | **0.233** | 0.396 | 0.270 |
| | 192 | 0.384 | **0.393** | **0.253** | **0.312** | **0.434** | **0.430** | **0.378** | **0.397** | **0.178** | **0.301** | 0.226 | 0.259 | **0.162** | **0.253** | 0.239 | 0.263 | 0.416 | 0.277 |
| | 336 | **0.416** | **0.414** | 0.319 | 0.354 | **0.487** | **0.457** | **0.417** | **0.429** | **0.336** | **0.420** | 0.281 | **0.299** | 0.175 | 0.267 | 0.248 | 0.273 | 0.431 | 0.285 |
| | 720 | **0.483** | **0.451** | **0.414** | **0.406** | 0.496 | 0.483 | **0.424** | **0.444** | 0.842 | **0.692** | **0.356** | **0.347** | 0.211 | 0.301 | **0.249** | 0.275 | 0.465 | 0.302 |
| iTrans + FMR | 96 | **0.340** | **0.373** | **0.183** | **0.265** | **0.382** | **0.398** | **0.299** | **0.350** | **0.084** | **0.204** | **0.180** | 0.222 | **0.141** | **0.235** | **0.199** | 0.237 | 0.393 | **0.268** |
| | 192 | **0.377** | **0.389** | **0.249** | **0.309** | **0.434** | **0.429** | **0.379** | **0.399** | **0.176** | **0.299** | **0.222** | **0.258** | 0.157 | **0.250** | **0.233** | **0.259** | **0.413** | **0.276** |
| | 336 | **0.412** | **0.411** | **0.314** | **0.350** | **0.483** | **0.454** | 0.419 | **0.430** | 0.339 | 0.423 | **0.279** | 0.300 | **0.171** | **0.264** | **0.242** | **0.269** | 0.428 | **0.282** |
| | 720 | **0.481** | **0.450** | 0.418 | 0.409 | **0.492** | **0.480** | 0.426 | 0.445 | **0.834** | **0.690** | **0.356** | 0.350 | 0.233 | 0.316 | **0.244** | **0.273** | **0.458** | 0.299 |

## A.5 VISUALIZATION OF SPECTRUM

To demonstrate that the proposed FMR preserves signal periodicity and enhances the input signal, we performed a Fourier Transform on a real signal from ETTh1. We then plotted the spectrum of the original signal, the spectrum of the embedding obtained by direct single-linear-layer upsampling of the signal, and the spectrum after processing with the proposed FMR, as shown in Figure 11. Compared to direct linear embedding in the time domain (the commonly adopted approaches in existing methods include embedding techniques that incorporate timestamps), FMR retains more low-frequency information (where the signal's primary information is preserved, as seen in the second subplot) by learning variable-independent masks and performing linear interpolation in the frequency domain. Simultaneously, FMR exhibits a mid-to-high frequency energy distribution closer to that of the real signal, demonstrating better periodicity information retention capabilities than linear embedding directly in time domain.

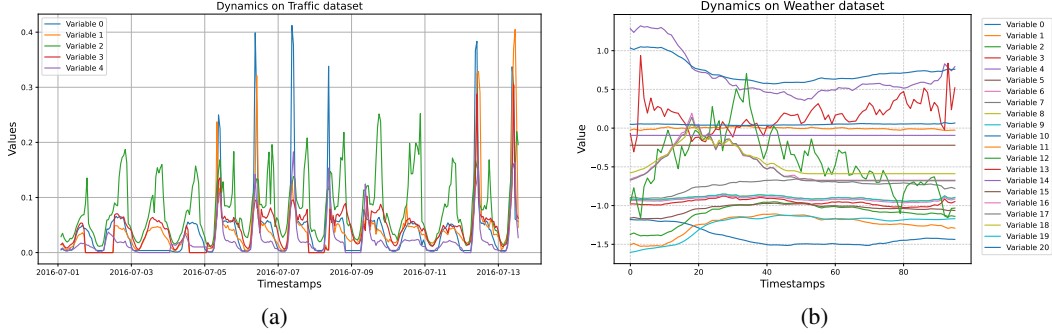

(a)          (b)

Figure 10: Time series trends of different variables in the Traffic (a) and Weather (b) datasets.

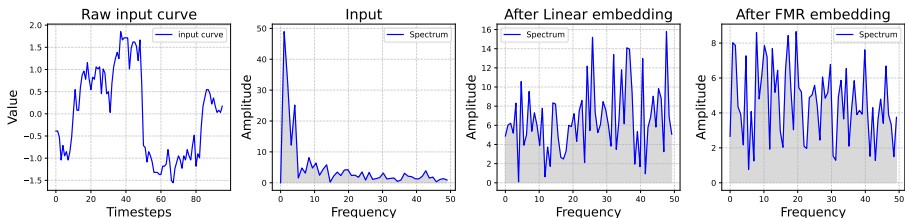

Figure 11: Visualization of spetrum as for raw signal and different embedding methods.

As depicted in Figure 12, we visualize the learned masks for variables 0, 2, 3, and 6 within the ETTh1 dataset. A high degree of similarity is observed between the masks for variables 0 and 2,

which is consistent with their strong interdependency (PCC = 1.0 in Figure 1). Conversely, the masks for variables 3 and 6, being learned independently, exhibit notable distinctions. Specifically, compared to other variables, the mask for variable 3 suppresses more high-frequency components, which may be because variable 3 exhibits greater volatility and noise. Importantly, we also observe that the masks for all variables predominantly preserve low-frequency components, which contain the signal's periodic and trend information. This highlights the ability of our FMR to learn adaptive, variable-specific masks that align with the unique properties of each series.

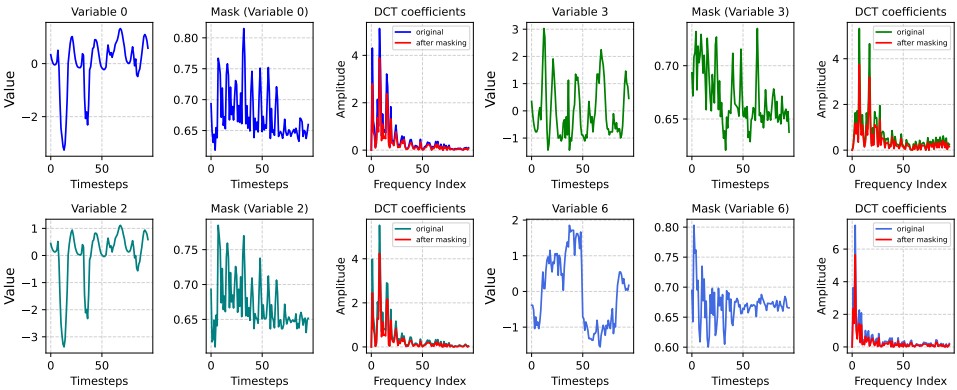

Figure 12: Visualization of learned masks on variable 0, 2, 3, and 6 of ETTh1.

### A.6 ADDITIONAL EVALUATION METRICS

To evaluate the correlations and similarities among variables in multivariate time series, we introduce Dynamic Time Warping (DTW) (Müller, 2007) and Pearson Correlation Coefficient (PCC) (Benesty et al., 2009).

**Dynamic Time Warping.** Dynamic Time Warping (DTW) calculates the similarity between two time series by finding the optimal matching path between them. DTW effectively handles irregularities such as temporal shifts and varying speeds within sequences, demonstrating strong performance in practical problems like speech and gesture recognition. Given two time series $Y = \{y_0, y_1, ..., y_{T-1}\} \in \mathbb{R}^T$ and $\hat{Y} = \{\hat{y}_0, \hat{y}_0, ..., \hat{y}_{T-1}\} \in \mathbb{R}^T$, the DTW distance can be formulated as:

$$\text{DTW}(Y, \hat{Y}) = \min_{\mathbf{A} \in \mathcal{A}(Y, \hat{Y})} \sum_{(i,j) \in \mathbf{A}} d(y_i, \hat{y}_j) = \sum_{(i,j) \in \mathbf{A}^*} d(y_i, \hat{y}_j), \tag{13}$$

Here, $d(\cdot, \cdot)$ represents a distance metric, commonly the squared Euclidean distance. A warping path, denoted by $\mathbf{A}$, comprises $K$ index pairs $\{(i_0, j_0), (i_1, j_1), \ldots, (i_{K-1}, j_{K-1})\}$, with indices $i_k, j_k$ ranging from 0 to $T - 1$. The collection of all valid warping paths is given by $\mathcal{A}(Y, \hat{Y})$. The optimal path, $\mathbf{A}^* \in \mathcal{A}(Y, \hat{Y})$, is the one that minimizes the cumulative distance across aligned time steps. A warping path $\mathbf{A}$ is deemed valid if it fulfills the subsequent conditions:

- **Boundary Constraint:** $(i_0, j_0) = (0, 0)$ and $(i_{K-1}, j_{K-1}) = (T - 1, T - 1)$.
- **Monotonicity Constraint:** The indices must be non-decreasing along the path, specifically $i_{k+1} \geq i_k$ and $j_{k+1} \geq j_k$ for all $k \in [0, K - 2]$.
- **Step Size Constraint:** Each step from $(i_k, j_k)$ to $(i_{k+1}, j_{k+1})$ must advance by one unit horizontally, vertically, or diagonally. Formally, $(i_{k+1} - i_k, j_{k+1} - j_k) \in \{(1, 0), (0, 1), (1, 1)\}$, for all $k \in [0, K - 2]$.

**Pearson Correlation Coefficient.** Pearson Correlation Coefficient (PCC) evaluates how strongly two variables are linearly related. Given two tokens $Y = \{y_0, y_1, ..., y_{T-1}\} \in \mathbb{R}^T$ and $\hat{Y} = \{\hat{y}_0, \hat{y}_0, ..., \hat{y}_{T-1}\} \in \mathbb{R}^T$ and their mean values $\bar{y}$ and $\bar{\hat{y}}$, PCC can be defined as:

$$\text{PCC}(Y, \hat{Y}) = \frac{\sum_{t=0}^{T-1}(y_t - \bar{y})(\hat{y}_t - \bar{\hat{y}})}{\sqrt{\sum_{t=0}^{T-1}(y_t - \bar{y})^2} \cdot \sqrt{\sum_{t=0}^{T-1}(\hat{y}_t - \bar{\hat{y}})^2}} \tag{14}$$

## A.7 TRANSFORMERS ARE FULLY-CONNECTED GNNs

GNNs employ the graph's connective structure to propagate and aggregate information among adjacent nodes. Let $h_i$ denote the node attributes of node $i$. In Graph Attention Networks (GATs) (Veličković et al., 2018), the relationship between the attributes of nodes $i$ and its neighbors $j \in \mathcal{N}_i$ can be computed as:

$$
\begin{aligned}
\psi(h_i^l, h_j^l) &= \text{Attention}(W_Q^l h_i^l, \{W_K^l h_j^l, \forall j \in \mathcal{N}_i\}, \{W_V^l h_j^l, \forall j \in \mathcal{N}_i\}), \\
&= \frac{\exp(W_Q^l h_i^l \cdot W_K^l h_j^l)}{\sum_{j' \in \mathcal{N}_i} \exp(W_Q^l h_i^l \cdot W_K^l h_{j'}^l)} \cdot W_V^l h_j^l,
\end{aligned}
\tag{15}
$$

where $W_Q^l, W_K^l, W_V^l \in \mathbb{R}^{d \times d}$ are learnable weight matrices. The $\psi(h_i^l, h_j^l)$ allows GATs to determine the significance of each neighbor for a given node in the aggregation process. The updated attribute features for node $i$ is derived by combining the information from all of its adjacent nodes:

$$
h_i^{l+1} = h_i^l + \sum_{j \in \mathcal{N}_i} \psi(h_i^l, h_j^l),
\tag{16}
$$

In variate Transformer, the self-attention captures correlations between all input tokens in MTS input $X$ as follows:

$$
\begin{aligned}
\psi(h_i^l, h_j^l) &= \text{Attention}(W_Q^l h_i^l, \{W_K^l h_j^l, \forall j \in X\}, \{W_V^l h_j^l, \forall j \in X\}), \\
&= \frac{\exp(W_Q^l h_i^l \cdot W_K^l h_j^l)}{\sum_{j' \in X} \exp(W_Q^l h_i^l \cdot W_K^l h_{j'}^l)} \cdot W_V^l h_j^l,
\end{aligned}
\tag{17}
$$

Here, $\psi(h_i^l, h_j^l)$ determines the message between the token pairs $(i, j)$, with each token's relative significance derived through an attention mechanism. Subsequently, these weighted messages from all tokens within the $X$ are combined via summation. Then, the token representations for token $i$ are updated using residual connection (He et al., 2016), layer normalization and MLP:

$$
h_i^{l+1} = \phi(h_i^l, m_i^l) = \text{MLP}(\text{LayerNorm}(h_i^l + \sum_{j \in X} \psi(h_i^l, h_j^l))).
\tag{18}
$$

Equation. 15 bears a strong resemblance to the self-attention mechanism within the Transformer. The primary distinction lies in the scope of the aggregation: whereas in GNN the index $j$ is constrained to the local neighborhood of node $i$, in Transformer's self-attention, the aggregation is performed over the entire set of tokens in the sequence. This effectively means the Transformer can be interpreted as a special instance of a GNN operating on a dynamically-weighted, fully-connected graph, where every token is considered a neighbor to all others.

iTransformer presented insightful experiments (see Table 3 in iTransformer paper) where they replaced the FFN with a self-attention layer, essentially constructing a Transformer with two self-attention layers. *The experimental results indicated that simply stacking multiple self-attention layers did not facilitate the learning of correct inter-variate dependencies and temporal patterns. Therefore, based on the aforementioned analysis, we resort to GNNs for modeling inter-variate dependencies within the deeper layers of the Transformer.* Theoretically, GNNs and self-attention layers are closely linked in their ability to capture global relationships. **This insight forms the basis of our novel, theoretically grounded perspective: how to effectively integrate graph-learned dependencies with inter-variate relationships captured by Transformers.**

## A.8 WHY WE USE MULTI-HOP GRAPH CONVOLUTION NETWORK WITH THE SAME GRAPH STRUCTURE?

We reformulate Equation. 16 as Equation. 20. We note that the key distinction between GAT and GCN lies in their adjacency matrix weights: in GAT, the weights of the adjacency matrix are learned dynamically and are different for each layer $l$ (a.k.a., each hop in multi-hop GNN), whereas a standard GCN employs a fixed adjacency matrix for feature propagation.

$$
H^{l+1} = H^l + \sigma(A^l H^l W^l),
\tag{19}
$$

Table 5: Comparative Performance of different GNNs in Modeling of Inter-Variable Dependencies in the Network Deep Layer.

| GNN | $F$ | ETTm1 MSE | ETTm1 MAE | ETTh1 MSE | ETTh1 MAE | Weather MSE | Weather MAE | ECL MSE | ECL MAE |
|---|---|---|---|---|---|---|---|---|---|
| GCN | 96 | **0.315** | **0.344** | **0.372** | **0.387** | 0.152 | 0.190 | **0.137** | **0.227** |
| | 192 | **0.366** | **0.372** | **0.424** | **0.418** | 0.203 | 0.239 | **0.155** | **0.243** |
| | 336 | **0.398** | **0.395** | 0.473 | 0.443 | **0.257** | **0.279** | **0.170** | **0.259** |
| | 720 | 0.472 | 0.435 | 0.473 | 0.464 | **0.338** | **0.334** | **0.198** | **0.283** |
| GAT | 96 | 0.321 | 0.348 | 0.375 | 0.388 | **0.150** | **0.190** | 0.141 | 0.231 |
| | 192 | 0.370 | 0.374 | 0.428 | 0.420 | 0.209 | 0.242 | 0.165 | 0.247 |
| | 336 | 0.405 | 0.399 | **0.469** | **0.441** | 0.262 | 0.281 | 0.175 | 0.262 |
| | 720 | **0.465** | **0.432** | **0.471** | **0.463** | 0.344 | 0.337 | 0.213 | 0.289 |

where $\sigma$ is the activation function, $H$ and $W$ are node features and learnable weights, respectively. In our CGTFra, graph structures are dynamically learned from global inputs via linear transformations and gating mechanisms, rather than being predetermined. Consequently, the typical distinctions between GCN and GAT in terms of their edge weights fixed or varying cross different hops are attenuated. As presented in Table 5, we compare the performance difference within the CGT-Fra framework when using either identical or distinct graph weights for information aggregation at each hop in DGL (Essentially, based on input-constructed graph structures, we implement standard GCN and GAT). Notably, when each hop employs a dynamically relearned graph structure based on its current input, we apply a consistency constraint (CAL) to the graph structure of the final hop. The results indicate that using dynamically updated edge weights at each hop does not yield significant performance gains. We attribute this to the fact that **shallow self-attention layers capture inter-variate dependencies based on global tokens, learning association weights only once**. Although GNNs in DGL employ multi-hop strategies to aggregate information from broader nodes, the graph structure proposed is also dynamically learned from the global input tokens (i.e., the output of the self-attention layer). Therefore, utilizing the same graph structure across all hops is more conducive to subsequent consistent alignment of inter-variate dependencies. **Therefore, the DGL within the proposed CGTFra framework employs consistent adjacency matrix weights across all hops, akin to a standard multi-hop GCN. Furthermore, for different layers ($L$ in Figure 4) of CGTFra, the graph structure in DGL is distinct (input-dependent), which aligns with the re-computation of attention scores in each self-attention layer.**

### A.9 THE THEORETICAL GUARANTEES OF CAL FROM INFORMATION BOTTLENECK PRINCIPLE.

The Information Bottleneck (IB) principle (Tishby et al., 2000) aims to find a compressed representation, denoted as $Z$, that maximally preserves information about a target variable $Y$ while simultaneously compressing the input $X$. This objective is typically formulated as the following optimization problem:

$$\max I(Z;Y) - \beta * I(Z;X), \tag{20}$$

where $I(\cdot;\cdot)$ represents mutual information and $\beta$ is a Lagrange multiplier. Within our CGTFra framework, we can interpret the self-attention map (MCM) as a high-bandwidth, yet potentially noisy, representation of the inter-variable relationships in the input $X$. While its mutual information with the input, $I(\text{MCM};X)$, is high, much of this information may constitute noise irrelevant to the final prediction target $Y$. Conversely, the GNN's adjacency matrix, $A$, is intended to be the compressed and cleaner representation $Z$ that we seek to learn. The goal is for $A$ to discard the noise present in MCM and retain only the structured information pertinent to predicting $Y$. In this context, our alignment loss, $\mathcal{L}_{align} = \text{KL}(\text{MCM}||A)$, can be viewed as a proxy or an upper bound for the compression term, $I(Z;X)$, in the IB objective. By minimizing $\text{KL}(\text{MCM}||A)$, we encourage the learned adjacency matrix $A$ not to deviate excessively from the attention map MCM. This implicitly controls the mutual information $I(A;\text{MCM})$, and by extension, $I(A;X)$, aligning our method with the core IB principle of learning a compressed yet informative representation.

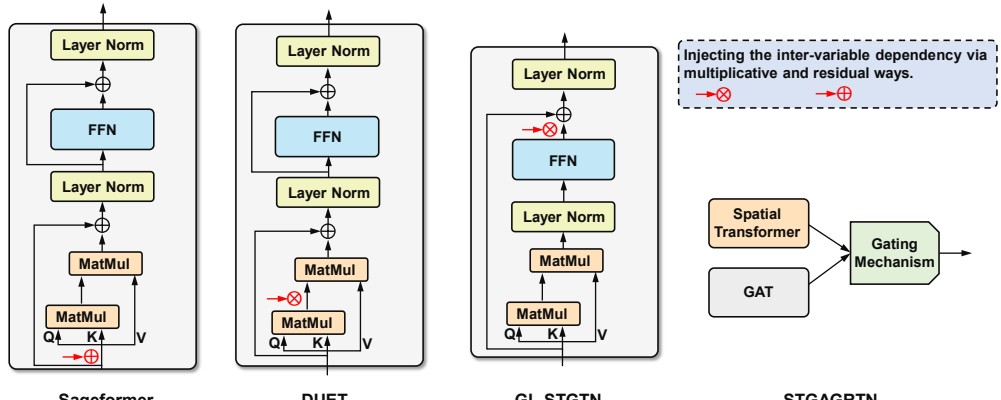

Figure 13: Typical Transformer-based approaches to modeling inter-variable dependencies.

### A.10  EXISTING TRANSFORMER-BASED METHODS MODELING IVD.

We analyze four representative Transformer-based approaches for modeling inter-variable dependencies, namely Sageformer (Zhang et al., 2024), DUET (Qiu et al., 2025), GL-STGTN (Li et al., 2024), and STGAGRTN (Wu et al., 2023a). As illustrated in Figure 13, these methods embed inter-variable dependencies primarily by incorporating them as masks or biases within the Transformer, which we categorize as Figure 3(b).

- Sageformer (Zhang et al., 2024): SageFormer first employs a GNN (with totally self-learned graph structure) to capture inter-variate correlations from the input MTS. The resulting global, graph-enhanced embeddings are then fused with the original series to serve as the input for a **vanilla Transformer** (i.e., temporal Transformer), which subsequently models temporal dependencies.

- DUET (Qiu et al., 2025): DUET captures IVD in the frequency domain using metric learning. The resulting dependency is then integrated into the self-attention mechanism as a mask for the attention scores of **variate Transformer**.

- GL-STGTN (Li et al., 2024): GL-STGTN learns the graph structure from both global and local perspectives, and then the learned inter-variable dependencies are then encoded into a spatial attention mechanism.

- STGAGRTN (Wu et al., 2023a): STGAGRTN utilizes a gating mechanism to fuse the inter-variable dependencies learned separately by a GAT and a proposed spatial Transformer.

However, such approaches do not adequately address the challenge of modeling inter-variable dependencies in deeper layers. Although GL-STGTN introduces inter-variable relations after the feed-forward network (FFN), the additional branch is prone to capturing spurious correlations. More importantly, unlike our work, **GL-STGTN does not explore and account for the consistency between shallow- and deep-layer modeling of inter-variable dependencies.**

Unlike GL-STGTN with graph learning, which uses the **raw input** $X$, or methods like Sageformer and MSGNet that rely **solely on self-learned node embeddings**, our graph constructor is uniquely informed by a combination of **outputs from the self-attention layer** and **learnable node embeddings**. Importantly, to the best of our knowledge, we are the first to comprehensively analyze the connections and differences between self-attention (within Variate Transformers) and GNNs for modeling IVD. Furthermore, as we emphasize earlier, we have compactly integrated the two linear layers of the original FFN into our DGL module. It is this compact architecture that allows our DGL to serve as a general-purpose IVD modeling method for the deeper layers of Variate Transformers—**a level of universality not achieved by existing dynamic graph learning techniques**. Crucially, the introduction of DGL and CAL significantly accelerates the convergence of both training and validation losses, achieving an 8.78% reduction in MSE of iTransformer (see Figure 8). This substantial performance gain is achieved with only a minimal computational overhead—time complexity of $\mathcal{O}(N(D + nd + D * nd))$ (see Equation 5), which is linear with respect to the number of

variables $N$, where $nd$ is a small hyperparameter (e.g., 8, 10, or 32), and $D$ is the hidden dimension. We believe the introduced performance and efficiency is promising.

## A.11 DATASET DETAILS

As shown in Table 6, total 13 datasets utilized in our study encompass data from five domains: Temperature, Finance, Weather, Electricity, and Transportation, providing a comprehensive assessment of a model's effectiveness and generality. * Forecastability is computed by one minus the entropy of Fourier decomposition, a lower value indicating worse predictability.

Table 6: Details of different datasets.

| Datasets | Variables | Dataset Size | Frequency | Forecastability* | Information |
|---|---|---|---|---|---|
| ETTm1 | 7 | (34465, 11521, 11521) | 15min | 0.46 | Temperature |
| ETTm2 | 7 | (34465, 11521, 11521) | 15min | 0.55 | Temperature |
| ETTh1 | 7 | (8545, 2881, 2881) | 15min | 0.38 | Temperature |
| ETTh2 | 7 | (8545, 2881, 2881) | 15min | 0.45 | Temperature |
| Exchange | 8 | (5120, 665, 1422) | Daily | - | Finance |
| Weather | 21 | (36792, 5271, 10540) | Hourly | 0.75 | Weather |
| Solar-Energy | 137 | (36601, 5161, 10417) | 10min | 0.33 | Electricity |
| Electricity | 321 | (18317, 2633, 5261) | 10min | 0.77 | Electricity |
| Traffic | 862 | (12185, 1757, 3509) | Hourly | 0.68 | Transportation |
| PEMS03 | 358 | (15701, 5216, 434) | 5min | 0.65 | Transportation |
| PEMS04 | 307 | (10172, 3375, 281) | 5min | 0.45 | Transportation |
| PEMS07 | 883 | (16911, 5622, 468) | 5min | 0.58 | Transportation |
| PEMS08 | 170 | (10690, 3548, 265) | 5min | 0.52 | Transportation |

## A.12 IMPLEMENTATION DETAILS

All experiments are conducted on two NVIDIA GeForce RTX 3090 GPUs. We use Adam optimizer with $\mathcal{L} = \mathcal{L}_{MAE} + \lambda \mathcal{L}_{align}$ as the loss function for model optimization and evaluate the prediction performance with the Mean Squared Error: MSE $= \frac{1}{n} \sum_{i=1}^{n} (y_i - \hat{y}_i)^2$ and MAE, where $y_i$ and $\hat{y}_i$ represent the ground truth and predicted value at time $i$, respectively.

By default, we employ Kullback-Leibler (KL) divergence as the $\mathcal{L}_{align}$. And the number of stacked layers for CGTFra is selected from 1, 2, or 4, with 1 or 2 layers typically used for datasets with fewer variables, and 2 or 4 layers for those with more variables. DGL's default number of hops is 2, and the batch size is set between 16 and 128. Hyperparameter sensitivity analysis is provided in Appendix A.16.

## A.13 ADDITIONAL RESULTS

In this section, we present the complete comparison results for both long-term and short-term forecasting, as shown in Table 7 and Table 8, respectively. To further compare model performance under longer input horizons, we also provide results with an input length of 336 in Table 9. Across short-term forecasting, long-term forecasting, and extended input lengths, CGTFra consistently demonstrates superior predictive performance, underscoring its overall effectiveness.

Theoretically, a longer given historical input enables models to capture more information, leading to more accurate predictions. However, higher dimensionality can also introduce side effects such as model overfitting and training difficulty. To investigate the performance differences of various models across different historical input lengths, as illustrated in Figure 14, we evaluate the performance of five methods. We observed that all models achieved relatively comparable prediction accuracy when the given input length is 336. Further increasing the input length, however, potentially led to a decline in performance. Consequently, as shown in Table 9, we also conducted a detailed comparison of how different models perform when predicting four distinct output lengths, with an input length of 336.

Table 7: Long-term forecasting results with forecasting horizons $F \in \{96, 192, 336, 720\}$ and fixed look-back length $T$=96. **Bold**/underline: Best/second best one. "-" indicates that the original method was not evaluated in the corresponding scenario.

| Models | CGTFra (ours) | | DUET (KDD'25) | | TimePro (ICML'25) | | Soatten (AAAI'25) | | VCformer (IJCAI'24) | | FilterNet (NeurIPS'24) | | iTransformer (ICLR'24) | | MSGNet (AAAI'24) | | PatchTST (ICLR'23) | |
|---|---|---|---|---|---|---|---|---|---|---|---|---|---|---|---|---|---|---|
| Metrics | MSE | MAE | MSE | MAE | MSE | MAE | MSE | MAE | MSE | MAE | MSE | MAE | MSE | MAE | MSE | MAE | MSE | MAE |
| ETTm1 96 | **0.315** | **0.344** | 0.324 | 0.354 | 0.326 | 0.364 | 0.329 | 0.365 | 0.319 | 0.359 | 0.318 | 0.358 | 0.334 | 0.368 | 0.319 | 0.366 | 0.329 | 0.367 |
| ETTm1 192 | 0.366 | **0.372** | 0.369 | 0.379 | 0.367 | 0.383 | 0.37 | 0.387 | **0.364** | 0.382 | **0.364** | 0.383 | 0.377 | 0.391 | 0.376 | 0.397 | 0.367 | 0.385 |
| ETTm1 336 | 0.398 | **0.395** | 0.404 | 0.402 | 0.402 | 0.409 | 0.401 | 0.407 | 0.399 | 0.405 | **0.396** | 0.406 | 0.426 | 0.420 | 0.417 | 0.422 | 0.399 | 0.410 |
| ETTm1 720 | 0.472 | **0.435** | 0.463 | 0.437 | 0.469 | 0.446 | 0.474 | 0.447 | 0.467 | 0.442 | 0.456 | 0.444 | 0.491 | 0.459 | 0.481 | 0.458 | **0.454** | 0.439 |
| ETTm2 96 | **0.171** | **0.249** | 0.174 | 0.255 | 0.178 | 0.260 | 0.180 | 0.264 | 0.180 | 0.266 | 0.174 | 0.257 | 0.180 | 0.264 | 0.177 | 0.262 | 0.175 | 0.259 |
| ETTm2 192 | **0.238** | **0.293** | 0.243 | 0.302 | 0.242 | 0.303 | 0.245 | 0.306 | 0.245 | 0.306 | 0.240 | 0.300 | 0.250 | 0.309 | 0.247 | 0.307 | 0.241 | 0.302 |
| ETTm2 336 | 0.300 | **0.333** | 0.304 | 0.341 | 0.303 | 0.342 | 0.312 | 0.349 | 0.307 | 0.345 | **0.297** | 0.339 | 0.311 | 0.348 | 0.312 | 0.346 | 0.305 | 0.343 |
| ETTm2 720 | 0.397 | **0.391** | 0.399 | 0.397 | 0.400 | 0.399 | 0.411 | 0.406 | 0.406 | 0.402 | **0.392** | 0.393 | 0.412 | 0.407 | 0.414 | 0.403 | 0.402 | 0.400 |
| ETTh1 96 | **0.372** | **0.387** | 0.377 | 0.393 | 0.375 | 0.398 | 0.383 | 0.406 | 0.376 | 0.397 | 0.375 | 0.394 | 0.386 | 0.405 | 0.390 | 0.411 | 0.414 | 0.419 |
| ETTh1 192 | **0.424** | **0.418** | 0.429 | 0.425 | 0.427 | 0.429 | 0.440 | 0.433 | 0.431 | 0.427 | 0.436 | 0.422 | 0.441 | 0.436 | 0.442 | 0.442 | 0.460 | 0.445 |
| ETTh1 336 | 0.473 | **0.443** | **0.471** | 0.446 | 0.472 | 0.450 | 0.475 | 0.449 | 0.473 | 0.449 | 0.476 | **0.443** | 0.487 | 0.458 | 0.480 | 0.468 | 0.501 | 0.466 |
| ETTh1 720 | **0.473** | **0.464** | 0.496 | 0.480 | 0.476 | 0.474 | 0.491 | 0.477 | 0.476 | 0.474 | 0.474 | 0.469 | 0.503 | 0.491 | 0.494 | 0.488 | 0.500 | 0.488 |
| ETTh2 96 | **0.288** | **0.336** | 0.296 | 0.345 | 0.293 | 0.345 | 0.295 | 0.348 | 0.292 | 0.344 | 0.292 | 0.343 | 0.297 | 0.349 | 0.328 | 0.371 | 0.302 | 0.348 |
| ETTh2 192 | **0.364** | **0.384** | 0.368 | 0.389 | 0.367 | 0.394 | 0.380 | 0.398 | 0.377 | 0.396 | 0.369 | 0.395 | 0.380 | 0.400 | 0.402 | 0.414 | 0.388 | 0.400 |
| ETTh2 336 | **0.410** | **0.422** | 0.411 | **0.422** | 0.419 | 0.431 | 0.420 | 0.431 | 0.417 | 0.430 | 0.420 | 0.432 | 0.428 | 0.432 | 0.435 | 0.443 | 0.426 | 0.433 |
| ETTh2 720 | 0.414 | **0.433** | **0.412** | 0.434 | 0.427 | 0.445 | 0.419 | 0.441 | 0.423 | 0.443 | 0.430 | 0.446 | 0.427 | 0.445 | 0.417 | 0.441 | 0.431 | 0.446 |
| Exchange 96 | **0.083** | **0.202** | 0.086 | 0.205 | 0.085 | 0.204 | 0.085 | 0.204 | 0.085 | 0.205 | **0.083** | **0.202** | 0.086 | 0.206 | 0.102 | 0.23 | 0.088 | 0.205 |
| Exchange 192 | **0.173** | **0.296** | 0.182 | 0.305 | 0.178 | 0.299 | 0.175 | 0.299 | 0.176 | 0.299 | 0.174 | **0.296** | 0.177 | 0.299 | 0.195 | 0.317 | 0.176 | 0.299 |
| Exchange 336 | 0.324 | 0.412 | 0.310 | 0.403 | 0.328 | 0.414 | 0.330 | 0.417 | 0.328 | 0.415 | 0.326 | 0.413 | 0.331 | 0.417 | 0.359 | 0.436 | **0.301** | **0.397** |
| Exchange 720 | **0.668** | **0.619** | 0.693 | 0.624 | 0.817 | 0.679 | 0.844 | 0.695 | 0.830 | 0.688 | 0.840 | 0.670 | 0.847 | 0.691 | 0.940 | 0.738 | 0.901 | 0.714 |
| Weather 96 | **0.152** | **0.190** | 0.163 | 0.202 | 0.166 | 0.207 | 0.161 | 0.206 | 0.171 | 0.220 | 0.162 | 0.207 | 0.174 | 0.214 | 0.163 | 0.212 | 0.177 | 0.218 |
| Weather 192 | **0.203** | **0.239** | 0.218 | 0.252 | 0.216 | 0.254 | 0.208 | 0.250 | 0.230 | 0.266 | 0.210 | 0.250 | 0.221 | 0.254 | 0.212 | 0.254 | 0.225 | 0.259 |
| Weather 336 | **0.257** | **0.279** | 0.274 | 0.294 | 0.273 | 0.296 | 0.264 | 0.291 | 0.265 | 0.290 | 0.265 | 0.290 | 0.278 | 0.296 | 0.272 | 0.299 | 0.278 | 0.297 |
| Weather 720 | **0.338** | **0.334** | 0.349 | 0.343 | 0.351 | 0.346 | 0.347 | 0.346 | 0.352 | 0.344 | 0.342 | 0.340 | 0.358 | 0.347 | 0.350 | 0.348 | 0.354 | 0.348 |
| Electricity 96 | **0.137** | **0.227** | 0.145 | 0.233 | 0.139 | 0.234 | **0.137** | 0.232 | 0.150 | 0.242 | 0.147 | 0.245 | 0.148 | 0.240 | 0.165 | 0.274 | 0.181 | 0.270 |
| Electricity 192 | **0.155** | **0.243** | 0.163 | 0.248 | 0.156 | 0.249 | **0.155** | 0.247 | 0.167 | 0.250 | 0.160 | 0.250 | 0.162 | 0.253 | 0.184 | 0.292 | 0.188 | 0.274 |
| Electricity 336 | **0.170** | **0.259** | 0.175 | 0.262 | 0.172 | 0.267 | 0.171 | 0.265 | 0.182 | 0.270 | 0.173 | 0.267 | 0.178 | 0.269 | 0.195 | 0.302 | 0.204 | 0.293 |
| Electricity 720 | **0.198** | **0.283** | 0.204 | 0.291 | 0.209 | 0.299 | 0.200 | 0.290 | 0.221 | 0.302 | 0.210 | 0.309 | 0.225 | 0.317 | 0.231 | 0.332 | 0.246 | 0.324 |
| Solar 96 | **0.191** | **0.205** | 0.200 | 0.207 | 0.196 | 0.237 | 0.198 | 0.239 | - | - | - | - | 0.203 | 0.237 | - | - | 0.234 | 0.286 |
| Solar 192 | **0.218** | **0.225** | 0.228 | 0.233 | 0.231 | 0.263 | 0.228 | 0.259 | - | - | - | - | 0.233 | 0.261 | - | - | 0.267 | 0.310 |
| Solar 336 | **0.238** | **0.240** | 0.262 | 0.244 | 0.250 | 0.281 | 0.244 | 0.272 | - | - | - | - | 0.248 | 0.273 | - | - | 0.290 | 0.315 |
| Solar 720 | 0.249 | **0.242** | 0.258 | 0.249 | 0.253 | 0.285 | **0.246** | 0.275 | - | - | - | - | 0.249 | 0.275 | - | - | 0.289 | 0.317 |
| Traffic 96 | **0.387** | **0.239** | 0.407 | 0.252 | - | - | 0.401 | 0.270 | 0.454 | 0.310 | 0.430 | 0.294 | 0.395 | 0.268 | - | - | 0.544 | 0.359 |
| Traffic 192 | **0.417** | **0.249** | 0.431 | 0.262 | - | - | 0.424 | 0.281 | 0.468 | 0.315 | 0.452 | 0.307 | **0.417** | 0.276 | - | - | 0.540 | 0.354 |
| Traffic 336 | 0.434 | **0.261** | 0.456 | 0.269 | - | - | 0.445 | 0.288 | 0.485 | 0.316 | 0.470 | 0.316 | **0.433** | 0.283 | - | - | 0.551 | 0.358 |
| Traffic 720 | **0.472** | **0.279** | 0.509 | 0.292 | - | - | 0.479 | 0.306 | 0.524 | 0.348 | 0.498 | 0.323 | 0.467 | 0.302 | - | - | 0.586 | 0.375 |
| $1^{st}$ Count | **25** | **35** | 2 | 1 | 0 | 0 | 3 | 0 | 1 | 0 | 5 | 3 | 3 | 0 | 0 | 0 | 2 | 1 |

Table 8: Short-term forecasting results with forecasting horizons $F \in \{12, 24, 48, 96\}$ and fixed look-back length $T$=96.

| Models | CGTFra (ours) | | iTransformer (ICLR'24) | | RLinear (ArXiv'23) | | PatchTST (ICLR'23) | | Crossformer (ICLR'23) | | TiDE (TMLR'23) | | TimesNet (ICLR'23) | | DLinear (AAAI'23) | |
|---|---|---|---|---|---|---|---|---|---|---|---|---|---|---|---|---|
| Metrics | MSE | MAE | MSE | MAE | MSE | MAE | MSE | MAE | MSE | MAE | MSE | MAE | MSE | MAE | MSE | MAE |
| PEMS03 12 | **0.060** | **0.159** | 0.071 | 0.174 | 0.126 | 0.236 | 0.099 | 0.216 | 0.090 | 0.203 | 0.178 | 0.305 | 0.085 | 0.192 | 0.122 | 0.243 |
| PEMS03 24 | **0.079** | **0.184** | 0.093 | 0.201 | 0.246 | 0.334 | 0.142 | 0.259 | 0.121 | 0.240 | 0.257 | 0.371 | 0.118 | 0.223 | 0.201 | 0.317 |
| PEMS03 48 | **0.119** | **0.228** | 0.125 | 0.236 | 0.551 | 0.529 | 0.211 | 0.319 | 0.202 | 0.317 | 0.379 | 0.463 | 0.155 | 0.260 | 0.333 | 0.425 |
| PEMS03 96 | 0.173 | 0.278 | **0.164** | **0.275** | 1.057 | 0.787 | 0.269 | 0.370 | 0.262 | 0.367 | 0.490 | 0.539 | 0.228 | 0.317 | 0.457 | 0.515 |
| PEMS04 12 | **0.070** | **0.169** | 0.078 | 0.183 | 0.138 | 0.252 | 0.105 | 0.224 | 0.098 | 0.218 | 0.219 | 0.340 | 0.087 | 0.195 | 0.148 | 0.272 |
| PEMS04 24 | **0.084** | **0.187** | 0.095 | 0.205 | 0.258 | 0.348 | 0.153 | 0.275 | 0.131 | 0.256 | 0.292 | 0.398 | 0.103 | 0.215 | 0.224 | 0.340 |
| PEMS04 48 | **0.112** | **0.220** | 0.120 | 0.233 | 0.572 | 0.544 | 0.229 | 0.339 | 0.205 | 0.326 | 0.409 | 0.478 | 0.136 | 0.250 | 0.355 | 0.437 |
| PEMS04 96 | 0.153 | **0.260** | **0.150** | 0.262 | 1.137 | 0.820 | 0.291 | 0.389 | 0.402 | 0.457 | 0.492 | 0.532 | 0.190 | 0.303 | 0.452 | 0.504 |
| PEMS07 12 | **0.056** | **0.146** | 0.067 | 0.165 | 0.118 | 0.235 | 0.095 | 0.207 | 0.094 | 0.200 | 0.173 | 0.304 | 0.082 | 0.181 | 0.115 | 0.242 |
| PEMS07 24 | **0.075** | **0.167** | 0.088 | 0.190 | 0.242 | 0.341 | 0.150 | 0.262 | 0.139 | 0.247 | 0.271 | 0.383 | 0.101 | 0.204 | 0.210 | 0.329 |
| PEMS07 48 | **0.101** | **0.197** | 0.110 | 0.215 | 0.562 | 0.541 | 0.253 | 0.340 | 0.311 | 0.369 | 0.446 | 0.495 | 0.134 | 0.238 | 0.398 | 0.458 |
| PEMS07 96 | 0.144 | **0.242** | **0.139** | 0.245 | 1.096 | 0.795 | 0.346 | 0.404 | 0.396 | 0.442 | 0.628 | 0.577 | 0.181 | 0.279 | 0.594 | 0.553 |
| PEMS08 12 | **0.071** | **0.167** | 0.079 | 0.182 | 0.133 | 0.247 | 0.168 | 0.232 | 0.165 | 0.214 | 0.227 | 0.343 | 0.112 | 0.212 | 0.154 | 0.276 |
| PEMS08 24 | **0.096** | **0.193** | 0.115 | 0.219 | 0.249 | 0.343 | 0.224 | 0.281 | 0.215 | 0.260 | 0.318 | 0.409 | 0.141 | 0.238 | 0.248 | 0.353 |
| PEMS08 48 | **0.152** | 0.243 | 0.186 | **0.235** | 0.569 | 0.544 | 0.321 | 0.354 | 0.315 | 0.355 | 0.497 | 0.510 | 0.198 | 0.283 | 0.440 | 0.470 |
| PEMS08 96 | 0.263 | 0.299 | **0.221** | **0.267** | 1.166 | 0.814 | 0.408 | 0.417 | 0.377 | 0.397 | 0.721 | 0.592 | 0.320 | 0.351 | 0.674 | 0.565 |
| $1^{st}$ Count | **13** | **13** | 4 | 3 | 0 | 0 | 0 | 0 | 0 | 0 | 0 | 0 | 0 | 0 | 0 | 0 |

Furthermore, to validate the effectiveness of our Graph Transformer, which is predicated on modeling consistency between shallow and deep IVD, we conducted a comparison against two other sota Graph Transformer methods: Ada-MSHyper (Shang et al., 2024) and Sageformer (Zhang et al., 2024). The results are presented in Table 10. While Ada-MSHyper exhibits certain advantages on the ETT datasets, this is primarily reflected in its slightly better MSE scores. In contrast, CGTFra demonstrates a more pronounced advantage on datasets with a larger number of variables, such as ECL and Traffic. For instance, on the Traffic dataset, CGTFra achieves an average reduction of 7.89% in MAE compared to Ada-MSHyper.

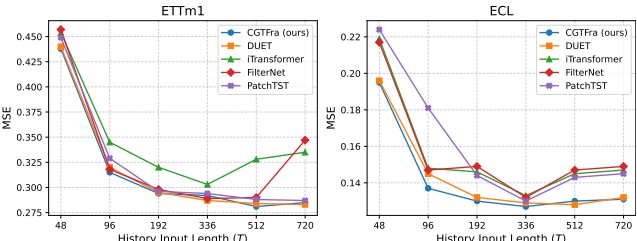

Figure 14: Performance comparison with different historical input lengths (Predict $F$=96).

Table 9: Multivariate forecasting results with forecasting horizons $F \in \{96, 192, 336, 720\}$ and fixed look-back window size $T = 336$.

| Models | | CGTFra (ours) | | FilterNet (NeurIPS'24) | | iTransformer (ICLR'24) | | PatchTST (ICLR'23) | | TimesNet (ICLR'23) | |
|---|---|---|---|---|---|---|---|---|---|---|---|
| Metrics | | MSE | MAE | MSE | MAE | MSE | MAE | MSE | MAE | MSE | MAE |
| ETTm1 | 96 | 0.292 | **0.335** | **0.289** | 0.344 | 0.303 | 0.357 | 0.294 | 0.345 | 0.335 | 0.380 |
| | 192 | **0.326** | **0.361** | 0.331 | 0.369 | 0.345 | 0.383 | 0.334 | 0.371 | 0.358 | 0.388 |
| | 336 | 0.366 | **0.381** | **0.364** | 0.389 | 0.382 | 0.405 | 0.371 | 0.392 | 0.406 | 0.418 |
| | 720 | **0.422** | **0.417** | 0.425 | 0.423 | 0.443 | 0.439 | 0.421 | 0.419 | 0.449 | 0.443 |
| ETTh1 | 96 | **0.378** | **0.396** | 0.379 | 0.404 | 0.402 | 0.418 | 0.381 | 0.405 | 0.398 | 0.418 |
| | 192 | **0.415** | **0.420** | 0.417 | 0.428 | 0.450 | 0.449 | 0.442 | 0.446 | 0.447 | 0.449 |
| | 336 | 0.438 | **0.435** | **0.437** | 0.443 | 0.479 | 0.470 | 0.445 | 0.454 | 0.493 | 0.468 |
| | 720 | **0.442** | **0.428** | 0.458 | 0.472 | 0.584 | 0.548 | 0.490 | 0.493 | 0.518 | 0.504 |
| Exchange | 96 | **0.087** | **0.211** | 0.087 | 0.216 | 0.099 | 0.226 | 0.093 | 0.213 | 0.117 | 0.253 |
| | 192 | 0.169 | **0.299** | **0.163** | 0.301 | 0.216 | 0.337 | 0.194 | 0.315 | 0.298 | 0.410 |
| | 336 | 0.312 | 0.417 | **0.287** | **0.399** | 0.395 | 0.466 | 0.354 | 0.435 | 0.456 | 0.513 |
| | 720 | 0.673 | 0.621 | **0.413** | **0.492** | 0.962 | 0.745 | 0.903 | 0.712 | 1.608 | 0.961 |
| Weather | 96 | **0.147** | 0.187 | 0.150 | **0.183** | 0.164 | 0.216 | 0.151 | 0.197 | 0.172 | 0.220 |
| | 192 | **0.188** | 0.229 | 0.193 | **0.221** | 0.205 | 0.251 | 0.197 | 0.244 | 0.219 | 0.261 |
| | 336 | **0.241** | 0.272 | 0.246 | **0.258** | 0.256 | 0.290 | 0.251 | 0.285 | 0.280 | 0.306 |
| | 720 | **0.308** | 0.331 | **0.308** | **0.295** | 0.326 | 0.338 | 0.321 | 0.335 | 0.365 | 0.359 |
| Electricity | 96 | **0.127** | **0.219** | 0.132 | 0.224 | 0.133 | 0.229 | 0.130 | 0.222 | 0.168 | 0.272 |
| | 192 | **0.137** | **0.216** | 0.143 | 0.237 | 0.156 | 0.251 | 0.148 | 0.240 | 0.184 | 0.289 |
| | 336 | **0.153** | **0.253** | 0.155 | **0.253** | 0.172 | 0.267 | 0.167 | 0.261 | 0.198 | 0.300 |
| | 720 | **0.193** | **0.285** | 0.195 | 0.292 | 0.209 | 0.304 | 0.202 | 0.291 | 0.220 | 0.320 |
| $1^{st}$ Count | | **15** | **14** | 7 | 8 | 0 | 0 | 0 | 0 | 0 | 0 |

Table 10: Performance comparison of CGTFra and two other graph Transformers.

| Models | | ETTm1 | | ETTm2 | | ETTh1 | | ETTh2 | | Exchange | | Weather | | ECL | | Traffic | |
|---|---|---|---|---|---|---|---|---|---|---|---|---|---|---|---|---|---|
| Metrics | | MSE | MAE | MSE | MAE | MSE | MAE | MSE | MAE | MSE | MAE | MSE | MAE | MSE | MAE | MSE | MAE |
| CGTFra | 96 | 0.315 | **0.344** | **0.171** | **0.249** | 0.372 | **0.387** | **0.288** | **0.336** | 0.083 | 0.202 | **0.152** | **0.190** | 0.137 | 0.227 | **0.387** | **0.239** |
| | 192 | 0.366 | **0.372** | 0.238 | **0.293** | **0.424** | **0.418** | **0.364** | **0.384** | 0.173 | 0.296 | **0.203** | 0.239 | 0.155 | 0.243 | **0.417** | **0.249** |
| | 336 | 0.398 | **0.395** | 0.300 | 0.333 | 0.473 | **0.443** | 0.410 | 0.422 | 0.324 | 0.412 | 0.257 | 0.279 | 0.170 | 0.259 | 0.434 | 0.261 |
| | 720 | 0.472 | **0.435** | 0.397 | **0.391** | 0.473 | **0.464** | 0.414 | 0.433 | 0.668 | 0.619 | 0.338 | 0.334 | 0.198 | 0.283 | 0.472 | **0.279** |
| | Avg | 0.388 | **0.386** | 0.277 | **0.316** | **0.436** | 0.428 | 0.369 | 0.394 | 0.312 | 0.382 | 0.238 | 0.260 | 0.165 | 0.253 | 0.427 | 0.257 |
| Ada-MSHyper | 96 | **0.309** | 0.357 | 0.173 | 0.261 | 0.376 | 0.395 | 0.291 | 0.338 | - | - | 0.161 | 0.202 | 0.144 | 0.241 | 0.405 | 0.263 |
| | 192 | **0.362** | 0.385 | **0.235** | 0.307 | 0.436 | **0.418** | 0.370 | 0.389 | - | - | 0.209 | 0.248 | 0.160 | 0.247 | 0.419 | 0.275 |
| | 336 | **0.394** | 0.409 | **0.295** | 0.340 | 0.468 | 0.447 | 0.426 | 0.434 | - | - | 0.263 | 0.289 | 0.176 | 0.273 | 0.439 | 0.278 |
| | 720 | **0.461** | 0.447 | **0.389** | 0.402 | **0.469** | 0.472 | 0.418 | 0.439 | - | - | 0.349 | 0.346 | 0.212 | 0.293 | **0.467** | 0.299 |
| | Avg | **0.382** | 0.400 | **0.273** | 0.328 | 0.437 | 0.433 | 0.376 | 0.400 | - | - | 0.246 | 0.271 | 0.173 | 0.264 | 0.433 | 0.279 |
| Sageformer | 96 | 0.333 | 0.366 | 0.175 | 0.259 | 0.377 | 0.394 | 0.291 | 0.339 | **0.082** | **0.201** | 0.165 | 0.207 | 0.148 | 0.246 | - | - |
| | 192 | 0.371 | 0.389 | 0.241 | 0.301 | 0.428 | 0.426 | 0.376 | 0.394 | 0.177 | 0.299 | 0.211 | 0.251 | 0.163 | 0.248 | - | - |
| | 336 | 0.406 | 0.409 | 0.302 | 0.341 | **0.466** | 0.448 | 0.417 | 0.428 | 0.333 | 0.418 | 0.269 | 0.292 | 0.181 | 0.265 | - | - |
| | 720 | 0.478 | 0.449 | 0.399 | 0.396 | 0.487 | 0.476 | 0.422 | 0.441 | 0.866 | 0.702 | 0.347 | 0.345 | 0.209 | 0.306 | - | - |
| | Avg | 0.397 | 0.403 | 0.279 | 0.324 | 0.440 | 0.436 | 0.377 | 0.401 | 0.365 | 0.405 | 0.248 | 0.274 | 0.175 | 0.266 | - | - |

A.14    VERIFICATION OF FRAMEWORK GENERALITY

To validate the extensibility of the three core designs proposed in this work, we perform corresponding module replacements or introduce CAL for seven existing models, including DUET, iTransformer, VCformer, CASA, FilterNet, iFlashformer, iFlowformer, iInformer, and iReformer. **To ensure a fair comparison, all baseline experiments were conducted using their released code and hyperparameters, under identical hyperparameters, random seeds, and experimental hardware and software environment versions. Additionally, our released code includes the source files, scripts, and documentation necessary to reproduce these experiments**. As shown in Table 11, we observe that **introducing FMR alone yields only marginal gains**. In contrast, **incorporating DGL significantly yields greater performance improvements to a certain extent, highlighting the importance of explicitly modeling IVD in deeper layers**. Building on this, the introduction of CAL further improves forecasting performance across multiple heterogeneous datasets, with the effect **being more pronounced for VCformer**. For instance, on the ETTh1 dataset, MSE decreases from 0.398 to 0.382. Moreover, we have to acknowledge that achieving performance gains by modifying sota methods while using identical hyperparameters poses considerable challenges.

- DUET (Qiu et al., 2025): DUET captures IVD by employing metric learning in the frequency domain, subsequently feeding IVD as masks to the self-attention scores within a variable Transformer. As DUET does not involve linear upsampling, our proposed FMR cannot be directly validated. Concurrently, DUET also lacks deep-layer IVD modeling. Therefore, we embed DGL and CAL into DUET for comparative experiments. `https://github.com/decisionintelligence/DUET`

- iTransformer (Liu et al., 2024): iTransformer encodes timestamp information into the input signals via concatenation, and computes inter-variable correlations among tokens corresponding to individual variables, and then employs FFNs to capture deep temporal dynamics. Its architecture is consistent with the standard Transformer (i.e., temporal Transformer), except that inverted token embedding. Accordingly, we replace the FFN in iTransformer with DGL to emphasize the importance of deep layer IVD, and further incorporate CAL on top of DGL to enhance consistent IVD modeling across both deep and shallow layers. `https://github.com/thuml/iTransformer`

- VCformer (Yang et al., 2024): VCformer likewise encodes timestamp information into the input via concatenation, and computes the inter-series correlation on different lags between queries and keys, and employ another Koopman theory-based temporal learner (namely KTD) to replace the FFN. Therefore, VCformer also captures IVD only at shallow layers. To validate extensibility, we replace their input embedding layer with our FMR, and replace KTD with the proposed DGL. `https://github.com/CSyyn/VCformer`

- CASA (Lee et al., 2025): CASA replaces the self-attention layer in the Transformer with a CNN autoencoder-based score attention, and **is therefore not a Transformer architecture. Since CASA encodes inputs using a single linear layer, we only replace its input embedding method with the proposed FMR to evaluate the generality of FMR**. `https://github.com/lmh9507/CASA`

The complete results for the other variate Transformers are reported in Table 12. As some variant Transformers redesign more efficient self-attention layers that may lose explicit attention scores, CAL cannot be integrated into these variate Transformers.

In Figure 1, we present the DTW and PCC of the ETTh1 dataset, they characterize the true similarities and dependencies among variables in multivariate time series. **MSE and MAE, focusing solely on point-wise numerical discrepancies, overlook overall time-series shape similarity and fail to measure inter-variable correlations.** Therefore, we conduct the effectiveness verification of DGL and CAL on three existing baselines with DTW and PCC as comparative metrics (their definitions are provided in Appendix A.6). **DTW prioritizes trend pattern matching, while PCC quantifies the model's capacity to capture co-variation among variables.** As shown in Table 13, **incorporating DGL and CAL enables iTransformer to achieve lower MSE and MAE values, along with superior DTW and PCC, indicating that deep modeling of IVD enhances forecasting of future fluctuations (in terms of magnitude) and improves the accuracy of dependency modeling (in terms of similarity). These results collectively demonstrate the strong generalizability of DGL and CAL.**

Table 11: Verification of Framework Generality. Full results for four prediction length and fixed input $T$=96. All results were reproduced using their released code and hyperparameters. "iTrans" is iTransformer. "-" indicates that the original method was not evaluated in the corresponding scenario or we faced the issue of out of memory. Additional evaluation metrics are provided in Table 13.

| Models | | Len | ETTm1 | | ETTm2 | | ETTh1 | | ETTh2 | | Exchange | | Weather | | ECL | | Solar | | Traffic | |
|---|---|---|---|---|---|---|---|---|---|---|---|---|---|---|---|---|---|---|---|---|
| | Metrics | | MSE | MAE | MSE | MAE | MSE | MAE | MSE | MAE | MSE | MAE | MSE | MAE | MSE | MAE | MSE | MAE | MSE | MAE |
| DUET | original | 96 | **0.322** | 0.354 | 0.174 | 0.254 | 0.389 | 0.400 | 0.295 | 0.345 | 0.084 | 0.203 | 0.162 | 0.201 | 0.146 | 0.233 | 0.249 | 0.269 | **0.407** | 0.252 |
| | | 192 | 0.370 | 0.380 | 0.239 | 0.299 | 0.431 | 0.426 | 0.370 | 0.391 | 0.179 | 0.300 | 0.217 | 0.251 | 0.163 | 0.249 | **0.221** | **0.230** | **0.431** | **0.261** |
| | | 336 | 0.407 | 0.403 | **0.301** | **0.339** | 0.473 | 0.450 | 0.408 | **0.419** | 0.285 | 0.390 | 0.268 | 0.290 | 0.174 | 0.261 | **0.245** | **0.242** | 0.458 | 0.271 |
| | | 720 | 0.464 | 0.439 | 0.400 | **0.396** | 0.502 | 0.484 | 0.415 | 0.435 | 0.686 | 0.625 | 0.342 | 0.339 | 0.204 | 0.288 | **0.247** | **0.244** | 0.504 | 0.291 |
| DUET | + DGL | 96 | 0.322 | 0.353 | **0.171** | **0.251** | 0.383 | 0.398 | 0.290 | 0.341 | **0.083** | **0.202** | 0.166 | 0.208 | **0.139** | **0.228** | 0.230 | 0.249 | 0.410 | 0.253 |
| | | 192 | **0.367** | **0.376** | 0.237 | 0.295 | 0.427 | 0.423 | 0.366 | 0.388 | 0.177 | 0.299 | 0.212 | 0.250 | **0.155** | **0.243** | 0.233 | 0.246 | 0.434 | 0.262 |
| | | 336 | 0.406 | 0.403 | 0.302 | 0.339 | 0.476 | 0.448 | **0.407** | 0.421 | 0.287 | 0.391 | 0.268 | 0.291 | **0.169** | **0.257** | 0.258 | 0.271 | **0.453** | **0.269** |
| | | 720 | **0.460** | **0.433** | 0.399 | 0.396 | 0.488 | 0.476 | **0.409** | **0.431** | 0.636 | 0.600 | 0.342 | 0.339 | 0.202 | 0.291 | 0.251 | 0.267 | **0.496** | **0.288** |
| DUET | + CAL | 96 | 0.324 | 0.356 | 0.174 | 0.254 | **0.378** | **0.393** | **0.289** | **0.340** | 0.083 | 0.202 | 0.152 | 0.192 | **0.139** | **0.228** | 0.231 | 0.250 | 0.412 | 0.253 |
| | | 192 | 0.370 | 0.379 | 0.239 | 0.297 | **0.427** | **0.421** | 0.369 | 0.389 | 0.177 | 0.299 | 0.205 | 0.244 | **0.155** | **0.243** | 0.238 | 0.248 | 0.433 | **0.260** |
| | | 336 | **0.405** | **0.399** | 0.307 | 0.344 | **0.472** | **0.445** | 0.415 | 0.424 | **0.285** | **0.388** | 0.256 | 0.280 | **0.169** | 0.258 | 0.247 | 0.245 | 0.462 | 0.273 |
| | | 720 | 0.466 | 0.437 | 0.406 | 0.403 | **0.475** | **0.469** | 0.420 | 0.434 | 0.673 | 0.615 | **0.333** | **0.338** | **0.193** | **0.284** | 0.253 | 0.268 | 0.501 | 0.291 |
| iTrans | original | 96 | 0.342 | 0.377 | 0.186 | 0.272 | 0.387 | 0.405 | 0.301 | 0.350 | 0.086 | 0.206 | 0.181 | 0.221 | 0.148 | 0.239 | 0.201 | 0.234 | **0.392** | **0.268** |
| | | 192 | 0.383 | 0.396 | 0.254 | 0.314 | 0.441 | 0.436 | 0.381 | 0.399 | 0.181 | 0.303 | 0.226 | 0.259 | 0.167 | 0.258 | 0.239 | 0.263 | **0.413** | 0.277 |
| | | 336 | 0.418 | 0.418 | 0.317 | 0.353 | 0.491 | 0.462 | 0.423 | 0.432 | 0.338 | 0.422 | 0.283 | 0.300 | 0.181 | 0.275 | 0.248 | 0.272 | **0.425** | 0.283 |
| | | 720 | 0.487 | 0.456 | 0.416 | 0.408 | 0.509 | 0.494 | 0.430 | 0.446 | 0.869 | 0.704 | 0.359 | 0.351 | 0.209 | 0.299 | 0.250 | 0.275 | 0.459 | 0.300 |
| iTrans | + FMR | 96 | 0.340 | 0.373 | **0.183** | **0.265** | **0.382** | **0.398** | 0.299 | 0.350 | 0.084 | 0.204 | 0.180 | 0.222 | 0.141 | 0.235 | **0.199** | 0.237 | 0.393 | **0.268** |
| | | 192 | 0.377 | **0.389** | **0.249** | **0.309** | 0.434 | 0.429 | 0.379 | 0.399 | 0.176 | 0.299 | 0.222 | 0.258 | 0.157 | 0.250 | **0.233** | **0.259** | 0.413 | 0.276 |
| | | 336 | 0.412 | **0.411** | **0.314** | **0.350** | 0.483 | 0.454 | 0.419 | 0.430 | 0.339 | 0.423 | 0.279 | 0.300 | 0.171 | 0.264 | **0.242** | **0.269** | 0.428 | 0.282 |
| | | 720 | **0.481** | **0.450** | 0.418 | 0.409 | 0.492 | 0.480 | **0.426** | 0.445 | **0.834** | **0.690** | 0.356 | 0.350 | 0.233 | 0.316 | **0.244** | **0.273** | 0.458 | 0.299 |
| iTrans | + DGL | 96 | **0.331** | 0.368 | **0.183** | 0.268 | 0.384 | 0.403 | 0.305 | 0.355 | 0.086 | 0.207 | 0.167 | 0.211 | 0.137 | 0.233 | 0.200 | 0.238 | 0.410 | 0.281 |
| | | 192 | **0.376** | 0.392 | 0.253 | 0.313 | 0.435 | 0.432 | 0.392 | 0.406 | 0.179 | 0.303 | 0.214 | 0.254 | **0.154** | 0.248 | 0.239 | 0.264 | 0.421 | 0.281 |
| | | 336 | **0.409** | 0.412 | 0.317 | 0.352 | 0.481 | 0.453 | 0.426 | 0.436 | **0.336** | 0.420 | 0.275 | 0.299 | **0.167** | 0.263 | 0.248 | 0.274 | 0.440 | 0.286 |
| | | 720 | 0.483 | 0.453 | 0.417 | 0.408 | 0.494 | 0.481 | 0.435 | 0.451 | 0.870 | 0.704 | 0.353 | 0.349 | 0.219 | 0.307 | 0.249 | 0.276 | 0.466 | 0.304 |
| iTrans | + CAL | 96 | 0.333 | 0.368 | 0.184 | 0.269 | **0.382** | 0.402 | 0.306 | 0.354 | 0.086 | 0.207 | **0.166** | **0.211** | **0.135** | **0.232** | 0.199 | **0.234** | 0.402 | 0.272 |
| | | 192 | 0.378 | 0.390 | 0.253 | 0.314 | **0.434** | 0.431 | 0.384 | 0.401 | 0.177 | 0.301 | **0.213** | **0.254** | **0.154** | **0.248** | 0.234 | 0.262 | 0.437 | 0.282 |
| | | 336 | 0.415 | 0.412 | 0.316 | 0.352 | **0.481** | 0.455 | 0.430 | 0.435 | 0.338 | 0.422 | **0.267** | **0.294** | **0.167** | **0.262** | 0.251 | 0.275 | 0.451 | 0.287 |
| | | 720 | 0.482 | 0.450 | **0.415** | **0.407** | **0.479** | **0.473** | 0.426 | 0.444 | 0.861 | 0.702 | **0.350** | **0.347** | **0.194** | **0.288** | 0.249 | 0.276 | 0.471 | 0.304 |
| VCformer | original | 96 | 0.331 | 0.364 | 0.184 | 0.266 | 0.405 | 0.410 | 0.302 | **0.349** | 0.085 | 0.206 | 0.186 | 0.224 | 0.152 | 0.246 | - | - | - | - |
| | | 192 | 0.379 | 0.389 | 0.250 | 0.309 | 0.455 | 0.439 | 0.383 | 0.396 | **0.175** | 0.300 | 0.238 | 0.266 | 0.170 | 0.261 | - | - | - | - |
| | | 336 | 0.419 | 0.416 | 0.318 | 0.352 | 0.530 | 0.476 | 0.421 | **0.430** | 0.327 | 0.415 | 0.288 | 0.303 | 0.186 | 0.277 | - | - | - | - |
| | | 720 | 0.487 | 0.453 | 0.414 | 0.407 | 0.561 | 0.515 | **0.429** | 0.446 | 0.844 | 0.691 | 0.365 | 0.352 | 0.235 | 0.328 | - | - | - | - |
| VCformer | + FMR | 96 | 0.333 | 0.365 | 0.183 | 0.265 | 0.393 | 0.401 | 0.308 | 0.351 | 0.088 | 0.211 | 0.184 | 0.224 | 0.147 | 0.241 | - | - | - | - |
| | | 192 | **0.373** | **0.387** | 0.250 | 0.309 | 0.451 | **0.434** | 0.383 | **0.395** | 0.176 | 0.300 | 0.231 | 0.264 | 0.163 | 0.255 | - | - | - | - |
| | | 336 | **0.410** | **0.410** | 0.314 | 0.350 | **0.484** | 0.450 | **0.419** | **0.430** | 0.336 | 0.421 | 0.285 | 0.302 | 0.177 | 0.272 | - | - | - | - |
| | | 720 | 0.476 | 0.447 | 0.415 | 0.406 | 0.499 | **0.480** | 0.431 | 0.447 | 0.869 | 0.703 | 0.361 | 0.351 | 0.241 | 0.332 | - | - | - | - |
| VCformer | + DGL | 96 | **0.323** | **0.359** | 0.184 | 0.269 | 0.398 | 0.410 | 0.305 | 0.352 | 0.085 | 0.206 | 0.165 | **0.208** | 0.139 | 0.236 | - | - | - | - |
| | | 192 | 0.379 | 0.389 | 0.249 | 0.310 | 0.447 | 0.439 | 0.387 | 0.401 | **0.175** | **0.299** | 0.210 | 0.252 | 0.158 | **0.248** | - | - | - | - |
| | | 336 | 0.415 | 0.412 | 0.310 | 0.348 | 0.484 | 0.456 | 0.425 | 0.433 | **0.326** | **0.412** | 0.270 | **0.293** | 0.173 | 0.266 | - | - | - | - |
| | | 720 | **0.473** | **0.445** | **0.412** | **0.405** | 0.494 | 0.482 | 0.440 | 0.453 | 0.868 | 0.700 | **0.351** | 0.347 | 0.224 | 0.314 | - | - | - | - |
| VCformer | + CAL | 96 | 0.328 | 0.364 | **0.179** | **0.262** | **0.382** | **0.400** | 0.300 | **0.349** | 0.085 | 0.205 | **0.164** | **0.208** | **0.135** | **0.233** | - | - | - | - |
| | | 192 | 0.375 | 0.389 | **0.246** | **0.307** | 0.438 | 0.434 | 0.380 | 0.397 | 0.211 | **0.299** | 0.211 | 0.251 | **0.157** | 0.249 | - | - | - | - |
| | | 336 | 0.415 | 0.412 | **0.309** | **0.347** | 0.485 | 0.457 | 0.436 | 0.440 | 0.339 | 0.423 | 0.271 | 0.295 | **0.170** | 0.264 | - | - | - | - |
| | | 720 | 0.485 | 0.453 | 0.414 | 0.406 | 0.497 | 0.486 | 0.436 | 0.452 | 0.846 | 0.696 | **0.351** | **0.346** | **0.207** | 0.299 | - | - | - | - |
| CASA | original | 96 | 0.322 | **0.359** | 0.175 | 0.257 | **0.378** | 0.403 | 0.298 | 0.347 | - | - | **0.162** | **0.207** | 0.140 | 0.236 | 0.193 | 0.234 | **0.392** | **0.260** |
| | | 192 | **0.368** | **0.386** | 0.241 | 0.300 | 0.428 | 0.429 | 0.375 | 0.396 | - | - | 0.209 | 0.251 | 0.160 | **0.253** | 0.227 | 0.260 | **0.415** | **0.274** |
| | | 336 | **0.407** | **0.409** | 0.299 | 0.339 | **0.478** | **0.453** | 0.420 | 0.431 | - | - | 0.267 | 0.292 | 0.181 | 0.274 | 0.240 | 0.274 | **0.434** | 0.281 |
| | | 720 | 0.468 | 0.447 | 0.399 | 0.397 | 0.482 | 0.476 | 0.439 | 0.451 | - | - | 0.359 | 0.352 | 0.206 | 0.298 | 0.242 | 0.276 | **0.468** | 0.296 |
| CASA | + FMR | 96 | **0.321** | **0.359** | 0.174 | 0.256 | 0.378 | 0.401 | **0.294** | **0.346** | - | - | 0.163 | **0.207** | **0.136** | **0.232** | 0.192 | 0.233 | 0.405 | 0.262 |
| | | 192 | 0.369 | **0.386** | 0.240 | 0.299 | 0.426 | 0.428 | 0.372 | 0.395 | - | - | **0.207** | **0.248** | 0.159 | 0.253 | 0.222 | 0.258 | 0.432 | **0.274** |
| | | 336 | 0.418 | 0.416 | 0.298 | 0.337 | 0.480 | 0.454 | 0.418 | 0.430 | - | - | **0.264** | 0.291 | 0.179 | 0.273 | 0.238 | 0.272 | 0.447 | **0.280** |
| | | 720 | **0.460** | **0.444** | 0.398 | 0.396 | 0.484 | 0.477 | 0.429 | 0.446 | - | - | **0.347** | **0.344** | 0.204 | 0.295 | 0.240 | 0.274 | 0.492 | 0.301 |
| FilterNet | original | 96 | **0.317** | **0.357** | 0.175 | 0.257 | 0.381 | 0.399 | 0.296 | 0.346 | - | - | 0.164 | 0.210 | 0.147 | 0.242 | - | - | 0.431 | 0.295 |
| | | 192 | **0.364** | 0.384 | 0.239 | 0.300 | 0.440 | 0.428 | 0.369 | 0.396 | - | - | 0.214 | 0.256 | 0.162 | 0.254 | - | - | 0.448 | 0.298 |
| | | 336 | 0.396 | 0.407 | **0.295** | 0.337 | 0.487 | 0.451 | 0.420 | **0.432** | - | - | 0.273 | 0.299 | **0.177** | **0.272** | - | - | 0.465 | 0.303 |
| | | 720 | 0.457 | 0.444 | 0.398 | **0.395** | 0.494 | 0.471 | 0.432 | 0.447 | - | - | 0.359 | 0.353 | 0.228 | 0.318 | - | - | 0.497 | 0.320 |
| FilterNet | + FMR | 96 | 0.318 | 0.359 | 0.174 | 0.255 | 0.376 | 0.397 | 0.293 | 0.343 | - | - | 0.160 | 0.206 | 0.144 | 0.239 | - | - | **0.422** | **0.289** |
| | | 192 | 0.364 | 0.383 | 0.238 | 0.299 | 0.438 | 0.427 | 0.368 | 0.394 | - | - | 0.209 | 0.252 | 0.159 | 0.252 | - | - | **0.445** | **0.295** |
| | | 336 | **0.395** | 0.406 | 0.295 | 0.337 | 0.489 | 0.451 | 0.416 | 0.433 | - | - | 0.270 | 0.296 | 0.177 | 0.273 | - | - | **0.462** | **0.301** |
| | | 720 | **0.455** | **0.442** | 0.396 | 0.395 | 0.496 | 0.472 | 0.438 | 0.450 | - | - | 0.353 | 0.350 | 0.228 | 0.321 | - | - | **0.494** | **0.317** |

Table 12: Verification of Framework Generality on Variate Transformers (fixed input length $T$=96). As some variant Transformers redesign more efficient self-attention layers that may lose explicit attention scores, CAL cannot be integrated into these variate Transformers. To further evaluate the effectiveness of DGL and CAL, we employed additional evaluation metrics as seen in Table 13.

| Datasets | | | ETTh1 | | Weather | | ECL | | Solar | |
|---|---|---|---|---|---|---|---|---|---|---|
| Metrics | | | MSE | MAE | MSE | MAE | MSE | MAE | MSE | MAE |
| iTransformer | original | 96 | 0.387 | 0.405 | 0.181 | 0.221 | 0.148 | 0.239 | 0.201 | **0.234** |
| | | 192 | 0.441 | 0.436 | 0.226 | 0.259 | 0.167 | 0.258 | 0.239 | 0.263 |
| | | 336 | 0.491 | 0.462 | 0.283 | 0.300 | 0.181 | 0.275 | 0.248 | 0.272 |
| | | 720 | 0.509 | 0.494 | 0.359 | 0.351 | 0.209 | 0.299 | 0.250 | **0.275** |
| | + DGL | 96 | 0.384 | 0.403 | 0.167 | 0.212 | 0.137 | 0.233 | 0.200 | 0.238 |
| | | 192 | 0.435 | 0.432 | 0.214 | 0.255 | **0.154** | **0.248** | 0.236 | 0.264 |
| | | 336 | **0.481** | **0.453** | 0.275 | 0.299 | **0.167** | 0.263 | **0.248** | 0.274 |
| | | 720 | 0.494 | 0.481 | 0.353 | 0.349 | 0.219 | 0.307 | **0.249** | 0.276 |
| | + CAL | 96 | **0.382** | **0.402** | **0.166** | **0.211** | **0.135** | **0.232** | **0.199** | **0.234** |
| | | 192 | **0.434** | **0.431** | **0.213** | **0.254** | **0.154** | **0.248** | 0.234 | 0.262 |
| | | 336 | **0.481** | 0.455 | **0.267** | **0.294** | 0.167 | 0.262 | 0.251 | 0.275 |
| | | 720 | **0.479** | **0.473** | **0.350** | **0.347** | **0.194** | **0.288** | 0.249 | 0.276 |
| iFlashformer | original | 96 | 0.388 | 0.406 | 0.180 | 0.221 | 0.164 | 0.254 | 0.213 | 0.251 |
| | | 192 | **0.438** | 0.435 | 0.227 | 0.259 | **0.175** | **0.263** | **0.242** | **0.275** |
| | | 336 | 0.487 | 0.458 | 0.283 | 0.300 | **0.192** | **0.280** | **0.263** | **0.291** |
| | | 720 | 0.504 | 0.491 | 0.360 | 0.351 | **0.232** | **0.314** | **0.267** | **0.296** |
| | + DGL | 96 | **0.384** | **0.402** | **0.171** | **0.216** | **0.160** | **0.253** | **0.209** | **0.250** |
| | | 192 | 0.441 | **0.434** | **0.216** | **0.255** | **0.175** | 0.265 | 0.246 | 0.275 |
| | | 336 | **0.484** | **0.455** | **0.278** | **0.299** | 0.193 | 0.283 | 0.266 | 0.292 |
| | | 720 | **0.500** | **0.483** | **0.352** | **0.348** | **0.232** | 0.315 | 0.273 | 0.298 |
| iFlowformer | original | 96 | **0.385** | **0.402** | 0.187 | 0.226 | 0.169 | 0.255 | **0.215** | 0.255 |
| | | 192 | 0.446 | 0.437 | 0.230 | 0.262 | 0.180 | 0.265 | **0.246** | **0.277** |
| | | 336 | 0.503 | 0.470 | 0.285 | 0.301 | 0.198 | **0.283** | **0.266** | **0.292** |
| | | 720 | 0.559 | 0.522 | 0.363 | 0.352 | **0.238** | **0.317** | **0.272** | **0.297** |
| | + DGL | 96 | 0.387 | 0.403 | **0.176** | **0.220** | **0.163** | **0.254** | 0.218 | **0.254** |
| | | 192 | **0.443** | **0.435** | **0.220** | **0.257** | **0.174** | **0.265** | 0.251 | 0.279 |
| | | 336 | **0.484** | **0.454** | **0.273** | **0.296** | **0.197** | 0.285 | 0.277 | 0.297 |
| | | 720 | **0.500** | **0.481** | **0.351** | **0.345** | **0.238** | 0.319 | 0.285 | 0.304 |
| iInformer | original | 96 | 0.388 | 0.404 | 0.169 | 0.213 | 0.168 | 0.255 | **0.220** | 0.264 |
| | | 192 | 0.445 | 0.436 | 0.217 | 0.254 | 0.181 | **0.266** | **0.254** | 0.287 |
| | | 336 | 0.492 | 0.461 | 0.273 | 0.296 | 0.198 | **0.284** | **0.278** | **0.304** |
| | | 720 | 0.504 | 0.490 | 0.353 | 0.348 | 0.242 | **0.319** | **0.280** | **0.305** |
| | + DGL | 96 | 0.390 | 0.405 | 0.168 | 0.214 | 0.164 | 0.255 | 0.223 | 0.261 |
| | | 192 | 0.445 | 0.435 | 0.212 | 0.253 | 0.179 | 0.267 | 0.263 | 0.287 |
| | | 336 | 0.489 | 0.457 | 0.271 | 0.295 | 0.198 | 0.286 | 0.287 | 0.305 |
| | | 720 | 0.501 | 0.482 | 0.351 | 0.347 | 0.241 | 0.321 | 0.292 | 0.308 |
| | + CAL | 96 | **0.386** | **0.401** | **0.168** | **0.213** | **0.159** | **0.251** | 0.223 | **0.260** |
| | | 192 | **0.443** | **0.433** | **0.211** | **0.252** | **0.178** | 0.267 | 0.261 | **0.286** |
| | | 336 | **0.482** | **0.453** | **0.268** | **0.293** | **0.197** | 0.285 | 0.290 | 0.305 |
| | | 720 | **0.494** | **0.478** | **0.347** | **0.344** | **0.240** | 0.320 | 0.293 | 0.308 |
| iReformer | original | 96 | 0.386 | 0.402 | 0.185 | 0.226 | 0.169 | 0.257 | 0.222 | 0.263 |
| | | 192 | 0.447 | 0.437 | 0.230 | 0.262 | 0.180 | **0.266** | **0.255** | 0.285 |
| | | 336 | 0.502 | 0.469 | 0.283 | 0.301 | 0.198 | **0.284** | **0.277** | **0.302** |
| | | 720 | 0.548 | 0.516 | 0.359 | 0.349 | 0.241 | **0.319** | **0.280** | **0.303** |
| | + DGL | 96 | **0.383** | **0.401** | **0.176** | **0.220** | **0.161** | **0.254** | **0.221** | **0.259** |
| | | 192 | **0.442** | **0.434** | **0.222** | **0.259** | **0.176** | **0.266** | 0.258 | **0.283** |
| | | 336 | **0.480** | **0.452** | **0.274** | **0.296** | **0.195** | 0.285 | 0.285 | **0.302** |
| | | 720 | **0.492** | **0.478** | **0.351** | **0.346** | **0.239** | 0.320 | 0.289 | 0.305 |

Table 13: Additional evaluation metrics for evaluating the effectiveness of DGL and CAL.

| Datasets | | | ETTh1 MSE ↓ | DTW ↓ | PCC ↑ | Weather MSE ↓ | DTW ↓ | PCC ↑ | ECL MSE ↓ | DTW ↓ | PCC ↑ | Solar MSE ↓ | DTW ↓ | PCC ↑ |
|---|---|---|---|---|---|---|---|---|---|---|---|---|---|---|
| DUET | original | 96 | 0.389 | 13.95 | 0.559 | 0.162 | 16.16 | 0.398 | 0.146 | 65.68 | 0.901 | 0.249 | 45.68 | 0.841 |
| | | 192 | 0.431 | 21.09 | 0.532 | 0.217 | 27.12 | 0.362 | 0.163 | 97.05 | 0.897 | **0.221** | **66.59** | **0.917** |
| | | 336 | 0.473 | 29.41 | 0.506 | 0.268 | 40.31 | 0.339 | 0.174 | 132.90 | 0.894 | **0.245** | **94.00** | 0.882 |
| | | 720 | 0.502 | 44.36 | 0.468 | 0.342 | 67.91 | 0.315 | 0.204 | 218.83 | 0.879 | **0.247** | **140.27** | 0.866 |
| | | Avg | 0.449 | 27.20 | 0.516 | 0.247 | 37.88 | 0.354 | 0.172 | 128.62 | 0.893 | **0.241** | 86.64 | **0.877** |
| | + DGL | 96 | 0.383 | 13.85 | 0.562 | 0.166 | 16.19 | 0.392 | **0.139** | 63.70 | **0.913** | 0.230 | **42.30** | 0.875 |
| | | 192 | **0.427** | 20.01 | 0.526 | 0.212 | 26.88 | 0.366 | **0.155** | 95.67 | 0.903 | 0.233 | 67.25 | 0.899 |
| | | 336 | 0.476 | 29.62 | 0.494 | 0.268 | 40.33 | 0.338 | **0.169** | 132.73 | **0.899** | 0.258 | 94.90 | 0.870 |
| | | 720 | 0.488 | 44.28 | 0.466 | 0.342 | 67.89 | 0.317 | 0.202 | 216.49 | 0.882 | 0.251 | 140.29 | 0.858 |
| | | Avg | 0.444 | 26.94 | 0.512 | 0.247 | 37.82 | 0.353 | 0.166 | 127.15 | 0.899 | 0.243 | 86.19 | 0.876 |
| | + CAL | 96 | **0.378** | **13.77** | **0.564** | **0.152** | **16.08** | **0.405** | **0.139** | **63.41** | 0.908 | 0.231 | 42.36 | 0.873 |
| | | 192 | **0.427** | **19.98** | **0.530** | **0.205** | **26.63** | **0.371** | **0.155** | **95.49** | **0.909** | 0.238 | 67.68 | 0.892 |
| | | 336 | **0.472** | **29.36** | **0.500** | **0.256** | **40.01** | **0.349** | **0.169** | **132.64** | 0.899 | 0.247 | 94.07 | **0.883** |
| | | 720 | **0.475** | **43.21** | **0.470** | **0.333** | **66.89** | **0.326** | **0.193** | **210.58** | **0.894** | 0.253 | 140.32 | 0.850 |
| | | Avg | **0.438** | **26.58** | 0.516 | **0.237** | **37.40** | **0.363** | **0.164** | **125.53** | **0.903** | 0.242 | **86.11** | 0.875 |
| iTransformer | original | 96 | 0.387 | 13.92 | 0.561 | 0.181 | 16.97 | 0.364 | 0.148 | 65.77 | 0.906 | 0.201 | 39.56 | 0.902 |
| | | 192 | 0.441 | 21.19 | 0.527 | 0.226 | 27.74 | 0.354 | 0.167 | 99.55 | 0.899 | 0.239 | 67.96 | **0.895** |
| | | 336 | 0.491 | 29.87 | 0.496 | 0.283 | 41.04 | 0.334 | 0.181 | 137.33 | 0.892 | **0.248** | 94.01 | 0.880 |
| | | 720 | 0.509 | 44.49 | 0.468 | 0.359 | 68.12 | 0.300 | 0.209 | 218.70 | 0.878 | 0.250 | 140.50 | 0.862 |
| | | Avg | 0.457 | 27.37 | 0.513 | 0.262 | 38.47 | 0.338 | 0.176 | 130.33 | 0.894 | 0.235 | 85.51 | 0.885 |
| | + DGL | 96 | 0.384 | 13.89 | 0.562 | 0.167 | **16.27** | 0.383 | 0.137 | 63.31 | 0.911 | 0.200 | 39.80 | 0.904 |
| | | 192 | 0.435 | 21.23 | 0.526 | 0.214 | **26.81** | 0.363 | **0.154** | 95.75 | 0.903 | 0.236 | 68.17 | **0.895** |
| | | 336 | **0.481** | 29.66 | 0.494 | 0.275 | **40.24** | 0.332 | **0.167** | 132.85 | 0.896 | **0.248** | 94.20 | **0.881** |
| | | 720 | 0.494 | 44.40 | 0.466 | 0.353 | **67.95** | 0.290 | 0.219 | 223.27 | 0.875 | **0.249** | 140.32 | 0.863 |
| | | Avg | 0.449 | 27.30 | 0.512 | 0.252 | **37.82** | 0.342 | 0.169 | 128.79 | 0.896 | **0.233** | 85.62 | **0.886** |
| | + CAL | 96 | **0.382** | **13.83** | **0.564** | **0.166** | 16.28 | **0.394** | **0.135** | **62.96** | **0.912** | **0.199** | **39.53** | **0.905** |
| | | 192 | **0.434** | 21.07 | **0.530** | **0.213** | 27.01 | 0.367 | **0.154** | **95.44** | **0.905** | 0.234 | **67.32** | **0.895** |
| | | 336 | **0.481** | **29.55** | 0.500 | **0.267** | 40.36 | **0.340** | **0.167** | 132.68 | 0.898 | 0.251 | 94.60 | 0.880 |
| | | 720 | **0.479** | **43.55** | 0.470 | **0.350** | 68.07 | **0.304** | **0.194** | 210.73 | 0.885 | **0.249** | 140.31 | 0.864 |
| | | Avg | **0.444** | **27.00** | **0.516** | **0.249** | 37.93 | **0.351** | **0.163** | **125.45** | **0.900** | **0.233** | **85.44** | **0.886** |
| VCformer | original | 96 | 0.405 | 14.096 | 0.5496 | 0.186 | 17.41 | 0.346 | 0.152 | 66.03 | 0.892 | - | - | - |
| | | 192 | 0.455 | 21.57 | 0.528 | 0.238 | 28.12 | 0.339 | 0.170 | 99.79 | 0.896 | - | - | - |
| | | 336 | 0.530 | 30.89 | 0.502 | 0.288 | 41.53 | 0.336 | 0.186 | 138.81 | 0.890 | - | - | - |
| | | 720 | 0.561 | 47.28 | 0.4528 | 0.365 | 68.49 | 0.287 | 0.235 | 231.67 | 0.871 | - | - | - |
| | | Avg | 0.488 | 28.46 | 0.508 | 0.269 | 38.89 | 0.327 | 0.186 | 134.01 | 0.887 | - | - | - |
| | + DGL | 96 | 0.398 | 14.08 | 0.561 | 0.165 | **16.14** | 0.376 | 0.139 | 64.51 | 0.909 | - | - | - |
| | | 192 | 0.447 | 21.18 | 0.526 | **0.210** | 26.89 | **0.365** | 0.158 | 96.92 | 0.900 | - | - | - |
| | | 336 | **0.484** | 29.77 | 0.500 | 0.270 | 40.32 | 0.329 | 0.173 | 135.46 | 0.897 | - | - | - |
| | | 720 | **0.494** | 43.92 | 0.470 | 0.351 | 68.23 | 0.299 | 0.224 | 225.94 | 0.873 | - | - | - |
| | | Avg | 0.456 | **27.24** | 0.514 | **0.249** | 37.90 | **0.342** | 0.174 | 130.70 | 0.895 | - | - | - |
| | + CAL | 96 | **0.382** | **13.79** | **0.567** | **0.164** | 16.19 | **0.381** | **0.135** | **63.09** | **0.910** | - | - | - |
| | | 192 | **0.438** | 21.17 | **0.535** | 0.211 | **26.71** | 0.353 | **0.157** | 96.26 | **0.907** | - | - | - |
| | | 336 | 0.485 | 29.68 | **0.504** | 0.271 | 40.33 | 0.328 | **0.170** | 134.00 | 0.899 | - | - | - |
| | | 720 | 0.497 | 44.33 | **0.475** | 0.351 | **68.01** | 0.294 | **0.207** | 218.66 | 0.876 | - | - | - |
| | | Avg | **0.451** | 27.24 | **0.520** | **0.249** | 37.81 | 0.339 | **0.167** | 128.00 | 0.898 | - | - | - |

Table 14: Performance comparison of CGTFra and two variants without deep IVD modeling.

| Models | | | ETTm1 MSE | MAE | ETTm2 MSE | MAE | ETTh1 MSE | MAE | ETTh2 MSE | MAE | Exchange MSE | MAE | Weather MSE | MAE | ECL MSE | MAE | Solar MSE | MAE | Traffic MSE | MAE |
|---|---|---|---|---|---|---|---|---|---|---|---|---|---|---|---|---|---|---|---|---|
| CGTFra | original | 96 | 0.315 | 0.344 | **0.171** | **0.249** | 0.372 | **0.387** | 0.288 | 0.336 | 0.083 | 0.202 | 0.152 | 0.190 | 0.137 | 0.227 | 0.191 | 0.205 | 0.387 | 0.239 |
| | | 192 | **0.366** | **0.372** | **0.238** | **0.293** | 0.424 | 0.418 | 0.364 | 0.384 | 0.173 | 0.296 | 0.203 | 0.239 | 0.155 | 0.243 | 0.218 | 0.225 | 0.417 | 0.249 |
| | | 336 | **0.398** | **0.395** | **0.300** | **0.333** | 0.473 | 0.443 | 0.410 | 0.422 | 0.324 | 0.412 | 0.257 | 0.279 | 0.170 | 0.259 | 0.238 | 0.240 | 0.434 | 0.261 |
| | | 720 | **0.472** | **0.435** | **0.397** | **0.391** | 0.473 | 0.464 | 0.414 | 0.433 | 0.668 | 0.631 | 0.338 | 0.334 | 0.198 | 0.283 | 0.249 | 0.242 | 0.472 | 0.279 |
| | | Avg | **0.388** | **0.386** | **0.277** | **0.316** | 0.436 | 0.428 | 0.369 | 0.394 | 0.312 | 0.382 | 0.238 | 0.260 | 0.165 | 0.253 | 0.224 | 0.228 | 0.427 | 0.257 |
| CGTFra | shallow bias | 96 | 0.319 | 0.346 | 0.177 | 0.253 | 0.372 | 0.388 | 0.296 | 0.340 | 0.086 | 0.205 | 0.159 | 0.195 | 0.142 | 0.230 | 0.193 | 0.207 | 0.395 | 0.245 |
| | | 192 | 0.370 | 0.375 | 0.243 | 0.296 | 0.435 | 0.423 | 0.369 | 0.386 | 0.179 | 0.301 | 0.211 | 0.244 | 0.158 | 0.245 | 0.221 | 0.227 | 0.419 | 0.254 |
| | | 336 | 0.404 | 0.396 | 0.305 | 0.336 | 0.478 | 0.445 | 0.420 | 0.429 | 0.353 | 0.429 | 0.266 | 0.285 | **0.170** | **0.259** | 0.242 | 0.242 | 0.448 | 0.272 |
| | | 720 | 0.501 | 0.445 | 0.411 | 0.401 | 0.487 | 0.473 | 0.426 | 0.438 | 0.797 | 0.675 | 0.345 | 0.339 | 0.201 | 0.285 | 0.248 | **0.242** | 0.485 | 0.284 |
| | | Avg | 0.399 | 0.391 | 0.284 | 0.322 | 0.443 | 0.432 | 0.378 | 0.398 | 0.354 | 0.403 | 0.245 | 0.266 | 0.168 | 0.255 | 0.226 | 0.230 | 0.437 | 0.264 |
| CGTFra | shallow mask | 96 | **0.314** | **0.343** | 0.172 | 0.249 | **0.371** | **0.387** | 0.291 | 0.337 | 0.085 | 0.205 | 0.162 | 0.198 | 0.144 | 0.231 | 0.197 | 0.212 | 0.408 | 0.252 |
| | | 192 | 0.369 | 0.375 | **0.238** | **0.293** | 0.438 | 0.424 | **0.364** | **0.383** | 0.181 | 0.303 | 0.211 | 0.243 | 0.159 | 0.244 | 0.224 | 0.226 | 0.426 | 0.258 |
| | | 336 | 0.414 | 0.401 | 0.302 | 0.335 | 0.484 | 0.446 | 0.424 | 0.427 | 0.338 | 0.421 | 0.266 | 0.284 | 0.174 | 0.261 | 0.246 | 0.242 | 0.449 | 0.270 |
| | | 720 | 0.479 | 0.437 | 0.410 | 0.339 | 0.493 | 0.475 | 0.432 | 0.440 | 1.040 | 0.773 | 0.345 | 0.338 | 0.213 | 0.295 | **0.248** | **0.242** | 0.505 | 0.292 |
| | | Avg | 0.394 | 0.389 | 0.281 | 0.319 | 0.447 | 0.433 | 0.378 | 0.397 | 0.411 | 0.426 | 0.246 | 0.266 | 0.173 | 0.258 | 0.229 | 0.231 | 0.447 | 0.268 |

### A.15 FURTHER ANALYSIS OF CONSISTENT INTER-SERIES DEPENDENCY MODELING

To further investigate the necessity and effectiveness of modeling inter-variable dependencies in the deeper layers of the network, we conducted additional experiments on model modifications. Specifically, we removed the DGL and CAL modules from our CGTFra framework, retaining only the FMR module. Concurrently, we integrated the proposed dynamically constructed graph structure into the self-attention scores (a method similar to that used in DUET, see Figure 13) by using two fusion strategies: element-wise addition (acting as bias or guidance) and element-wise multiplication (acting as masking). The comparative results are presented in Table 14. The results indicate that merely guiding the self-attention mechanism with dynamic graph information is insufficient to achieve superior performance. We attribute this to the fact that this approach fails to model inter-variable dependencies in the deeper network layers, a limitation previously discussed in this paper. This finding implicitly underscores the necessity and effectiveness of consistently modeling inter-variable dependencies across both the shallow and deep layers of the network architecture.

### A.16 EFFECT OF HYPERPARAMETERS

To investigate the influence of hyperparameters on CGTFra's prediction performance, we conducted a series of experiments on CGTFra's stacking layers ($L$), the number of heads in the self-attention mechanism, the number of hops in DGL, the type of loss function used in CAL, and the consistency loss weight ($\lambda$) within the loss function. The results are presented in Figure 15. We present the following analysis: (1) **stacking layers ($L$)**: Stacking multiple layers in CGTFra enables the model to adapt to datasets of varying complexity, with the learning of multiple feature levels enhancing its representational capacity. Experimental results in ECL indicate that stacking multiple CGTFra layers improves performance for shorter prediction horizons (96 and 192), while showing an inverse, negative effect for longer horizons (336 and 720). (2) **the number of heads in the self-attention mechanism**: Multi-head attention allows the model to capture differentiated features and enhances parallelism. Concurrently, in our CGTFra, the number of heads influences the granularity of the alignment loss function's calculation. CGTFra achieves favorable performance gains when utilizing 4 or 8 heads. (3) **the number of hops in DGL**: While multi-hop propagation can achieve a larger global receptive field, it may also lead to negative effects such as oversmoothing, attenuation of node relevance, and amplified noise. In CGTFra, the default number of hops used is 2. We observe that as the number of hops increases, predicting excessively long sequences, such as those of length 720, exhibits significant performance fluctuations. (4) **the type of loss function used in CAL**: To investigate the impact of different similarity measures on CGTFra's performance and the effectiveness of the regularization term, we explore various loss functions as regularizers, including Kullback-Leibler (KL) divergence, Mean Absolute Error (MAE), Mean Squared Error (MSE), and Cosine Similarity. The results indicate that KL divergence, MAE, and MSE yield comparable performance, whereas Cosine Similarity leads to a significant performance degradation. This is likely attributable to Cosine Similarity's exclusive focus on vector direction, disregarding magnitude. Consequently, when evaluating the discrepancy between inter-variate dependencies captured by self-attention and GNNs, it merely promotes directional alignment without encouraging similar scales or absolute values for the tensors. Therefore, Cosine Similarity is unsuitable for quantifying inter-variate dependency differences between shallow and deep layers. (5) **the consistency loss weight ($\lambda$)**: In Equation 8, we use $\lambda$ to control the contribution of CAL. In Figure 15, different values of $\lambda$ most significantly impact performance for scenarios with a prediction length of 720.

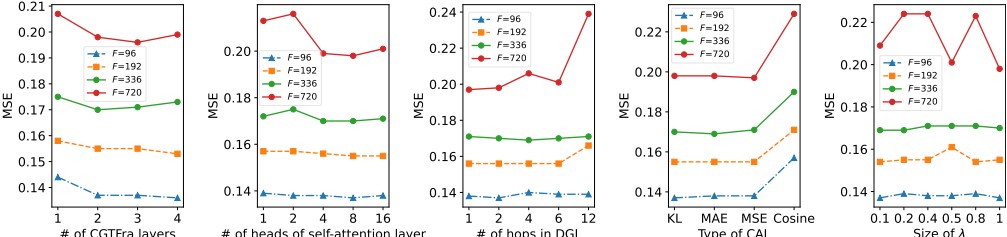

Figure 15: Sensitivity analysis of CGTFra's hyperparameters on ECL dataset for forecasting four future lengths $\{96, 192, 336, 720\}$ with fixed input length 96.

Table 15: Ablation studies on five diverse datasets.

| Part | FMR | DGL | CAL | $F$ | ETTm1 MSE | ETTm1 MAE | ETTh1 MSE | ETTh1 MAE | Weather MSE | Weather MAE | ECL MSE | ECL MAE | Traffic MSE | Traffic MAE |
|------|-----|-----|-----|-----|-----------|-----------|-----------|-----------|-------------|-------------|---------|---------|-------------|-------------|
| CGTFra | ✓ | ✓ | ✓ | 96 | **0.315** | **0.344** | 0.372 | 0.387 | **0.152** | **0.190** | **0.137** | **0.227** | **0.387** | **0.239** |
| | | | | 192 | **0.366** | **0.372** | **0.424** | **0.418** | **0.203** | **0.239** | **0.155** | **0.243** | 0.417 | **0.249** |
| | | | | 336 | **0.398** | **0.395** | **0.473** | **0.443** | **0.257** | **0.279** | **0.170** | **0.259** | **0.434** | **0.261** |
| | | | | 720 | 0.472 | 0.435 | **0.473** | 0.464 | **0.338** | **0.334** | **0.198** | **0.283** | 0.472 | 0.279 |
| aba1 | ✓ | ✗ | ✗ | 96 | 0.324 | 0.354 | 0.372 | 0.387 | 0.158 | 0.194 | 0.142 | 0.229 | 0.393 | 0.244 |
| | | | | 192 | 0.374 | 0.377 | 0.425 | 0.418 | 0.211 | 0.242 | 0.158 | 0.244 | **0.416** | 0.253 |
| | | | | 336 | 0.407 | 0.401 | 0.485 | 0.448 | 0.266 | 0.285 | 0.174 | 0.263 | 0.437 | 0.264 |
| | | | | 720 | 0.481 | 0.440 | 0.487 | 0.472 | 0.345 | 0.338 | 0.204 | 0.286 | 0.478 | 0.281 |
| aba2 | ✓ | ✓ | ✗ | 96 | **0.310** | 0.350 | 0.373 | **0.386** | 0.157 | 0.195 | 0.138 | 0.228 | 0.393 | 0.242 |
| | | | | 192 | 0.373 | 0.375 | 0.431 | 0.421 | 0.207 | 0.243 | 0.156 | 0.243 | 0.419 | 0.254 |
| | | | | 336 | 0.402 | 0.399 | **0.468** | **0.440** | 0.265 | 0.286 | 0.171 | 0.260 | 0.438 | 0.263 |
| | | | | 720 | **0.470** | **0.434** | 0.475 | **0.462** | 0.340 | 0.338 | 0.207 | 0.291 | **0.469** | **0.278** |
| aba3 | ✗ | ✓ | ✓ | 96 | 0.321 | 0.353 | **0.370** | **0.386** | 0.158 | 0.197 | 0.140 | 0.230 | 0.414 | 0.252 |
| | | | | 192 | 0.371 | 0.375 | 0.431 | 0.422 | 0.209 | 0.244 | 0.155 | 0.244 | 0.425 | **0.249** |
| | | | | 336 | 0.401 | 0.397 | 0.474 | 0.444 | 0.262 | 0.284 | 0.172 | 0.261 | 0.442 | 0.268 |
| | | | | 720 | 0.474 | 0.436 | **0.473** | 0.465 | 0.344 | 0.339 | 0.225 | 0.306 | 0.493 | 0.280 |

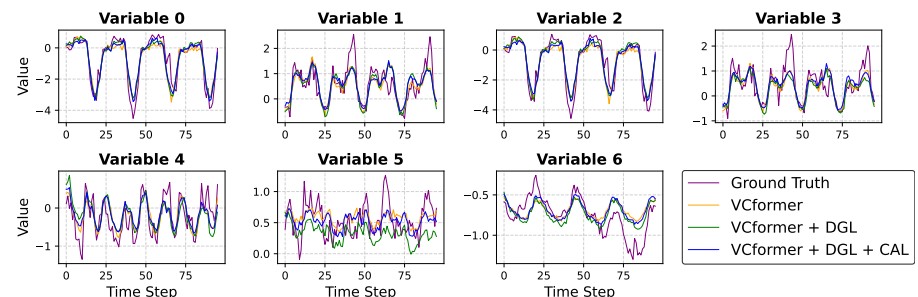

Figure 16: Prediction curves for VCformer and variates with our DGL and CAL on ETTh1 dataset.

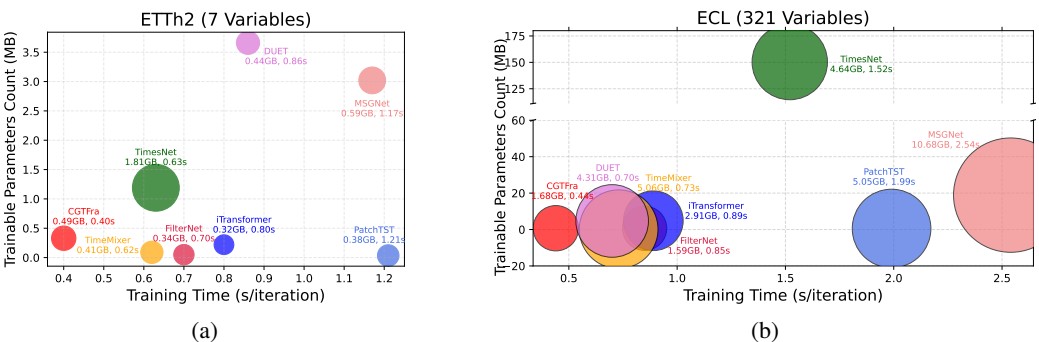

(a)           (b)

Figure 17: Computation effectiveness analysis for seven methods on ETTh2 and ECL. The size of the circle indicates the GPU memory footprint. For fair comparison, all batch sizes are set to 32.

A.17 ANALYSIS OF INTER-SERIES DEPENDENCY MODELING

To further evaluate the effectiveness of the DGL and CAL, as shown in Figure 16, we visualized the VCformer's actual prediction curves for 7 variables of ETTh1 in Figure 1. We observe that VC-former, when embedded with DGL and CAL, achieves superior prediction accuracy in most cases, indicating the efficacy of modeling IVD simultaneously in both shallow and deep network layers. Furthermore, we note that for variable 5, the introduction of DGL alone leads to worse prediction. However, with the consistency constraint of CAL, thanks to bidirectionally validated inter-variate dependencies, significantly improved prediction capabilities are obtained, demonstrating that the introduction of CAL effectively promotes the model's optimization of deep-layer feature embed-dings. Furthermore, comprehensive evaluation metrics are provided in Appendix A.14 (Table 13) to validate the effectiveness of DGL and CAL.

Figure 18 presents a comparison of prediction curves for four variables from the Weather dataset. CGTFra demonstrates superior trend forecasting performance compared to iTransformer and DUET, both of which are also capable of modeling inter-variable dependencies.

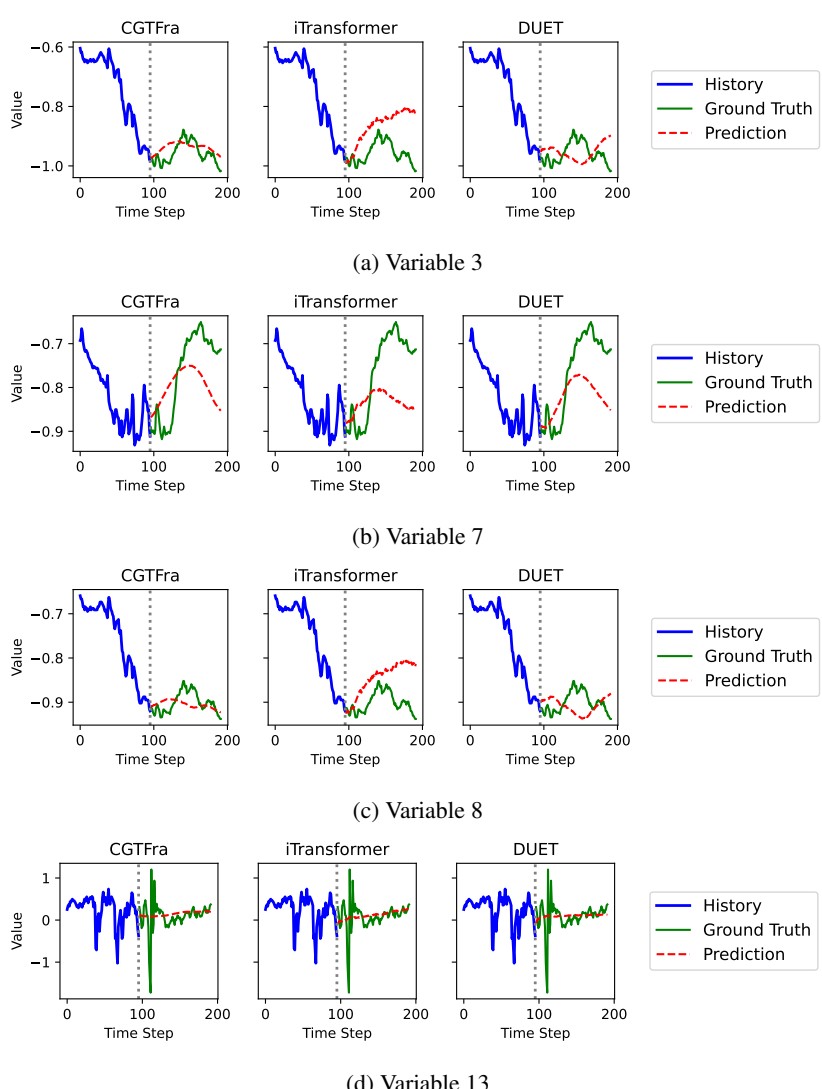

Figure 18: Actual prediction curves for three models capturing IVD on Variables 3, 7, 8, and 13 of Weather dataset.

## A.18 EFFICIENCY COMPARISON

We fairly compare the training time, running GPU memory, and trainable parameter count against 7 sota methods in Figure 17. Benefiting from the computational efficiency of DGL in capturing variable dependencies and the performance gains of CAL without introducing additional learnable parameters, CGTFra achieves strong performance and computational efficiency with relatively less trainbale parameters. Compared to another sota method-DUET (Qiu et al., 2025), known for its high run-time efficiency, CGTFra reduces GPU memory usage by 61% and demonstrates a training speed improvement of approximately 42.86% on the complex ECL dataset, indicating the high effectiveness and efficiency of CGTFra.

## A.19 LIMITATIONS

Although our study significantly enhances the performance of existing studies by introducing deep inter-variate dependency modeling (DGL) within the Variable Transformer and further optimizing inter-variate associations across both deep and shallow layers through explicit dependency constraints (CAL), we still observe that Variable Transformers incorporating DGL and CAL, such as DUET, and iTransformer—exhibit limited improvements or even performance degradation on datasets like Solar and Traffic (see Table 2 and Table 12). We posit that there are two primary reasons for the limited performance improvement, and in some cases degradation, of our proposed DGL and CAL on datasets with a very large number of variables.

Primarily, as the number of variables ($N$) increases, **the probability of spurious correlations between any two variables rises dramatically**. The self-attention mechanism, designed to find relationships within an $N \times N$ matrix, is compelled to assign attention weights across all variable pairs. **In such a high-dimensional space, these weights are more likely to reflect coincidental noise within a sample rather than genuine, stable dependencies.** Consequently, when CAL is applied, it forces the adjacency matrix $A$ learned by DGL to align with this noisy Correlation Map (MCM), **effectively instigating negative knowledge transfer instead of beneficial regularization**. The GNN is thus coerced into encoding numerous useless or even erroneous connections in its graph structure, which undermines its ability to perform effective information propagation. This can lead to performance that is even worse than that of a simple FFN, which at least makes a harmless "variable independence" assumption.

Furthermore, the self-attention mechanism, particularly after the softmax operation, naturally produces a dense attention map. **This inherent density creates a significant discrepancy with the potentially sparse nature of the adjacency matrix learned by the GNN** (see Figure 6(a)), thereby posing a fundamental challenge to the alignment process.

We will improve upon this in future work by proposing a more general method for modeling correlation constraint between deep and shallow layers.

