# OpenReview forum: "CGTFra: General Graph Transformer Framework for Consistent Inter-series Dependency Modeling in Multivariate Time Series"
_ICLR.cc/2026/Conference — Submitted to ICLR 2026_

### Official Review · Reviewer_xnWj · 2025-10-29

**Soundness:** 2
**Presentation:** 2
**Contribution:** 2
**Rating:** 2
**Confidence:** 4

**Summary:**

This paper proposes CGTFra, a General Graph Transformer Framework for multivariate time series forecasting. The core idea is to model both the temporal dependencies and inter-variable correlations using a unified graph-transformer paradigm. Experiments on several real-world datasets demonstrate that CGTFra achieves consistent improvements over state-of-the-art baselines.

**Strengths:**

1. This paper offers a new perspective on modeling inter-variable dependencies (IVD), providing fresh insights that could inspire future research in multivariate time series forecasting.

2. The experimental section compares CGTFra with a wide range of strong baselines, and the results consistently show performance gains on multiple benchmark datasets.

**Weaknesses:**

1. The paper incorrectly states that iTransformer employs positional encoding. In fact, iTransformer does not use positional encodings, as it models temporal order implicitly within feature embeddings rather than through token positions.

2. The paper also misinterprets the role of Feed-Forward Networks (FFNs) in the Transformer architecture. FFNs do not explicitly model intra-series dependencies; instead, they act as nonlinear mappings that refine representations after the attention operation, capturing temporal relationships between the input and predicted future values.

**Questions:**

1. In some datasets (e.g., ETT), the variables appear to be largely independent. Why is the Inter-Variable Dependency (IVD) module necessary in such cases?

2. What are the specific experimental settings used to generate the results in Figure 2?

3. Can the FMR be combined with architectures other than Transformers? In other words, is FMR a model-agnostic module or specifically tailored to Transformer-based frameworks?

---

> ### Author Response · Authors · 2025-11-17
> **Response to Reviewer xnWj (Part 1)**
>
> Thank you for your feedback. In response to your suggestions, we have made point-by-point revisions.
>
> **Q1: iTransformer employing timestamps information**: Thanks for your feedback regarding the presentation of details in our manuscript.
>
> Please allow us to offer an additional explanation for our initial choice of terminology: Our reasoning traces back to the original Transformer paper [1], which states, "in order for the model to make use of **the order of the sequence**, **we must inject some information about the relative or absolute position of the tokens in the sequence**." From this, we derived our definition of positional encoding as **a method for preserving the relative or absolute position within a sequence**.
>
> Unlike in Natural Language Processing (NLP) tasks where each word corresponds to a token vector, in the time-series forecasting domain, each time step corresponds to a numerical point. In MTSF, each timestamp can reflect the global positional information of its corresponding numerical point (across the training, validation, and test sets). We therefore considered these timestamps to correspond to the “relative or absolute position” described in the Transformer paper. Furthermore, this approach is conceptually consistent with **learnable positional encodings**, which is why our manuscript initially adopted this term.
>
> We provide the above explanation to clarify the motivation behind our original phrasing. To avoid any potential confusion, we have revised the descriptions of “positional encoding” throughout the manuscript, **replacing the term with "timestamp information" or “timestamp encoding”**.
>
> **Q2: FFN model intra-series dependencies**: Thank you for your further comments on the details of **Figure 3 (Line 118)**. We fully understand your perspective on this matter. Please allow us to elaborate on our reasoning step by step.
>
>  (To begin with, we must emphasize that in our implementation, the input to the FFN has a dimension of $\mathbf{X} \in \mathbb{R}^{N \times L}$, where the FFN operates along the $L$ dimension. Crucially, $L$ represents the temporal dimension of the features. This is significantly different from embedding method of vanilla Transformer.)
>
> First, as defined in our manuscript, **inter-series** dependency refers to the dependencies among **different variables**, whereas **intra-series** dependency pertains to the **internal dynamics within a single time series**. To elaborate further, a Feed-Forward Network (FFN) in a Transformer consists of two linear layers with an activation function. Existing research supports our statement of FFNs as modeling intra-series dependencies.
>
> For instance, RLinear [2] and iTransformer [3] state the following (we quote directly from the original papers):
> “linear mapping can effectively capture **periodic features** in time series” [2]
> and
>
> “Recent revisiting on linear forecasters highlights that temporal features extracted by MLPs are supposed to be shared within distinct time series. We propose a rational explanation that the neurons of MLP are taught to **portray the intrinsic properties** of any time series, such as the **amplitude**, **periodicity**, and even **frequency spectrums** (neuron as a filter), serving as a more advantageous predictive **representation learner** than the self-attention applied on time points.” [3]
>
> The concepts mentioned in these studies, such as “periodic features,” “emporal features,” and “intrinsic properties,” are fundamentally temporal characteristics within an individual series. Therefore, we categorize the modeling of such features under the umbrella of intra-series dependency.
>
> Furthermore, since our framework typically stacks multiple layers of self-attention and FFNs, the FFNs in the hidden layers are equivalent to a 1x1 convolutional layer, which primarily serves to extract features along the temporal dimension $L$.
> We can illustrate this with an analogy from classification tasks: **the intermediate hidden layers learn the latent features** of input (in our case, each time series), while the final projection head acts as the classification layer. Similarly, for the task of time series forecasting, we believe that the **final projection head** is primarily responsible for what you referred to as “capturing temporal relationships between the input and predicted future values.”
>
> Ultimately, the main point we wish to emphasize in our paper is that the FFN module does **not** model **inter**-series dependencies (**a point on which we believe we are in agreement**). This leads to an inconsistency in how the overall architecture models such dependencies: they are modeled only in the shallower layers of the network.
>
> **To avoid any ambiguity, we have revised Figure 3 and replaced the term “Intra-series modeling” with “Non-inter-series modeling.”**
> We hope this explanation clarifies our rationale. We welcome any further discussion and would be happy to elaborate on any of these points.

---

> ### Author Response · Authors · 2025-11-17
> **Response to Reviewer xnWj (Part 2)**
>
> **Q3: The Necessity of IVD Modeling**: We fully understand your concerns. In many real-world, complex scenarios, datasets may exhibit insignificant or even non-existent dependencies among variables. **The method proposed in this paper is specifically designed to address such situations**.
>
> Our proposed DGL and CAL modules can adaptively model Inter-Variate Dependencies (IVD), effectively capturing **strong** IVD from the given multivariate signals while disregarding **weaker** ones.
>
> As illustrated in **Figure 1**, **Figure 9**, and **Figure 5**, **Figure 6** of the manuscript, we depict the dependency structures captured from the ETTh1 and Weather datasets. The learned dependency matrices are observed to be **sparse**. **This indicates that when the association between variables is weak, their corresponding weights are minimal** (for instance, the coordinate (6, 4) in **Figure 9**).
>
> Furthermore, **Figure 10(b) (Line 850)** demonstrates the complex dependency relationships among the 21 variables in Weather dataset (including the situation either very weak IVDs or no discernible relationships). **Figure 6** reveals a crucial insight: for the four variables with strong associations (variables 3, 7, 8, and 13), **their inter-dependencies are challenging to model reasonably after the introduction of DGL alone, which is also a limitation of existing methods** (please see the analysis of **Line 493-Line 515**). However, **with the subsequent integration of CAL, this limitation is significantly mitigated.** This demonstrates that our CGTFra can effectively model these complex dependencies in complex dataset.
>
> Furthermore, this performance enhancement is **generally applicable to existing Variate Transformers**, including iTransformer, DUET, and VCformer, **underscoring the necessity of explicitly modeling deep IVD**.
>
> **Q4: Experimental settings in Figure 2**: We thank you for raising the important concern, and we agree on the critical importance of ensuring fair comparisons in scientific research. To ensure a fair comparison, the hyperparameter configurations and all other settings used in **Figure 2** are identical to the publicly available experimental setup of iTransformer. Furthermore, all ablation studies were conducted using the exact same hyperparameter settings (from their publicly source code) and under the same software and hardware environments.
>
> We have added a corresponding clarification to the caption of **Table 4** (see **Line 810**).
>
> **Q5: The generalization of FMR**: Thank you for raising this concern. To validate the effectiveness of FMR, we conducted experiments on three sota methods. These include two Transformer-based models, iTransformer and VCformer, and one non-Transformer model, CASA [4].
>
> After an extensive literature search, we have expanded our empirical evaluation to include FilterNet [5], a non-Transformer model that features two effective frequency filtering techniques. And if we find any other non-Transformers that employ a upsampling directly on the (series-aware not patch-aware) input, we will update our comparisons.
> | Model | Variant | Horizon | ETTm1 MSE | ETTm1 MAE | ETTh1 MSE | ETTh1 MAE | Weather MSE | Weather MAE | ECL MSE | ECL MAE | Traffic MSE | Traffic MAE |
> | :--- | :--- | :---: | :---: | :---: | :---: | :---: | :---: | :---: | :---: | :---: | :---: | :---: |
> | FilterNet | original | 96 | **0.317** | **0.357** | 0.381 | 0.399 | 0.164 | 0.210 | 0.147 | 0.242 | 0.431 | 0.295 |
> | | | 192 | **0.364** | 0.384 |  0.440 | 0.428 | 0.214 | 0.256 | 0.162 | 0.254 | 0.448 | 0.298 |
> | | | 336 | 0.396 | 0.407 | **0.487** | **0.451** | 0.273 | 0.299 | **0.177** | **0.272** | 0.465 | 0.303 |
> | | | 720 | 0.457 | 0.444 | **0.494** | **0.471** | 0.359 | 0.353 | **0.228** | **0.318** | 0.497 | 0.320 |
> | | | **Avg** | 0.384 | **0.398** | 0.451 | **0.437** | 0.253 | 0.280 | 0.179 | 0.272 | 0.460 | 0.304 |
> | FilterNet | + FMR | 96 | 0.318 | 0.359 | **0.376** | **0.397** | **0.160** | **0.206** | **0.144** | **0.239** | **0.422** | **0.289** |
> | | | 192 | **0.364** | **0.383** | **0.438** | **0.427** | **0.209** | **0.252** | **0.159** | **0.252** | **0.445** | **0.295** |
> | | | 336 | **0.395** | **0.406** | 0.489 | **0.451** | **0.270** | **0.296** | **0.177** | 0.273 | **0.462** | **0.301** |
> | | | 720 | **0.455** | **0.442** | 0.496 | 0.472 | **0.353** | **0.350** | **0.228** | 0.321 | **0.494** | **0.317** |
> | | | **Avg** | **0.383** | **0.398** | **0.450** | **0.437** | **0.248** | **0.276** | **0.177** | **0.271** | **0.455** | **0.300** |
>
> Given that FilterNet already establishes a strong sota baseline on the ETT datasets, the fact that integrating FMR still yields performance improvements to a certain extent, even if marginal, underscores the potential effectiveness of FMR. We have incorporated these results into the revised manuscript (see **Table 2** and **Table 11**). Additionally, **the code to reproduce these results has been updated to anonymized repository**.

---

> ### Author Response · Authors · 2025-11-17
> **Response to Reviewer xnWj (Part 3)**
>
> (11/20 updated: We introduced an additional non-Transformer method, **TSMixer** [6], to validate the effectiveness of FMR. The code for reproducing these results has been uploaded to our anonymous GitHub repository. Due to limited space in **Table 11**, the following results are not included in the manuscript.)
> | Model | Variant | Horizon | ETTm1 MSE | ETTm1 MAE | ETTh1 MSE | ETTh1 MAE | Weather MSE | Weather MAE | ECL MSE | ECL MAE | Traffic MSE | Traffic MAE |
> | :--- | :--- | :---: | :---: | :---: | :---: | :---: | :---: | :---: | :---: | :---: | :---: | :---: |
> | TSMixer | original | 96 | 0.486 | 0.472 | 0.484 | 0.494 | 0.182 | 0.253 | 0.209 | 0.311 | 0.538 | 0.372 |
> | | | 192 | 0.479 |0.480 | **0.569** | 0.547 | 0.219 | 0.288 | 0.219 | 0.329 | 0.546 | 0.375 |
> | | | 336 | 0.527 | 0.516 | **0.669** | **0.609** | 0.264 | 0.324 | 0.240 | 0.349 | 0.568 | 0.384 |
> | | | 720 | 0.599 | 0.558 | **0.724** | **0.658** | 0.320 | 0.365 | 0.272 | 0.3726 | 0.611 |0.412 |
> | | | **Avg** | 0.523 |0.506 | **0.6115** | **0.577** | 0.246 | 0.307 | 0.235 | 0.340 | 0.566 | 0.386 |
> | TSMixer | + FMR | 96 | **0.460** | **0.458** | **0.483** | **0.492** | **0.177** | **0.249** | **0.206** | **0.307** | **0.528** | **0.356** |
> | | | 192 | **0.473** | **0.477** | 0.571 | **0.546** | **0.217** | **0.286** | **0.213** | **0.322** | **0.537** | **0.367** |
> | | | 336 | **0.526** | **0.516** | 0.675 | 0.611 | **0.262** | **0.322** | **0.234** | **0.343** | **0.561** | **0.380** |
> | | | 720 | **0.595** | **0.557** | 0.750 | 0.669 | **0.319** | **0.363** | **0.268** | **0.368** | **0.598** | **0.403** |
> | | | **Avg** | **0.513** | **0.502** | 0.620 | 0.579 | **0.248** | **0.305** | **0.230** | **0.335** | **0.556** | **0.376** |
>
> We thank you for your questions and suggestions regarding the presentation of our manuscript, the effectiveness of the proposed FMR, and its generalizability. In response to your comments, we have updated the manuscript’s statement and figures. Furthermore, we have included two additional non-Transformer methods to further validate the effectiveness of FMR.
>
> We hope that our point-by-point responses will address the suggestions you have raised.
>
> If there are any further comments from your side, we will happy to address them before the rebuttal period ends.
>
> [1] Attention is all you need. Advances in neural information processing systems. (2017).
>
> [2] Revisiting long-term time series forecasting: An investigation on linear mapping. arXiv preprint arXiv:2305.10721 (2023).
>
> [3] iTransformer: Inverted Transformers Are Effective for Time Series Forecasting. The Twelfth International Conference on Learning Representations. (2024)
>
> [4] Casa: Cnn autoencoder-based score attention for efficient multivariate long-term time-series forecasting, 2025. URL https://arxiv.org/abs/2505.02011.
>
> [5] Filternet: Harnessing frequency filters for time series forecasting. Advances in Neural Information Processing Systems. (2024)
>
> [6] Tsmixer: Lightweight mlp-mixer model for multivariate time series forecasting. ACM SIGKDD (2023)

---

### Official Review · Reviewer_bgj4 · 2025-10-31

**Soundness:** 3
**Presentation:** 2
**Contribution:** 3
**Rating:** 4
**Confidence:** 4

**Summary:**

This paper proposes a Graph Transformer framework CGTFra with adaptive frequency masking and resampling method and dynamic graph learning framework, which can diminish the importance of timestamps and promote consistent inter-variate dependencies (IVD) modeling. Experiments demonstrate the effectiveness of CGTFra.

**Strengths:**

1.Integrating the dynamic graph learning and consistency alignment loss to promote the modeling of consistent IVD is interesting.

2.The experiments are presented in considerable detail.

**Weaknesses:**

1.The dynamic graph learning lacks innovation and appears to be a standard adaptive graph construction method, which is widely explored in prior works. The authors should clarify the relationship between CGTFra and these works to further emphasize their own contribution.

2.The claim that IVD are modeled exclusively in shallow layers (on line 104) is unconvincing, as stacking multiple attention layers is a straightforward way to model IVD in deep layers. This oversight weakens the motivation for promoting consistent IVD modeling. The authors should clarify the oversight further to avoid misunderstanding.

3.Some visual comparisons should be quantified, as the claimed improvements are often subtle and difficult to assess from the plots alone, e.g., the claims on lines 143, 170, and 375. These claims should be backed by quantitative data, e.g., percentage gains, to make the comparisons clear.

4.The paper has some weaknesses in the experiments, which are not convincing enough:

(1)Considering CGTFra is a graph Transformer framework, more GNN-based models and even hypergraph-based models should be compared to further validate the effectiveness of CGTFra, e.g., Ada-MSHyper [1] and MTSF-DG [2].

(2)There are some overstatements and factual errors in the experimental analysis. For example, the claim that CGTFra consistently exhibits enhanced performance on ETT and solar datasets (on line 360) seems to be an overstatement. According to Table 1, CGTFra is actually outperformed by FilterNet [3] on both ETTm1 and ETTm2 datasets in terms of MSE. The claim that introducing DGL results in performance degradation on ECL (on line 438) seems to conflict with the results of Table 4. The authors should thoroughly review the analysis.

(3)There seem to be several inconsistencies in the bolding of results in Tables 2 and 3. For example, “iInformer + Solar” in Table 3, the best results of MAE are not bolded. The authors should carefully check all tables to avoid these mistakes.

[1]Shang Z, Chen L, Wu B, et al. Ada-MSHyper: Adaptive multi-scale hypergraph Transformer for time series forecasting. NIPS 2024.
[2]Zhao K, Guo C, Cheng Y, et al. Multiple time series forecasting with dynamic graph modeling. VLDB 2024.
[3]Yi K, Fei J, Zhang Q, et al. FilterNet: Harnessing frequency filters for time series forecasting. NIPS 2024.

**Questions:**

1.Some notations and formulas are confusing. For example, in the definition of MTS on line 263, f(t) often represents values at time t, why define f(t) as a 2D matrix? On Formula 1, the l in the summation is missing. On Formulas 5 and 6, are these trainable parameters the same? If not, why use the same notations? The Formula 7 seems incorrect for producing the described MCM on line 319, as the concat operation is missing. On Formula 8, the mathematical definition of KL divergence should be used instead of the engineering-style. Why calculate the KL divergence between P and Q instead of Q and P?

---

> ### Author Response · Authors · 2025-11-17
> **Response to Reviewer bgj4 (part 1)**
>
> We sincerely thank you for your careful review and highly valuable feedback. We also appreciate your interest in our proposed method. We are pleased to have this opportunity to discuss our work further with you.
>
> **Q1: Emphasize contribution**: We thank you for the suggestion to clarify the connection between dynamic graph learning (including graph construction) and CGTFra, thereby better highlighting our contributions. We agree that the construction of adaptive graph structures is indeed a relatively mature research area. Our core innovation is twofold: first, **we address the deep IVD information loss and optimization difficulties in existing Variate Transformers**. Second, building on this, we explore the **consistency of IVD modeling between shallow (self-attention) and deep (GNN) layers** (based on information bottleneck principle, see **Appendix A.9 (Line 1007)**). Consequently, our approach to graph construction differs fundamentally from existing methods. Unlike GL-STGTN [1] or other similar methods with graph learning, which uses the **raw input $X$**, or methods like Sageformer [2] and MSGNet [3] that rely **solely on self-learned node embeddings**, our graph constructor is uniquely informed by a combination of **outputs $X^{sa}$ from the self-attention layer** and **learnable node embeddings**.
>
> Importantly, **to the best of our knowledge, we are the first to comprehensively analyze the connections and differences** between self-attention (within Variate Transformers) and GNNs for modeling IVD. **We believe that the comprehensive analysis and proposed DGL and CAL will provide valuable insights and an effective reference for future research in Graph Transformer-based methods.**
>
> Furthermore, as we emphasize in the paper, we have compactly integrated the two linear layers of the original FFN into our DGL module. The first linear layer is utilized to aggregate information from multi-hop neighbors, while the second is used for further temporal feature extraction. It is this compact architecture that **allows our DGL and CAL to serve as a general-purpose IVD modeling method for the deeper layers of Variate Transformers**—a level of universality **not achieved by existing dynamic graph learning techniques**.
>
> Crucially, our choice of a simple graph construction method (e.g., **Equation (5)**, with only two linear layers) ensures that the DGL module maintains high computational efficiency **when integrated into existing other frameworks**. We have validated this assertion within iTransformer. The introduction of DGL significantly accelerates the convergence of both training and validation losses, achieving an 8.78% reduction in MSE (see **Figure 8 (Line 526)**). This substantial performance gain is achieved with **only a minimal computational overhead**: time complexity of **$\mathcal{O}(N(D+nd+D*nd))$** (introduced by the dynamic graph construction (see **Equation (5)**)), which is linear with respect to the number of variables $N$, where $nd$ is a small hyperparameter (e.g., 8, 10, or 32), and $D$ is the hidden dimension. **We believe the introduced performance and efficiency is promising**.
>
> We have updated the corresponding statements in the manuscript (please see **Appendix A.10** (begin with **Line 1069**).
>
> **Q2: Stacking multiple attention layers to model IVD**: We thank the reviewer for this intuitive insight. As you correctly pointed out, stacking multiple layers is a direct approach to modeling IVD. In fact, we briefly discussed a related finding in our manuscript (**Line 957**): the original iTransformer paper experimented with using self-attention in both shallow and deep layers to model IVD but obtained very poor prediction results, suggesting that merely stacking self-attention layers is insufficient for effectively capturing these dependencies.

---

> ### Author Response · Authors · 2025-11-17
> **Response to Reviewer bgj4 (part 2)**
>
> To further substantiate this argument, we have conducted additional experiments. We stacked a varying number of self-attention layers in our CGTFra framework while keeping the FFN component unchanged (for CGTFra, this means removing the DGL and CAL modules and replacing DGL with an FFN). The results are presented in the table below (**S2** denotes that a 2-layer self-attention is stacked, **S3** denotes 3 layers):
> | Dataset | Horizon | Ours MSE | Ours MAE | S2 MSE | S2 MAE | S3 MSE | S3 MAE |
> | :--- | :---: | :---: | :---: | :---: | :---: | :---: | :---: |
> | **ETTm1** | 96 | **0.315** | **0.344** | 0.336 | 0.362 | 0.339 | 0.363 |
> | | 192 | **0.366** | **0.372** | 0.375 | 0.384 | 0.382 | 0.392 |
> | | 336 | **0.398** | **0.395** | 0.421 | 0.413 | 0.427 | 0.416 |
> | | 720 | **0.472** | **0.435** | 0.484 | 0.440 | 0.479 | 0.438 |
> | | **AVG** | **0.388** | **0.386** | 0.404 | 0.400 | 0.407 | 0.402 |
> | **ETTh1** | 96 | **0.372** | **0.387** | 0.379 | 0.392 | 0.392 | 0.401 |
> | | 192 | **0.424** | **0.418** | 0.433 | 0.424 | 0.434 | 0.424 |
> | | 336 | **0.473** | **0.443** | 0.481 | 0.449 | 0.485 | 0.451 |
> | | 720 | **0.473** | **0.464** | 0.488 | 0.476 | 0.510 | 0.486 |
> | | **AVG** | **0.436** | **0.428** | 0.445 | 0.435 | 0.455 | 0.441 |
> | **Weather** | 96 | **0.152** | **0.190** | 0.164 | 0.198 | 0.173 | 0.206 |
> | | 192 | **0.203** | **0.239** | 0.221 | 0.252 | 0.229 | 0.255 |
> | | 336 | **0.257** | **0.279** | 0.276 | 0.294 | 0.280 | 0.297 |
> | | 720 | **0.338** | **0.334** | 0.352 | 0.345 | 0.365 | 0.351 |
> | | **AVG** | **0.238** | **0.260** | 0.253 | 0.272 | 0.262 | 0.277 |
> | **ECL** | 96 | **0.137** | **0.227** | 0.145 | 0.232 | 0.151 | 0.237 |
> | | 192 | **0.155** | **0.243** | 0.162 | 0.246 | 0.175 | 0.252 |
> | | 336 | **0.170** | **0.259** | 0.177 | 0.263 | 0.180 | 0.264 |
> | | 720 | **0.198** | **0.283** | 0.206 | 0.290 | 0.215 | 0.298 |
> | | **AVG** | **0.165** | **0.253** | 0.172 | 0.258 | 0.180 | 0.263 |
>
> The results confirm that stacking multiple self-attention layers does not effectively model inter-variable dependencies.
> Therefore, **the strategy adopted in our work is to treat the self-attention layer and the proposed DGL as an integrated block**. A deep network is then constructed by stacking this entire block for a certain number of layers (**$\times$ L** in **Figure 4**). Within this holistic block, the issue of inconsistent dependency modeling between shallow and deep layers, as described in the manuscript, naturally arises.
>
> **Q3: Quantifying visual comparisons**: We sincerely thank the reviewer for the insightful suggestion to quantify our performance claims. We agree that replacing qualitative descriptions like “significant improvement” with precise, quantitative metrics greatly enhances the scientific rigor of our manuscript. Therefore, we have made corresponding changes, noted in **Line 136**, **Line 145**, and **Line 373**.
>
> **Q4: Compare with other graph Transformer methods**：We sincerely thank the reviewer for this valuable suggestion. We agree that a broader comparison with other Graph Transformer-based methods is crucial for providing a more comprehensive evaluation of our framework.
> Following this advice, we have conducted additional experiments comparing our model against **Ada-MSHyper** [4], and **Sageformer** [2] (All experiments were conducted under the same hardware and software environment, using the hyperparameters reported in their officially released code). The results have been integrated into the revised manuscript in **Table 10 (Line 1226)**.

---

> ### Author Response · Authors · 2025-11-17
> **Response to Reviewer bgj4 (part 3)**
>
> | Model | Horizon | ETTm1 MSE | ETTm1 MAE | ETTm2 MSE | ETTm2 MAE | ETTh1 MSE | ETTh1 MAE | ETTh2 MSE | ETTh2 MAE | Exchange MSE | Exchange MAE | Weather MSE | Weather MAE | ECL MSE | ECL MAE | Traffic MSE | Traffic MAE |
> | :--- | :---: | :---: | :---: | :---: | :---: | :---: | :---: | :---: | :---: | :---: | :---: | :---: | :---: | :---: | :---: | :---: | :---: |
> | **CGTFra** | 96 | 0.315 | **0.344** | **0.171** | **0.249** | 0.372 | **0.387** | **0.288** | **0.336** | 0.083 | 0.202 | **0.152** | **0.190** | **0.137** | **0.227** | **0.387** | **0.239** |
> | | 192 | 0.366 | **0.372** | 0.238 | **0.293** | **0.424** | **0.418** | **0.364** | **0.384** | **0.173** | **0.296** | **0.203** | **0.239** | **0.155** | **0.243** | **0.417** | **0.249** |
> | | 336 | 0.398 | **0.395** | 0.300 | **0.333** | 0.473 | **0.443** | **0.410** | **0.422** | **0.324** | **0.412** | **0.257** | **0.279** | **0.170** | **0.259** | **0.434** | **0.261** |
> | | 720 | 0.472 | **0.435** | 0.397 | **0.391** | 0.473 | **0.464** | **0.414** | **0.433** | **0.668** | **0.619** | **0.338** | **0.334** | **0.198** | **0.283** | 0.472 | **0.279** |
> | | **Avg** | 0.388 | **0.386** | 0.277 | **0.316** | **0.436** | **0.428** | **0.369** | **0.394** | **0.312** | **0.382** | **0.238** | **0.260** | **0.165** | **0.253** | **0.427** | **0.257** |
> | **Ada-MSHyper** | 96 | **0.309** | 0.357 | 0.173 | 0.261 | 0.376 | 0.395 | 0.291 | 0.338 | - | - | 0.161 | 0.202 | 0.144 | 0.241 | 0.405 | 0.263 |
> | | 192 | **0.362** | 0.385 | **0.235** | 0.307 | 0.436 | **0.418** | 0.370 | 0.389 | - | - | 0.209 | 0.248 | 0.160 | 0.247 | 0.419 | 0.275 |
> | | 336 | **0.394** | 0.409 | **0.295** | 0.340 | 0.468 | 0.447 | 0.426 | 0.434 | - | - | 0.263 | 0.289 | 0.176 | 0.273 | 0.439 | 0.278 |
> | | 720 | **0.461** | 0.447 | **0.389** | 0.402 | **0.469** | 0.472 | 0.418 | 0.439 | - | - | 0.349 | 0.346 | 0.212 | 0.293 | **0.467** | 0.299 |
> | | **Avg** | **0.382** | 0.400 | **0.273** | 0.328 | 0.437 | 0.433 | 0.376 | 0.400 | - | - | 0.246 | 0.271 | 0.173 | 0.264 | 0.433 | 0.279 |
> | **Sageformer** | 96 | 0.333 | 0.366 | 0.175 | 0.259 | 0.377 | 0.394 | 0.291 | 0.339 | **0.082** | **0.201** | 0.165 | 0.207 | 0.148 | 0.246 | NaN | NaN |
> | | 192 | 0.371 | 0.389 | 0.241 | 0.301 | 0.428 | 0.426 | 0.376 | 0.394 | 0.177 | 0.299 | 0.211 | 0.251 | 0.163 | 0.248 | NaN | NaN |
> | | 336 | 0.406 | 0.409 | 0.302 | 0.341 | **0.466** | 0.448 | 0.417 | 0.428 | 0.333 | 0.418 | 0.269 | 0.292 | 0.181 | 0.265 | NaN | NaN |
> | | 720 | 0.478 | 0.449 | 0.399 | 0.396 | 0.487 | 0.476 | 0.422 | 0.441 | 0.866 | 0.702 | 0.347 | 0.345 | 0.209 | 0.306 | NaN | NaN |
> | | **Avg** | 0.397 | 0.403 | 0.279 | 0.324 | 0.440 | 0.436 | 0.377 | 0.401 | 0.365 | 0.405 | 0.248 | 0.274 | 0.175 | 0.266 | - | - |
>
> These new comparisons further demonstrate the effectiveness and superiority of our approach within the landscape of Graph Transformer architectures. On the Weather dataset, CGTFra reduces the MSE by **4.2%** and **3.36%** compared to Sageformer and Ada-MSHyper, respectively. Similarly, on the ECL dataset, it achieves reductions of **6.06%** and **4.85%**, respectively.
>
> Note: Regarding MTSF-DG [5], we were indeed aware of this work prior to our submission. However, we noted that its evaluation was exclusively conducted on datasets from the traffic and energy domains (i.e., METR-LA, PEMS-BAY, PEMS04, PEMS08, Solar-Energy, Electricity), which differ from the broader set of benchmarks commonly used in mainstream MTSF research.
> We initially intended to reproduce their results on the datasets common to both our studies (Solar-Energy, Electricity, PEMS04, and PEMS08) for a direct comparison. Unfortunately, the **detailed hyperparameters** for these four specific datasets **were not provided** in their paper or public repository. We attempted to perform a hyperparameter search ourselves; however, **due to the computational cost and runtime of their model, conducting a fair and comprehensive comparison within a reasonable timeframe proved to be infeasible**.
>
> Consequently, to ensure a robust and fair evaluation against a relevant Graph Transformer-based method, we opted to compare our model with Sageformer instead.
>
> **Q5: Presentation of results and the bold values**: We sincerely thank you for your valuable suggestions on clarifying our result descriptions and ensuring consistent use of bold formatting. This has significantly enhanced the rigor of our manuscript.
>
> We have re-examined our analysis of the results and have made point-by-point revisions according to the reviewer’s comments (see **Line 340**, **Line 407**). Furthermore, we have carefully checked the bolding in all tables to ensure a consistent and meaningful representation of the best-performing results (see blue text in **Table 2**, **Table 9**, **Table 11**, and **Table 12**).

---

> ### Author Response · Authors · 2025-11-17
> **Response to Reviewer bgj4 (part 4)**
>
> **Q6: KL divergence**: We appreciate your valuable comment as for the KL divergence. We apologize for any confusion this may have caused. We have revised the formula's presentation to adhere to a clearer and more standard mathematical formalism. Please allow us to clarify the **Formula 7** in detail: The **Formula 7** follows an implementation-friendly expression of KL divergence. Mathematically, it is equivalent to the standard definition:
> $$
> {\text{KL}}(P \parallel Q) = \sum_{l}^{L} P_l \log \frac{P_l}{Q_l}
> $$
> For **numerical stability**, we expand this into the **log-space form** used for computation:
> $$
> \mathcal{L}{\text{align}} = \sum_{l}^{L} P_l \log \frac{P_l}{Q_l}= \sum_{l=1}^{L} \sum_{k=1}^{N^2} P_{l,k} (\log P_{l,k} - \log Q_{l,k})=\sum_{l=1}^{L} \sum_{k=1}^{N^2} e^{p_{l,k}} (p_{l,k} - q_{l,k})
> $$
> where $p_l=\log P_l = \log\text{softmax}(\text{Vec}(\text{Avg}(\text{MCM}^l)))$. The primary motivation for using this log-space formulation is numerical stability (avoid log(0) errors). By operating on log-probabilities (which are typically the direct output of a LogSoftmax layer), we entirely avoid the log(0) issue and maintain numerical precision.
>
> We compute $\text{KL}(P \parallel Q)$ rather than $\text{KL}(Q \parallel P)$, which is motivated by the distinct yet complementary nature of the two mechanisms. Specifically, the self-attention mechanism (P) captures **global dependencies among all tokens**, whereas the GNN focuses on modeling **relationships between adjacent nodes**. Therefore, we leverage the global dependency distribution learned in the shallow layers as a reference, compelling the local, neighbor-based dependencies learned by the GNN in the deep layers to align with this global view, **aiming to learn a more stable and structured dependency graph**. This process enforces a consistent dependency structure across different layers of the model.
> An additional theoretical analysis grounded in the Information Bottleneck principle is provided in **Appendix A.9 (Line 1007)**.
>
> We would be happy to make further revisions and engage in more discussions if you have any additional suggestions or ideas.
>
> [1] Dynamic spatial aware graph transformer for spatiotemporal traffic flow forecasting. Knowledge-based systems. (2024)
>
> [2] Sageformer: Series-aware framework for long-term multivariate time-series forecasting. IEEE Internet of Things Journal. (2024)
>
> [3] Msgnet: Learning multi-scale inter-series correlations for multivariate time series forecasting. In Proceedings of the AAAI conference on artificial intelligence. (2024)
>
> [4] Ada-MSHyper: Adaptive multi-scale hypergraph Transformer for time series forecasting. NIPS 2024
>
> [5] Multiple time series forecasting with dynamic graph modeling. VLDB 2024.

---

### Official Review · Reviewer_eqn5 · 2025-11-01

**Soundness:** 3
**Presentation:** 3
**Contribution:** 3
**Rating:** 6
**Confidence:** 5

**Summary:**

This paper addresses two limitations of Transformers in multivariate time series forecasting by proposing the CGTFra framework: (1) it introduces frequency-domain masking and resampling methods to replace positional encoding, thereby reducing dependence on timestamp information; (2) it incorporates a dynamic graph learning framework to explicitly model inter-variable dependencies in deeper network layers, addressing the limitation that existing methods only capture dependencies in shallow self-attention layers. Additionally, this paper is the first to propose a consistency alignment loss to constrain the dependency structures learned in both shallow and deep layers. The authors validate the effectiveness of their approach across 13 datasets, though some theoretical justifications and technical details require further elaboration.

**Strengths:**

1. The problem motivation is clear and well-supported by empirical evidence.
2. The authors are the first to propose modeling the dependency relationship between shallow and deep representations from a "consistency modeling" perspective, explicitly constraining their alignment using KL divergence.
3. The authors conduct comprehensive comparisons with 13 state-of-the-art methods across 13 datasets and validate the contribution of each module through ablation studies. The experimental design is thorough.

**Weaknesses:**

1. The theoretical justification for the equivalence between self-attention and GNNs (Appendix A.6) is intuitive. For the proposed consistency alignment loss, there is a lack of theoretical or mathematical guarantees—no formal bounds or convergence guarantees are provided to justify the rationality of this constraint.
2. The paper explains that the traffic dataset exhibits fixed periodic patterns, which accounts for why FMR underperforms the original iTransformer in Table 6. This suggests that FMR may excessively suppress periodicity, a characteristic that is very common in time series data. Have the authors considered adaptively adjusting the masking intensity based on the periodicity characteristics of the dataset to mitigate this issue?
3. Regarding performance degradation on large-variable datasets such as solar and traffic: the discussion of when DGL or CAL might lead to performance decline is somewhat superficial (limitations in Appendix A.16). The impact of these modules on large-variable datasets is mixed, sometimes degrading performance, but the paper lacks deeper analysis beyond speculation about "alignment challenges."
4. The authors need to clarify some issues in the methodology section; see the questions below.

**Questions:**

1. In Equations 5-6, how are the node embeddings $\Theta$ initialized? What is the purpose of $Concat(X^sa, \Theta)$?
2. Section 3.1 proposes learning independent frequency masks for each variable, but the paper lacks analysis of the learned masks: How much do the masks differ across variables? Do the masks tend to preserve low-frequency or high-frequency components? Is there a relationship with the periodicity of the data? Such analysis would enhance the interpretability of FRM.
3. Regarding the consistency alignment design, Figure 4 shows that the two mechanisms exhibit similarities but also some differences. Could the forced alignment of these two mechanisms through Equation 8 lead to information loss?

---

> ### Author Response · Authors · 2025-11-17
> **Response to Reviewer eqn5 (Part1)**
>
> We thank you very much for agreeing the clear motivation behind our proposed method. We are especially grateful for the valuable suggestions, which have greatly encouraged us to actively engage in this rebuttal and discuss the details of our work.
>
> **Q1: Theoretical or mathematical guarantees**: Many thanks to you for this insightful and challenging comment. While providing a formal convergence guarantee for Consistency Alignment Loss (CAL) is a complex, open research problem, we can provide a stronger theoretical foundation for our CAL by grounding it in established principles from information theory:
>
> The Information Bottleneck (IB) principle [1] aims to find a compressed representation, denoted as $Z$, that maximally preserves information about a target variable $Y$ while simultaneously compressing the input $X$. This objective is typically formulated as the following optimization problem:
> $$
> \max_{p(z|x)} \quad I(Z;Y) - \beta I(Z;X)
> $$
> where $I(⋅;⋅)$ represents mutual information and $\beta$ is a Lagrange multiplier.
>
> Within our CGTFra framework, **since self-attention computes inter-variable dependencies globally for each variable**, we can interpret the self-attention map (MCM) as a **high-bandwidth**, **yet potentially noisy**, representation of the inter-variable relationships in the input $X$ . While its mutual information with the input, $I(\text{MCM};X)$, is high, much of this information may constitute noise irrelevant to the final prediction target $Y$. Conversely, the GNN’s adjacency matrix, $A$, is intended to be the compressed and cleaner representation $Z$ that we seek to learn. $A$’s goal is to discard the noise present in MCM and retain only the structured information pertinent to predicting $Y$.
>
>  In this context, our alignment loss, $\mathcal{L}_{align}=\text{KL} (\text{MCM}\parallel A)$, **can be viewed as a proxy for the compression term**, $I(Z;X)$, in the IB objective. By minimizing $\text{KL} (\text{MCM}\parallel A)$, we encourage the learned adjacency matrix $A$ not to deviate excessively from the attention map MCM. This implicitly controls the mutual information $I(A;\text{MCM})$, and by extension, $I(A;X)$, aligning our method with the core IB principle of learning a compressed yet informative representation.
>
> (Detailed analysis has been updated in Appendix **A.9** (**Line 1007**))
>
> **Q2: Adaptively adjusting the masking intensity**: We sincerely appreciate your careful and logically well-founded review. We agree with your assessment that FMR may excessively suppress periodicity. Following your suggestion, we have introduced an enhancement to our FMR module to address this potential issue. In this improved version, we first analyze a single batch from the training set to capture the dataset’s relatively stable periodicities (For computational efficiency, we do not perform this operation on a per-batch basis). Based on this analysis, we apply a larger, fixed weight (e.g., 1.0, bypassing the mask) to the top-k dominant frequency components, while the remaining frequencies are still processed by a learnable mask.
> Through this approach, which we denote as **FMR+**, we mitigate the risk of FMR excessively suppressing critical periodic features. As a result, we observed performance improvements for both iTransformer and CASA on the Traffic dataset. The results are presented below:
> | Horizon | | | iTrans MSE | iTrans MAE | iTrans_FMR MSE | iTrans_FMR MAE | iTrans_FMR+ MSE | iTrans_FMR+ MAE | CASA MSE | CASA MAE | CASA_FMR MSE | CASA_FMR MAE | CASA_FMR+ MSE | CASA_FMR+ MAE |
> | :--- | :--- | :---: | :---: | :---: | :---: | :---: | :---: | :---: | :---: | :---: | :---: | :---: | :---: | :---: |
> | | | 96 | **0.392** | **0.268** | 0.393 | **0.268** |  **0.392** | **0.268** | **0.392** | **0.260** | 0.405 | 0.262 | 0.400 | 0.263 |
> | | | 192 | 0.413 | 0.277 |  0.413 | 0.276 | **0.412** | **0.275** | **0.415** | **0.274** | 0.432 | **0.274** | 0.424 | 0.276 |
> | | | 336 | **0.425** | 0.283 | 0.428 | **0.282** | 0.427 | **0.282** | **0.434** | 0.281 | 0.447 | **0.280** | 0.447| **0. 280** |
> | | | 720 | 0.459 | 0.300 | **0.458** | **0.299** | **0.458** | **0.299** | **0.468** | **0.296** | 0.492 | 0.301 | 0.486 | 0.301 |
> | | | **Avg** | **0.422** | 0.282 | 0.423 | **0.281** | **0.422** | **0.281** | **0.427** | **0.278** | 0.444 | 0.279 | 0.439 | 0.280 |
>
> **The code for FMR+ has been updated in our anonymized GitHub repository.**
>
> (see the **FrequencyMaskResampleCASA** and **FrequencyMaskResampleITrans** class in https://anonymous.4open.science/r/CGTFra/FMR_DGL_CAL/layers/Embed.py, and their scripts: bash ./FMR_DGL_CAL/scripts/CASA_scripts/Traffic_script/CASA_FMR_1115.sh and
> bash ./FMR_DGL_CAL/scripts/iTransformer/Traffic/iTransformer_1115.sh)
>
> Additionally, to further validate the effectiveness of our FMR, we have introduced a new method (**FilterNet** [5]) for comparison (see **Table 2** and **Table 11**).

---

> ### Author Response · Authors · 2025-11-17
> **Response to Reviewer eqn5 (Part2)**
>
> **Q3: Deeper analysis of alignment challenges**: We sincerely appreciate your insightful comments. We agree that our original discussion on the ‘alignment challenges’ were indeed insufficiently detailed. Motivated by your suggestion, we conducted a deeper analysis (see **Appendix A.19 (Line 1630)**) to uncover the causes of the performance degradation of our approach on datasets with a large number of variables. We posit that the limited performance improvement of the proposed DGL and CAL on large datasets may stem primarily from two factors.
>
> Primarily, **as the number of variables increases, the probability of spurious correlations between any pair of variables rises sharply**. Since the self-attention mechanism is designed to model relationships within an $N\times N$ matrix, it is forced to allocate attention weights across all variable pairs.
>
> In such a high-dimensional setting, these weights are more likely to capture sample-specific **incidental noise** rather than genuine, stable dependencies. When CAL is applied, it compels the adjacency matrix $A$ learned by DGL to align with this noisy Correlation Map (MCM), undermining its ability to perform effective message passing. **This can lead to worse performance than that of a simple FFN, which at least operates under a harmless ‘variable independence’ assumption.**
>
> **Q4: Node embeddings**: We initialize $\Theta$ using random initialization. In the operation $\text{Concat}(X^{sa,l},\Theta)$, the node embedding is concatenated with the input $X^{sa,l}$ to generate the gating signal. Since both $X$ and $\Theta$ are used jointly to construct the adjacency matrix (see Eq. (5)), our original motivation was to allow the gating decision process (i.e., $\text{ReLU}(\text{tanh}(\cdot))$) to depend simultaneously on the “static identity” of each variable and the “dynamic state” encoded in $X^{sa,l}$, enabling dynamic control of information flow. This design differs from models such as Sageformer [2] and MSGNet [3], which rely solely on fully learnable embeddings. Following your suggestion, we have revised the description accordingly (see **Line 264** and **Line 269**).
>
> To investigate the effects of (1) different initialization strategies for the $\Theta$ and (2) concatenating $X^{sa,l}$ with the $\Theta$ for gating, we conducted additional experiments on four heterogeneous datasets. Specifically, we evaluated three initialization schemes: Kaiming Uniform, Kaiming Normal, and Xavier Uniform (denoted as **Ours_KU**, **_KN**, and **_XU**, respectively). In addition, we removed $X^{sa,l}$ and used only $\Theta$ to generate the gating signal while keeping all other components unchanged (denoted as **RX**). Results are shown in the table below.
> | Dataset | Horizon | Ours MSE | Ours MAE | Ours_KU MSE | Ours_KU MAE | Ours_KN MSE | Ours_KN MAE | Ours_XU MSE | Ours_XU MAE | RX MSE | RX MAE |
> | :--- | :---: | :---: | :---: | :---: | :---: | :---: | :---: | :---: | :---: | :---: | :---: |
> | **ETTm1** | 96 | **0.315** | **0.344** | 0.317 | 0.345 | 0.318 | 0.346 | 0.322 | 0.348 | 0.320 | 0.347 |
> | | 192 | **0.366** | **0.372** | 0.372 | 0.377 | 0.369 | 0.374 | 0.372 | 0.376 | 0.368 | 0.373 |
> | | 336 | **0.398** | **0.395** | 0.403 | 0.397 | 0.404 | 0.398 | 0.405 | 0.399 | 0.414 | 0.404 |
> | | 720 | **0.472** | **0.435** | 0.483 | 0.439 | 0.473 | 0.435 | 0.481 | 0.438 | 0.467 | 0.432 |
> | | **AVG** | **0.388** | **0.386** | 0.394 | 0.390 | 0.391 | 0.388 | 0.395 | 0.390 | 0.392 | 0.389 |
> | **ETTh1** | 96 | **0.372** | **0.387** | 0.373 | 0.388 | 0.374 | 0.388 | 0.374 | 0.388 | 0.373 | 0.387 |
> | | 192 | **0.424** | **0.418** | 0.436 | 0.425 | 0.427 | 0.419 | 0.436 | 0.425 | 0.426 | 0.420 |
> | | 336 | **0.473** | **0.443** | 0.478 | 0.446 | 0.478 | 0.446 | 0.478 | 0.447 | 0.471 | 0.442 |
> | | 720 | **0.473** | **0.464** | 0.483 | 0.468 | 0.488 | 0.475 | 0.483 | 0.469 | 0.488 | 0.469 |
> | | **AVG** | **0.436** | **0.428** | 0.443 | 0.432 | 0.442 | 0.432 | 0.443 | 0.432 | 0.4395 | 0.430 |
> | **Weather** | 96 | **0.152** | **0.190** | 0.156 | 0.194 | 0.157 | 0.194 | 0.157 | 0.194 | 0.157 | 0.194 |
> | | 192 | **0.203** | **0.239** | 0.206 | 0.242 | 0.208 | 0.243 | 0.206 | 0.241 | 0.205 | 0.240 |
> | | 336 | **0.257** | **0.279** | 0.265 | 0.285 | 0.264 | 0.284 | 0.264 | 0.285 | 0.262 | 0.284 |
> | | 720 | **0.338** | **0.334** | 0.342 | 0.336 | 0.348 | 0.340 | 0.344 | 0.338 | 0.347 | 0.340 |
> | | **AVG** | **0.238** | **0.260** | 0.242 | 0.264 | 0.244 | 0.265 | 0.243 | 0.265 | 0.243 | 0.264 |
> | **ECL** | 96 | **0.137** | **0.227** | 0.138 | 0.228 | 0.139 | 0.228 | 0.139 | 0.229 | 0.139 | 0.229 |
> | | 192 | **0.155** | **0.243** | 0.155 | 0.243 | 0.156 | 0.243 | 0.155 | 0.243 | 0.156 | 0.244 |
> | | 336 | **0.170** | **0.259** | 0.170 | 0.259 | 0.171 | 0.260 | 0.170 | 0.260 | 0.169 | 0.260 |
> | | 720 | **0.198** | **0.283** | 0.216 | 0.297 | 0.212 | 0.295 | 0.217 | 0.300 | 0.199 | 0.284 |
> | | **AVG** | **0.165** | **0.253** | 0.170 | 0.257 | 0.169 | 0.256 | 0.170 | 0.258 | 0.166 | 0.254 |

---

> ### Author Response · Authors · 2025-11-17
> **Response to Reviewer eqn5 (Part3)**
>
> The performance degradation observed when using these three initialization methods may be attributed to the fact that Kaiming and Xavier initialization aim to create a well-conditioned starting state that enables stable propagation. This often places the node embeddings in a relatively smooth region of the optimization landscape with well-behaved gradients; however, such regions may correspond to broad but **shallow local optima**.
>
> Furthermore, when both the input $X^{sa,l}$ (output of self-attention layer) and the learnable node embeddings $\Theta$ are used to construct the adjacency matrix, incorporating both sources of information into the gating mechanism yields noticeably more robust performance.
>
> **Q5: Analysis of the learned masks**: Thanks for your valuable suggestions. To more clearly demonstrate the effectiveness of the variable-specific masks and thereby enhance the interpretability of FMR, we have added **Figure 12 (Line 884)** and corresponding analysis in **Appendix A.5 (Line 863)** in the revised manuscript.
>
> As shown in **Figure 12**, although different variables learn distinct masks, these masks still exhibit similarities that reflect inter-variable dependencies. For example, the masks learned for variables 0 and 2 display high similarity. Moreover, because we employ soft masks rather than binary (0,1) hard masks, each variable is able to adaptively determine whether to preserve low-frequency or high-frequency components. The visualizations in the Figure show that the **learned masks tend to preserve low-frequency components**—consistent with the fact that low frequencies capture more trend and periodicity [4]—while also exhibiting variable-specific suppression patterns across the spectrum. In particular, variable 3 shows noticeably stronger suppression of high-frequency components, which may be attributed to its higher noise level and more pronounced fluctuations. We hope these additions improve the interpretability of FMR presented in the manuscript.
>
> **Q6: Consistency alignment**: Thank you very much for this insightful comment. We agree that forced alignment may lead to potential information loss, as the proposed $L_{align}$ restricts the space in which DGL can capture IVD dependencies to some extent. However, this limitation is also intentional, aiming to constrain the optimization direction of DGL.
>
> **Figure 9 (Line 720)** illustrates that both self-attention and DGL capture IVD with their respective strengths and limitations. After introducing CAL, the modeled dependencies between variables become more reasonable, though some information loss does occur; for instance, in Figure 9(b), the dependency modeled by the GNN at coordinate (4,1) is lost (whereas PCC and DTW in Figure 1 indicate their dependency exists). Importantly, DGL still preserves certain critical associations (e.g., coordinate (5,4)). Moreover, the visualizations of dependencies for the Weather dataset in **Figure 5 (Line 445)** and **Figure 6** indicate that **the introduction of CAL enhances the ability to correctly model IVD**. Therefore, while CAL may introduce some information loss, the overall performance gain outweighs this drawback, which underscores the advantage of using CAL as a regularization term in the total loss. **Notably, this alignment mechanism is effective not only for our CGTFra but also improves existing sota methods such as iTransformer, DUET, and VCformer.**
>
> [1] The information bottleneck method. arXiv preprint physics/0004057. (2000)
>
> [2] SageFormer: Series-aware framework for long-term multivariate time-series forecasting. IEEE Internet of Things Journal. (2024)
>
> [3] Msgnet: Learning multi-scale inter-series correlations for multivariate time series forecasting. In Proceedings of the AAAI conference on artificial intelligence. (2024)
>
> [4] Fedformer: Frequency enhanced decomposed transformer for long-term series forecasting. ICML (2022)
>
> [5] Filternet: Harnessing frequency filters for time series forecasting. Advances in Neural Information Processing Systems. (2024)

---

### Official Review · Reviewer_zVp1 · 2025-11-03

**Soundness:** 3
**Presentation:** 2
**Contribution:** 3
**Rating:** 4
**Confidence:** 5

**Summary:**

The author defines the problem of inter-variate dependencies (IVD), referring to the loss of dependency information among variables in deep self-attention layers. To address this issue, the authors propose CGTFra, a framework designed to promote consistent modeling of inter-variate dependencies.

**Strengths:**

S1. The author makes a noteworthy observation that time-based positional encodings do not necessarily improve performance in multivariate time series forecasting.

S2. The author enhances forecasting accuracy by addressing the inconsistency between shallow and deep attention scores.

**Weaknesses:**

W1. It remains unclear whether directly integrating the adaptive adjacency matrix $A$ into the MCM would offer a more structurally concise design, thereby eliminating the need to optimize two separate objectives, i.e., $L_{align}$ and $L_{mae}$.

W2. The improvement in forecasting accuracy attributed to maintaining consistency appears to be based solely on empirical results. The author should discuss whether there is any theoretical foundation supporting this effect.

W3. The coordinates mentioned in the caption of Figure 4 do not align with the values shown in the figure, and the overall presentation appears inconsistent. The author should verify or clarify this issue, and it may be more effective to use multiple figures to illustrate the inconsistency across attention layers.

**Questions:**

See W1-W3.

---

> ### Author Response · Authors · 2025-11-17
> **Response to Reviewer zVp1 (Part1)**
>
> We sincerely thank you for your meticulous review and insightful analysis. We will now discuss the raised points in detail. In accordance with your suggestions, we have also made point-by-point revisions to our manuscript.
>
> **Q1: Integrating the adaptive adjacency matrix into the MCM**: That is a very insightful point, and we thank you for raising it. In response, we have conducted additional experiments to address this suggestion directly. Specifically, we modified our CGTFra framework by removing the DGL and CAL modules, **retaining only the FMR**. We then integrated the dynamically constructed graph (as defined in Line 267-Line 272) into the self-attention mechanism using two distinct fusion strategies:
> (1) Element-wise addition, where the graph structure serves as a **bias** to guide the attention scores. (2) Element-wise multiplication, where the graph acts as a **mask**.
> The comparative results are presented below (and have been added to **Table 14 (Line 1443)** in the revised manuscript):
>
> | Model | Variant | Horizon | ETTm1 MSE | ETTm1 MAE | ETTh1 MSE | ETTh1 MAE | Exchange MSE | Exchange MAE | Weather MSE | Weather MAE | ECL MSE | ECL MAE | Solar MSE | Solar MAE | Traffic MSE | Traffic MAE |
> | :--- | :--- | :---: | :---: | :---: | :---: | :---: | :---: | :---: | :---: | :---: | :---: | :---: | :---: | :---: | :---: | :---: |
> | CGTFra | original | 96 | 0.315 | 0.344 | 0.372 | **0.387** | **0.083** | **0.202** | **0.152** | **0.190** | **0.137** | **0.227** | **0.191** | **0.205** | **0.387** | **0.239** |
> | | | 192 | **0.366** | **0.372** |  **0.424** | **0.418** | **0.173** | **0.296** | **0.203** | **0.239** | **0.155** | **0.243** | **0.218** | **0.225** | **0.417** | **0.249** |
> | | | 336 | **0.398** | **0.395** | **0.473** | **0.443** | **0.324** | **0.412** | **0.257** | **0.279** | **0.170** | **0.259** | **0.238** | **0.240** | **0.434** | **0.261** |
> | | | 720 | **0.472** | **0.435** | **0.473** | **0.464** | **0.668** | **0.619** | **0.338** | **0.334** | **0.198** | **0.283** | 0.249 | **0.242** | **0.472** | **0.279** |
> | | | **Avg** | **0.388** | **0.386** | **0.436** | **0.428** | **0.312** | **0.382** | **0.238** | **0.260** | **0.165** | **0.253** | **0.224** | **0.228** | **0.427** | **0.257** |
> | CGTFra | shallow bias | 96 | 0.319 | 0.346 | 0.372 | 0.388 | 0.086 | 0.205 | 0.159 | 0.195 | 0.142 | 0.230 | 0.193 | 0.207 | 0.395 | 0.245 |
> | | | 192 | 0.370 | 0.375 | 0.435 | 0.423 | 0.179 | 0.301 | 0.211 | 0.244 | 0.158 | 0.245 | 0.221 | 0.227 | 0.419 | 0.254 |
> | | | 336 | 0.404 | 0.396 | 0.478 | 0.445 | 0.353 | 0.429 | 0.266 | 0.285 | **0.170** | **0.259** | 0.242 | 0.242 | 0.448 | 0.272 |
> | | | 720 | 0.501 | 0.445 | 0.487 | 0.473 | 0.797 | 0.675 | 0.345 | 0.339 | 0.201 | 0.285 | 0.248 | **0.242** | 0.485 | 0.284 |
> | | | **Avg** | 0.399 | 0.391 | 0.443 | 0.432 | 0.354 | 0.403 | 0.245 | 0.266 | 0.168 | 0.255 | 0.226 | 0.230 | 0.437 | 0.264 |
> | CGTFra | shallow mask | 96 | **0.314** | **0.343** | **0.371** | **0.387** | 0.086 | 0.205 | 0.162 | 0.198 | 0.144 | 0.231 | 0.197 | 0.212 | 0.408 | 0.252 |
> | | | 192 | 0.369 | 0.375 | 0.438 | 0.424 | 0.181 | 0.303 | 0.211 | 0.243 | 0.159 | 0.244 | 0.224 | 0.226 | 0.426 | 0.258 |
> | | | 336 | 0.414 | 0.401 | 0.484 | 0.446 | 0.338 | 0.421 | 0.266 | 0.284 | 0.174 | 0.261 | 0.246 | 0.242 | 0.449 | 0.270 |
> | | | 720 | 0.479 | 0.437 | 0.493 | 0.475 | 1.040 | 0.773 | 0.345 | 0.338 | 0.213 | 0.295 | **0.248** | **0.242** | 0.505 | 0.292 |
> | | | **Avg** | 0.394 | 0.389 | 0.447 | 0.433 | 0.411 | 0.426 | 0.246 | 0.266 | 0.173 | 0.258 | 0.229 | 0.231 | 0.447 | 0.268 |
>
> The results demonstrate that merely guiding the self-attention mechanism with dynamic graph information is insufficient to achieve superior performance. We attribute this to the fact that this approach fails to model inter-variable dependencies in the deeper layers of the network, a limitation we have previously discussed.
>
> In fact, as we noted in **Appendix A.10** (**Line 1043**), methods like DUET also employ a similar strategy of integrating captured dependencies into attention scores as a mask. However, the comprehensive performance of our CGTFra framework significantly surpasses that of DUET. This comparison in **Table 14** further underscore the necessity and effectiveness of consistently modeling inter-variable dependencies across both the shallow and deep layers of the network architecture. Corresponding analysis has been updated in **Appendix A.15 (Line 1458)**

---

> ### Author Response · Authors · 2025-11-17
> **Response to Reviewer zVp1 (Part2)**
>
> **Q2: Theoretical or mathematical guarantees**: We appreciate your interest in our proposed consistency alignment. Exploring its theoretical underpinnings has indeed enhanced the research depth of proposed CAL. After thorough discussion and a review of the relevant literature, we now provide an analysis of CAL’s effectiveness from the Information Bottleneck theory: Information Bottleneck (IB) principle [1] aims to find a compressed representation $Z$, that maximally preserves information about a target variable $Y$ while simultaneously compressing the input $X$. This objective is typically formulated as the following optimization problem:
> $$
> \max_{p(z|x)} \quad I(Z;Y) - \beta I(Z;X)
> $$
> where $I(⋅;⋅)$ represents mutual information and $\beta$ is a Lagrange multiplier.
>
> Within our CGTFra framework, we can interpret the self-attention map (MCM) as a high-bandwidth, yet potentially noisy, representation of the inter-variable relationships in the input $X$. While its mutual information with the input, $I(\text{MCM};X)$, is high, much of this information may constitute noise irrelevant to the final prediction target $Y$. Conversely, the GNN's adjacency matrix, $A$, is intended to be the compressed and cleaner representation $Z$ that we seek to learn. The goal is for $A$ to discard the noise present in MCM and retain only the structured information pertinent to predicting $Y$. In this context, our alignment loss, $\mathcal{L}_{align}=\text{KL} (\text{MCM}\parallel A)$, **can be viewed as a proxy or an upper bound for the compression term**, $I(Z;X)$, in the IB objective. By minimizing $\text{KL} (\text{MCM}\parallel A)$, we encourage the learned adjacency matrix $A$ not to deviate excessively from the attention map MCM. This implicitly controls the mutual information $I(A;\text{MCM})$, and by extension, $I(A;X)$, aligning our method with the core IB principle of learning a compressed yet informative representation.
>
> **Q3: Verify the overall presentation**: We sincerely apologize for any confusion this may have caused. Upon review, we believe the confusion may stem from the fact that we did not provide an explanation for the coordinates in Figure 4 (which has been renumbered to **Figure 9** in the revised manuscript). While we had clarified this in Figure 1—stating that “(x-axis, y-axis) correspond to variable indices”—we regret this omission in the latter Figure 9.
> We have double-checked the coordinates referenced in the caption of Figure 4 (now **Figure 9**), and we would like to offer a detailed explanation.
>
> For coordinate (5,4) in Figure 9(a) (the same logic applies to Figure 9(b)):
> In the Attention Score subfigure (left), the value at (5,4) is 0.13.
> In the Graph Structure subfigure (right), the value at (5,4) is 0.66.
> A value of 0.13 is relatively small within the context of the entire attention score matrix. To illustrate the issue, let us compare it with coordinate (1,0), which has an attention score of 0.14. The similarity in scores (0.13 vs. 0.14) suggests that the self-attention mechanism perceives the dependencies for pairs (5,4) and (1,0) as having nearly equal strength. However, the ground-truth metrics from **Figure 1** (PCC and DTW) reveal a significant disparity: the dependency for (5,4) (PCC=0.6, DTW=5.3) is substantially stronger than that for (1,0) (PCC=-0.5, DTW=14.2). **This discrepancy indicates that the dependencies captured by the standard self-attention mechanism lack sufficient discriminative power**.
> In contrast, the “Graph Structure” subfigure in Figure 9(a) assigns a high value of 0.66 to coordinate (5,4), clearly distinguishing it from weaker dependencies.
>
> What we aim to illustrate here is that self-attention and GNNs have distinct advantages and limitations in capturing inter-variable dependencies. This example not only serves as a validation for the effectiveness of using DGL to model these dependencies in deeper layers but also provides the motivation for introducing our CAL module. After the introduction of CAL, the KL divergence between the attention scores and the graph adjacency matrix in Figure 9(a) decreased from 0.026 to 0.0249.
> To improve clarity and ensure our description is unambiguous, we have supplemented the caption of this **Figure 9** with a more detailed explanation.
>
> **Q4: Use multiple figures to illustrate the inconsistency**: We sincerely thank you for this valuable suggestion, which we have thoroughly considered and analyzed. Accordingly, to further elucidate the inconsistency in modeling dependencies between the shallow and deep layers of the network, we have incorporated a detailed analysis of the Weather dataset into the main body of the manuscript (See **Figure 5 (Line 445)** and **Figure 6**). We hope this additional analysis reinforces the argument for the necessity of modeling inter-variate dependencies in deeper layers and maintaining consistency across the entire network.
>
> [1] The information bottleneck method. arXiv preprint physics/0004057. (2000)

---

### Author Response · Authors · 2025-11-17
**Summary of Revisions**

We sincerely thank all reviewers for their valuable feedback and insightful suggestions, which have significantly contributed to improving our manuscript.

In this work, we propose a **general-purpose Graph Transformer framework** dedicated to addressing the **inconsistent modeling of inter-variable dependencies (IVD)** in existing Variate Transformers. To the best of our knowledge, **this is the first work** to address the inconsistency problem between shallow and deep-layer IVD modeling. Grounded in the similarities and differences between self-attention and GNNs for capturing IVD, as well as the Information Bottleneck theory, we introduce a dynamic graph learning framework and a consistency alignment loss. This allows us to construct a general Graph Transformer framework that **enhances the predictive performance of existing Variate Transformers while incurring only a minimal, linear complexity overhead**.

We are pleased that the reviewers have acknowledged our manuscript with positive remarks such as “**noteworthy observation, clear motivation, interesting, fresh insights**” (Reviewers zVp1, eqn5, bgj4, xnWj) and “**well-supported, comprehensive comparisons, considerable detailed experiments, performance gains**” (Reviewers eqn5, bgj4, xnWj).
The reviewers raised highly constructive concerns, and we have dedicated five days of intensive work to address all comments and discussions. Below is a summary of the major revisions:
*   **Theoretical Foundation (Reviewers zVp1, eqn5)**: We thank the reviewer for agreeing with our theoretical analysis on the similarities and differences between self-attention and GNNs in capturing IVD (this analysis serves as the primary motivation for introducing dynamic graph learning (DGL) to model IVD in the deeper network layers ). We have provided a theoretical analysis based on the Information Bottleneck principle to justify the effectiveness and establish a theoretical upper bound for our consistency modeling (CAL) of shallow and deep-layer IVD.
*   **Manuscript Presentation (Reviewers bgj4, xnWj)**: We have performed point-by-point revisions to improve the manuscript’s rigor, including refining the presentation of results, clarifying our claims, enhancing the formalism of equations, and ensuring consistent highlighting (bolding) of the best-performing results.
*   **Comprehensive Experimental Evaluation (Reviewers zVp1, eqn5, bgj4, xnWj)**: We have conducted all requested experiments (encompassing 300 + scenarios). These include validating the necessity of consistency modeling, enhancing the FMR module for better performance, conducting ablation studies on node embeddings, verifying the consistency within CGTFra, and comparing against additional Graph Transformer and non-Transformer baselines to further validate the effectiveness of CGTFra and FMR.

We sincerely thank all reviewers for their constructive feedback. All major revisions in the manuscript have been highlighted in blue (In the Tables, blue text is used to highlight corrections made to the bold formatting), and we have **uploaded** the **revision manuscript**. We hope these changes have improved the quality of our manuscript.

---

### Meta-Review · Area_Chair_QZLi · 2026-01-02

**Summary:**

This paper targets an important problem: modeling inter-series dependencies in multivariate time series. However, major concerns focused on limited novelty, as the framework largely re-combines existing graph Transformer components with incremental design choices. The motivation for consistency was seen as underspecified, and theoretical justification was weak. Empirically, reviewers questioned baseline strength, fairness, and ablation depth, and felt gains were modest. Presentation quality and clarity of contributions also affected confidence in the work.

**Reviewer Concerns:**

The rebuttal helped clarify implementation details, training procedures, and some design motivations, which alleviated minor confusion around architecture and experiments. However, core concerns remain unresolved: the conceptual novelty of the framework relative to prior graph-based TS models is still limited. The notion of dependency “consistency” is not rigorously defined or validated, and empirical evidence remains insufficient to justify a general framework claim. Missing stronger baselines and deeper ablations (e.g., isolating key modules) continue to limit the paper’s impact.

**Reviewer Scores:**

Reviewer zVp1: Likely unchanged or slightly higher, as most technical and presentation concerns were well addressed.

Reviewer eqn5: Possibly unchanged or a slight decrease.

Reviewer bgj4: Likely unchanged, rebuttal addressed many points, but concerns about novelty and overstatement remain central.

Reviewer xnWj: Unlikely to substantially change score; despite clarifications, their skepticism about motivation, correctness, and necessity of IVD modeling appears fundamental rather than superficial.

---

### Decision · Program_Chairs · 2026-01-26

Reject